# Non-invertible higher-categorical symmetries

**Lakshya Bhardwaj[1], Lea E. Bottini[1], Sakura Schäfer-Nameki[1] and Apoorv Tiwari[2]**

**1** Mathematical Institute, University of Oxford, Andrew-Wiles Building,
Woodstock Road, Oxford, OX2 6GG, UK
**2** Department of Physics, KTH Royal Institute of Technology,
Stockholm, Sweden

## Abstract

We sketch a procedure to capture general non-invertible symmetries of a $d$-dimensional quantum field theory in the data of a higher-category, which captures the local properties of topological defects associated to the symmetries. We also discuss fusions of topological defects, which involve condensations/gaugings of higher-categorical symmetries localized on the worldvolumes of topological defects. Recently some fusions of topological defects were discussed in the literature where the dimension of topological defects seems to jump under fusion. This is not possible in the standard description of higher-categories. We explain that the dimension-changing fusions are understood as higher-morphisms of the higher-category describing the symmetry. We also discuss how a 0-form sub-symmetry of a higher-categorical symmetry can be gauged and describe the higher-categorical symmetry of the theory obtained after gauging. This provides a procedure for constructing non-invertible higher-categorical symmetries starting from invertible higher-form or higher-group symmetries and gauging a 0-form symmetry. We illustrate this procedure by constructing non-invertible 2-categorical symmetries in 4d gauge theories and non-invertible 3-categorical symmetries in 5d and 6d theories. We check some of the results obtained using our approach against the results obtained using a recently proposed approach based on 't Hooft anomalies.



# 1  Introduction

The most unexpected, generalized symmetries [1] thus far are those that relax the group multiplication structure, often referred to as *non-invertible symmetries*. After a long and prosperous history in spacetime dimensions $d = 2, 3$ [2–16], non-invertible symmetries characterized by topological operators satisfying a fusion-algebra (as opposed to a group law), have only very recently been started to be systematically studied in $d = 4$, especially in non-topological QFTs. The approaches used in [17–19] use mixed anomalies and duality defects to construct non-invertible symmetries in 4d gauge theories. In [20,21] arguments were provided to construct non-invertible defects in $O(2)$ gauge theories, and related theories, by gauging charge conjugation in $U(1)$ gauge theories. Recently, in [22], condensation defects (see also [23]) in 3d were discussed, which provide examples of non-invertible symmetries. For topological theories some work on non-invertible defects in higher dimensions can be found here [24,25].

In this paper we propose a general procedure, applicable in any dimension, which constructs non-invertible symmetries by gauging 0-form sub-symmetries of invertible higher-form and higher-group symmetries.

These non-invertible symmetries and their properties, such as the possible gaugings and analogs of 't Hooft anomalies, are expected to be encoded in the structure of a higher-category, which can be understood as capturing the local properties of topological defects associated to these symmetries. We can thus call these symmetries as *higher-categorical symmetries*. The most general symmetry structure of a $d$-dimensional QFT is given by a $(d-1)$-category. Some mathematics literature on these higher categories can be found in [26–29].

Our approach is inspired by the one in [30] in 3d (see also [31]), where 0-form global symmetries of TQFTs are gauged. We generalize this to any dimension as follows: the starting point of our analysis is a theory $\mathfrak{T}$, whose symmetry category of topological defects, satisfies the group law. We also assume the presence of a 0-form symmetry $G^{(0)}$, generated by topological defects of dimension $d - 1$: $D_{d-1}$. We furthermore consider situations, where these 0-form symmetries act as outer automorphisms, in particular inducing a non-trivial action on the lower dimensional topological defects $D_{d-(p+1)}$ that generate the $p$-form symmetries.

We then gauge this 0-form symmetry, and determine the higher-category that is obtained after gauging. One set of topological operators in the gauged theory $\mathfrak{T}/G$ are the invariant combinations of topological defects $D_{d-(p+1)}$ in the initial category. After gauging the 0-form symmetry, there will be additional topological line operators, that generate the dual symmetry. We develop a consistent framework to combine these two sets of defects and determine their fusions. The resulting structure is naturally a higher-category, with a fusion product defined at every level of the category.

Examples that we apply this method to are

- $\mathbb{Z}_2^{(0)}$ outer automorphism gauging of Spin($4N$) and Spin($4N + 2$) pure gauge theories in

3d and 4d, generalized also to any $d$.

- $\mathbb{Z}_2^{(0)}$ outer automorphism gauging of discrete abelian gauge theories in 3d and 4d, where $\mathbb{Z}_2^{(0)}$ acts as electromagnetic duality in 3d and 'layer/flavor swap' in 4d.

- $O(2)$ and $\widetilde{SU(N)}$ gauge theories in 4d

- $S^3$-gauging of Spin(8) gauge theory in 3d and 4d

- An example of a quiver gauge theory in 4d, where dihedral $D_8$ 0-form symmetry group is gauged.

- 6d absolute theories with supersymmetry

- 5d theories with supersymmetry

In the second part of the paper – starting with section 8 – we develop an alternative approach, which is closely related to the one proposed in [17], where the authors construct non-invertible symmetries starting from a 4$d$ theory $\mathfrak{T}$ which has a mixed anomaly of suitable type between a 0-form symmetry $\Gamma^{(0)}$ and a 1-form symmetry $\Gamma^{(1)}$. In particular, they consider an anomaly $\mathcal{A}$ linear in the background gauge field $A_1$ for the 0-form symmetry and quadratic in the background gauge field $B_2$ for the 1-form symmetry and argue that gauging the 1-form symmetry $\Gamma^{(1)}$ results in a theory $\mathfrak{T}' = \mathfrak{T}/\Gamma^{(1)}$ with non-invertible symmetries. Indeed, consider the codimension-1 topological defects $D_3^{(g)}$, $g \in \Gamma^{(0)}$, associated to the 0-form symmetry of $\mathfrak{T}$. The mixed anomaly $\mathcal{A}$ implies that $D_3^{(g)}$ is anomalous under background gauge transformations of $B_2$. Once we gauge the 1-form symmetry and go to $\mathfrak{T}'$, this becomes a dependence on dynamical fields that makes the 0-form symmetry defects ill-defined. We can still preserve the symmetry associated to these topological defects by stacking them with an appropriate 3d TQFT $\mathcal{X}^{(g)}$, which has itself an anomaly that can absorb the bulk dependency of $D_3^{(g)}$ and restore gauge invariance. The price to pay (or the bonus) is that the topological codimension-1 defects of $\mathfrak{T}'$, namely $\mathcal{D}_3^{(g)} = D_3^{(g)} \otimes \mathcal{X}^{(g)}$, no longer satisfy a group law, but a non-invertible fusion-like algebra.

Our starting point is either a theory with a mixed anomaly, or a discrete 2-group symmetry [32–40].[1] Several theories in this list are amenable also to the approach proposed by [17] that we have just described. We develop this approach in dimensions $d \leqslant 6$, and construct a variety of theories with non-invertible symmetries. In particular we will consider theories with 2-groups symmetries

$$\delta_{A_1} B_2 = \phi^* \Theta. \tag{1}$$

The approach using twist is applicable when the Postnikov class $\Theta = 0$. When $\Theta$ is not necessarily zero and we can gauge the 1-form symmetry associated to $B_2$, then the resulting theory has a mixed anomaly, and the approach in [17] is applicable. This however has limitations, as it requires the mixed anomaly to be linear in the background field, whose topological defect becomes non-invertible after gauging. Moreover this latter approach is somewhat computationally intense beyond $\mathbb{Z}_2$ gaugings, and is currently unknown to be applicable in the case of non-abelian discrete symmetries. On the contrary, the higher-category approach is applicable for both abelian and non-abelian gauging of 0-form symmetries.

Thus, both approaches have a range of applicability, with advantages and limitations. In this paper we will explore both approaches and cross-connect them whenever possible. This will provide an important cross-check for our construction. In this comparison with [17] (and

---

[1]Other examples of 2-groups symmetries in higher-dimensional QFTs have recently appeared in [41–46], which however have continuous 0-form flavor symmetries.

also the fusion in [20] for $O(2)$) it is also noteworthy that our approach will yield fusion structures at all levels of the higher-category: i.e. for the objects, and the $n$-morphisms, thus refining the fusion that includes topological defects of different dimension that was proposed in [17]. We will show how these two descriptions are compatible.

Another important phenomenon that we comment upon is the appearance of condensations of higher-form symmetries in the fusion of non-invertible defects on arbitrary sub-manifolds of spacetime, as observed in [17, 18]. We provide examples generalizing this phenomena where the fusion products involve generalized gaugings of the higher-categorical symmetries localized on topological defects.

We should make one clarifying comment: the symmetry categories considered in this paper are obtained by gauging a 0-form symmetry. The resulting symmetry category has topological defects which descend from the ungauged symmetry category, but also includes condensation defects. Fusion of the former can include condensation defects, which we include. However we do not discuss the fusion of condensation defects themselves. This is done in subsequent work [47–50]. The full symmetry categories including condensation defects and their fusion, in the examples we consider here are discussed in [48].

The plan of this paper is as follows: we begin in section 2 with a general discussion of higher categories and their relevance for symmetries in QFTs. In section 3, we discuss higher-categorical symmetries localized on the world-volumes of defects and their generalized gauging/condensation. The concrete setting of 0-form gauging of higher-categories is discussed in detail in section 4, both in 3d and in higher dimensions. The subsequent three sections 5, 6 and 7, contain a multitude of examples in 3d, 4d, and 5d/6d, respectively. Each example is constructed by gauging a 0-form symmetry and deriving the higher-categorical fusion in the gauged theory.

In section 8 we change gears and derive numerous non-invertible symmetries from 2-groups and mixed anomalies. This then is used as a comparison to the earlier higher-category approach. Finally we conclude and supply some appendices with computational details.

# 2 Symmetries and Higher-Categories

In this section, we review why generalized symmetries are expected to form the mathematical structure of a higher-category.

## 2.1 Symmetries in Terms of Topological Defects

Generalized symmetries of a QFT correspond to the existence of topological defects of various dimensions in the QFT. These topological defects can be genuine or non-genuine. We begin with a discussion of genuine topological defects, that can be defined independently of other higher-dimensional topological defects. A genuine topological defect $D_p$ of dimension-$p$ is a defect operator that can be inserted along any dimension-$p$ sub-manifold $\Sigma_p$ of the $d$-dimensional spacetime $M_d$. The fact that it is topological means the following: Consider a correlation function $\langle \cdots D_p(\Sigma_p) \cdots \rangle$ containing $D_p$, where the dots denote other topological and non-topological defects of various dimensions. Then, we have the equality of correlation functions

$$\left\langle \cdots D_p(\Sigma_p) \cdots \right\rangle = \left\langle \cdots D_p\left(\Sigma_p'\right) \cdots \right\rangle, \tag{2}$$

where $\left\langle \cdots D_p\left(\Sigma_p'\right) \cdots \right\rangle$ denotes the correlation function obtained by changing the locus of $D_p$ from $\Sigma_p$ to $\Sigma_p'$ by a homotopy that does not intersect the loci of other defects participating in the correlation function, and the loci of other defects are not changed.

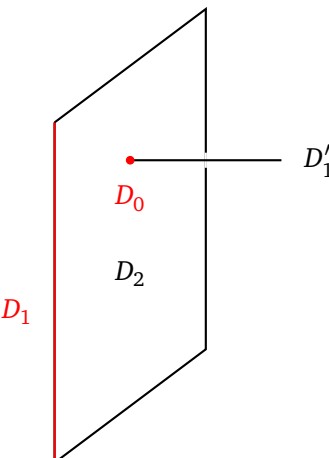

Figure 1: Example of non-genuine defects arising at junctions of genuine defects. Here $D_1'$ and $D_2$ are genuine line and surface defects respectively. $D_1$ is a non-genuine line defect arising at the end of $D_2$ and $D_0$ is a non-genuine local operator that can arise at an end of $D_1'$ along $D_2$.

Now, in order to discuss non-genuine topological defects, we begin by considering sub-defects arising at the intersections or junctions of genuine topological defects. See figure 1 for some examples. Consider a $p$-dimensional junction $\Sigma_p$ of genuine topological defects $D_{p_i}\left(\Sigma_{p_i}\right)$, where

$$\Sigma_p := \bigcap_i \Sigma_{p_i}. \tag{3}$$

We need to have $p_i > p$ for all $i$. There can be various kinds of sub-defects that can live at this junction $\Sigma_p$ for a fixed choice of $D_{p_i}$. In general, these include both topological and non-topological sub-defects, where a topological sub-defect $J_p$ satisfies

$$\left\langle \cdots \prod_i D_{p_i}\left(\Sigma_{p_i}\right) J_p(\Sigma_p) \cdots \right\rangle = \left\langle \cdots \prod_i D_{p_i}\left(\Sigma_{p_i}'\right) J_p\left(\Sigma_p'\right) \cdots \right\rangle, \tag{4}$$

which is an equality of correlation functions involving the configuration of defects $D_{p_i}$ and $J_p$, where $\Sigma_{p_i}'$ are related to $\Sigma_{p_i}$ by a homotopy that does not intersect the loci of other defects involved in the correlation function, and

$$\Sigma_p' := \bigcap_i \Sigma_{p_i}'. \tag{5}$$

Above is only a class of possible non-genuine topological defects. More generally, non-genuine topological defects arise at the junctions of genuine topological defects and non-genuine topological sub-defects arising at the junctions of genuine topological defects. See figure 2.

So far whatever we have discussed holds true for both discrete and continuous symmetries. A discrete symmetry is one for which the corresponding genuine and non-genuine topological defects are parametrized by discrete parameters. On the other hand, for a continuous symmetry, the corresponding genuine and non-genuine topological defects are parametrized by continuous parameters.

For a discrete symmetry, the associated topological defects and their configurations provide full information about the various possible backgrounds for the discrete symmetry that the QFT can be coupled to. However, for a continuous symmetry, the associated topological defects and their configurations only provide information about "flat" backgrounds of the continuous symmetry.

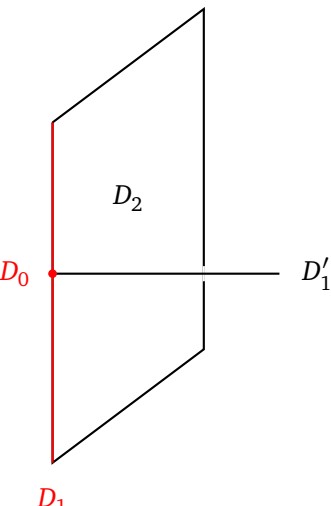

Figure 2: An example of a non-genuine defect arising at the junctions of genuine and other non-genuine defects. Here $D_0$ is a non-genuine local operator that can arise at an end of $D_1'$ along $D_1$, where $D_1'$ is a genuine line defect, while $D_1$ itself is a non-genuine line defect that can arise at an end of the genuine surface defect $D_2$.

## 2.2 From Topological Defects to Higher-Categories

**Symmetry category.** From the information about configurations of topological defects in a $d$-dimensional QFT $\mathfrak{T}$, we can construct a $(d-1)$-category $\mathcal{C}_{\mathfrak{T}}$, which we refer to as the symmetry category of $\mathfrak{T}$. For $d = 2$, it is a 1-category, or a standard category. For $d > 2$, it is a higher-category.

Recall that a $(d-1)$-category has $d$ levels. At the first level, we have objects of the category, which are also called 0-morphisms. At the second level, we have 1-morphisms between objects. At the third level, we have 2-morphisms between 1-morphisms. Continuing in this fashion, at the $i$-th level for $2 \leqslant i \leqslant d$, we have $(i-1)$-morphisms between $(i-2)$-morphisms.

**Objects.** The objects of $\mathcal{C}_{\mathfrak{T}}$ correspond to topological codimension-1 defects of $\mathfrak{T}$. We use the same labels $D_{d-1}$ to denote both topological codimension-1 defects and the corresponding objects of $\mathcal{C}_{\mathfrak{T}}$. There is an additive structure on the objects coming from the additive structure on the codimension-1 topological defects. A codimension-1 topological defect $D_{d-1} = \bigoplus_i n_i D_{d-1}^{(i)}$ with $n_i > 0$ is a sum of distinct codimension-1 topological defects $D_{d-1}^{(i)}$, which has the property that it has a total of $\sum_i n_i$ number of vacua, out of which in $n_i$ number of vacua it behaves like the defect $D_{d-1}^{(i)}$.

Simple objects are by definition those codimension-1 topological defects that have a single vacuum, or in other words, carry a single topological local operator on their worldvolume.

There is also a product/monoidal structure on the objects coming from fusing codimension-one topological defects. See figure 3, where we consider fusing two codimension-1 defects $D_{d-1}^{(1)}$ and $D_{d-1}^{(2)}$. The resulting codimension-1 defect is denoted as $D_{d-1}^{(12)}$, which we represent in equations as

$$D_{d-1}^{(1)} \otimes D_{d-1}^{(2)} = D_{d-1}^{(12)}, \tag{6}$$

or as

$$D_{d-1}^{(1)}(\Sigma_{d-1}) \otimes D_{d-1}^{(2)}(\Sigma_{d-1}) = D_{d-1}^{(12)}(\Sigma_{d-1}), \tag{7}$$

if we want to manifest the codimension-one submanifold $\Sigma_{d-1}$ of spacetime that the defects wrap.

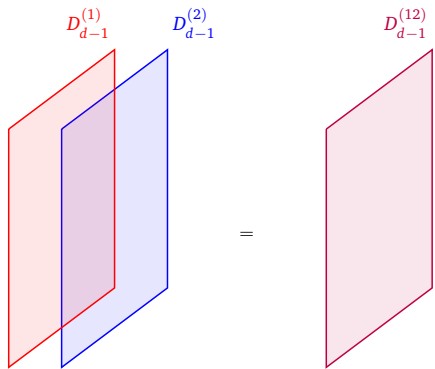

Figure 3: Fusion of codimension-1 topological defects that describes a monoidal structure on the objects in the symmetry higher-category.

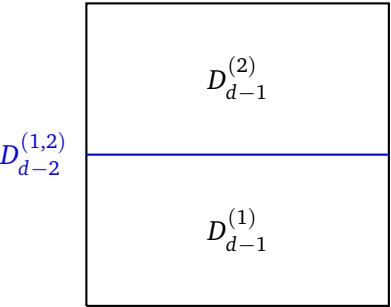

Figure 4: A 1-morphism $D_{d-2}^{(1,2)}$ from $D_{d-1}^{(1)}$ to $D_{d-1}^{(2)}$ is a codimension-2 topological defect living between codimension-1 topological defects $D_{d-1}^{(1)}$ and $D_{d-1}^{(2)}$. To specify the direction of the morphism, we need to pick a "time" direction, which is taken to run from bottom to top of the figure.

**1-morphisms.** The 1-morphisms of $\mathcal{C}_{\mathfrak{T}}$ correspond to topological codimension-2 defects living at the intersection of two topological codimension-1 defects. More precisely, a topological codimension-2 defect $D_{d-2}$ living between codimension-1 defects $D_{d-1}$ and $D'_{d-1}$ (with suitable choice of orientations) corresponds to a 1-morphism from $D_{d-1}$ to $D'_{d-1}$. See figure 4. There is an additive structure on 1-morphisms: Let $D_{d-2}^{(i)}$ be distinct 1-morphisms from fixed object $D_{d-1}$ to fixed object $D'_{d-1}$. Then

$$D_{d-2} := \bigoplus_i n_i D_{d-2}^{(i)}, \qquad (8)$$

for $n_i \geqslant 0$ is also a 1-morphism from $D_{d-1}$ to $D'_{d-1}$, which has (for each value of $i$) $n_i$ number of vacua in which it behaves like defect $D_{d-2}^{(i)}$.

Two 1-morphisms can be composed to obtain another 1-morphism. Given a 1-morphism $D_{d-2}^{(1,2)}$ from $D_{d-1}^{(1)}$ to $D_{d-1}^{(2)}$ and a 1-morphism $D_{d-2}^{(2,3)}$ from $D_{d-1}^{(2)}$ to $D_{d-1}^{(3)}$, we have a 1-morphism

$$D_{d-2}^{(2,3)} \circ D_{d-2}^{(1,2)}, \qquad (9)$$

from $D_{d-1}^{(1)}$ to $D_{d-1}^{(3)}$. This composition operation describes fusion of $D_{d-2}^{(1,2)}$ and $D_{d-2}^{(2,3)}$ along a codimension-1 locus containing all three codimension-1 defects $D_{d-1}^{(1)}$, $D_{d-1}^{(2)}$ and $D_{d-1}^{(3)}$. See figure 5.

Changing the time direction in the above fusion leading to composition of morphisms, we obtain a monoidal/fusion structure on 1-morphisms. However, it should be noted that we

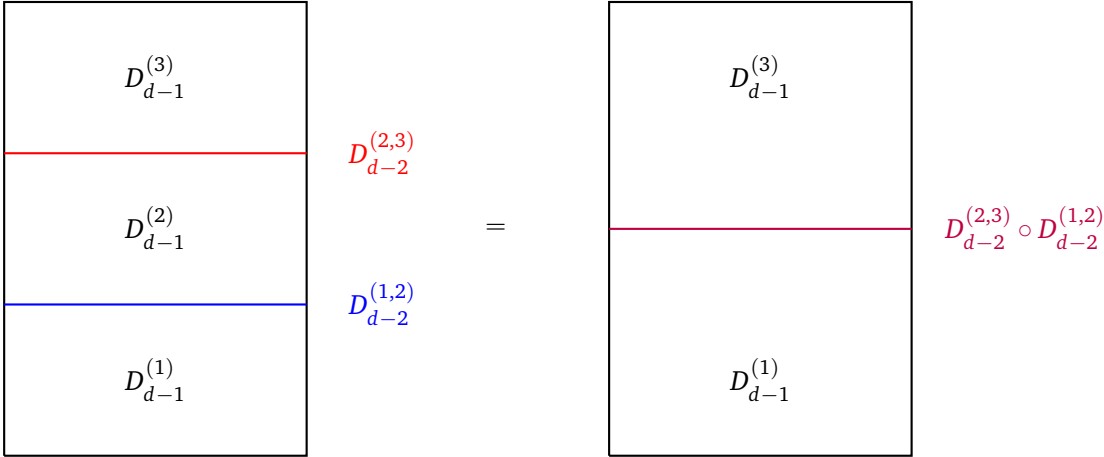

Figure 5: Fusing two codimension-2 defects $D_{d-2}^{(1,2)}$ and $D_{d-2}^{(2,3)}$ leads to the defect $D_{d-2}^{(2,3)} \circ D_{d-2}^{(1,2)}$. This is described in the higher-category as a composition of 1-morphisms, and to describe the direction of the morphisms and composition, we need to pick a "time" direction, which is taken to run from bottom to top of the figure.

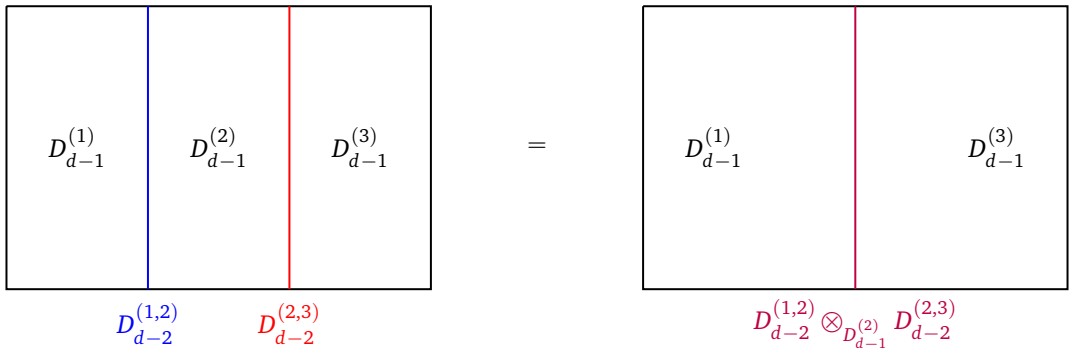

Figure 6: Here we have rotated the figure 5, while keeping the time direction going from bottom to top. The fusion of $D_{d-2}^{(1,2)}$ and $D_{d-2}^{(2,3)}$ now is represented as a monoidal operation on 1-morphisms. Such a monoidal operation is labeled by objects, as in equation (10).

define this fusion structure only if $\mathcal{C}_{\mathfrak{T}}$ admits 2-morphisms, i.e. if the theory $\mathfrak{T}$ has dimension $d \geqslant 3$. Given a 1-morphism $D_{d-2}^{(1,2)}$ from $D_{d-1}^{(1)}$ to $D_{d-1}^{(2)}$ and a 1-morphism $D_{d-2}^{(2,3)}$ from $D_{d-1}^{(2)}$ to $D_{d-1}^{(3)}$, we have a 1-morphism

$$D_{d-2}^{(1,2)} \otimes_{D_{d-1}^{(2)}} D_{d-2}^{(2,3)}, \tag{10}$$

from $D_{d-1}^{(1)}$ to $D_{d-1}^{(3)}$. See figure 6. Even though we have

$$D_{d-2}^{(1,2)} \otimes_{D_{d-1}^{(2)}} D_{d-2}^{(2,3)} = D_{d-2}^{(2,3)} \circ D_{d-2}^{(1,2)}, \tag{11}$$

we use both notions as they have different utilities. For example, we will see later that the fusion structure $\otimes_{D_{d-1}}$ on 1-morphisms from $D_{d-1}$ to $D_{d-1}$ descends to a fusion structure on objects of a higher-category of symmetries localized along $D_{d-1}$.

There is another fusion structure on 1-morphisms, which is defined for any $\mathcal{C}_{\mathfrak{T}}$, irrespective of whether it admits 2-morphisms or not. Given a 1-morphism $D_{d-2}^{(1,2)}$ from $D_{d-1}^{(1)}$ to $D_{d-1}^{(2)}$ and

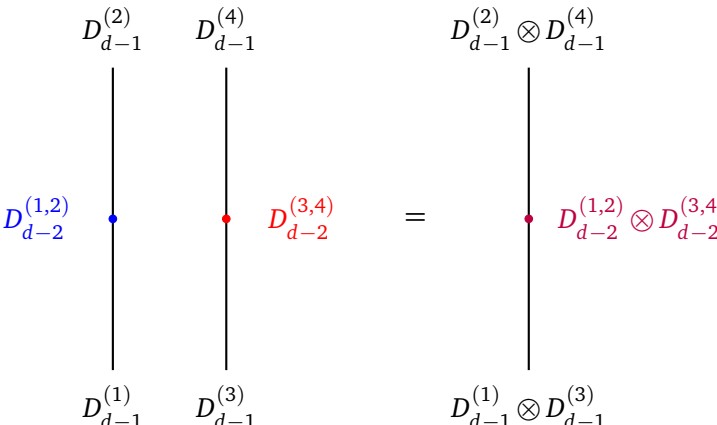

Figure 7: The fusion structure $\otimes$ on general codimension-2 topological defects.

a 1-morphism $D_{d-2}^{(3,4)}$ from $D_{d-1}^{(3)}$ to $D_{d-1}^{(4)}$ constructs a 1-morphism

$$D_{d-2}^{(1,2)} \otimes D_{d-2}^{(3,4)} \,, \tag{12}$$

from $D_{d-1}^{(13)}$ to $D_{d-1}^{(24)}$, where

$$\begin{aligned} D_{d-1}^{(13)} &:= D_{d-1}^{(1)} \otimes D_{d-1}^{(3)} \,, \\ D_{d-1}^{(24)} &:= D_{d-1}^{(2)} \otimes D_{d-1}^{(4)} \,. \end{aligned} \tag{13}$$

This fusion operation is described in figure 7.

**2-morphisms.** The 2-morphisms of $\mathcal{C}_{\mathfrak{T}}$ correspond to topological codimension-3 defects living at the intersection of two codimension-2 defects corresponding to 1-morphisms of $\mathcal{C}_{\mathfrak{T}}$. More precisely, consider two codimension-2 defects $D_{d-2}^{(1,2),(1)}$ and $D_{d-2}^{(1,2),(2)}$ both acting as 1-morphisms from the codimension-1 defect $D_{d-1}^{(1)}$ to the codimension-1 defect $D_{d-1}^{(2)}$. Then, 2-morphisms from $D_{d-2}^{(1,2),(1)}$ to $D_{d-2}^{(1,2),(2)}$ correspond to codimension-3 defects that live at the intersection of $D_{d-2}^{(1,2),(1)}$ and $D_{d-2}^{(1,2),(2)}$. See figure 8. There is again an additive structure on 2-morphisms similar to that for 1-morphisms and 0-morphisms discussed above. We can compose a 2-morphism $D_{d-3}^{(1,2)}$ from 1-morphism $D_{d-2}^{(1)}$ to 1-morphism $D_{d-2}^{(2)}$ with a 2-morphism $D_{d-3}^{(2,3)}$ from $D_{d-2}^{(2)}$ to 1-morphism $D_{d-2}^{(3)}$, to obtain a 2-morphism

$$D_{d-3}^{(2,3)} \circ D_{d-3}^{(1,2)} \,, \tag{14}$$

from $D_{d-2}^{(1)}$ to $D_{d-2}^{(3)}$.

There are again multiple fusion structures we can define. For any arbitrary $\mathcal{C}_{\mathfrak{T}}$ containing 2-morphisms, i.e. for any $\mathfrak{T}$ having $d \geqslant 3$, we have a fusion structure on 2-morphisms, which we denote by $\otimes$. Consider a 2-morphism $D_{d-3}^{(1,2),(1,2)}$ from a 1-morphism $D_{d-2}^{(1,2),(1)}$ to a 1-morphism $D_{d-2}^{(1,2),(2)}$, where each 1-morphism $D_{d-2}^{(1,2),(i)}$ is from an object $D_{d-1}^{(1)}$ to an object $D_{d-1}^{(2)}$. Similarly, consider another 2-morphism $D_{d-3}^{(3,4),(1,2)}$ from a 1-morphism $D_{d-2}^{(3,4),(1)}$ to a 1-morphism $D_{d-2}^{(3,4),(2)}$, where each 1-morphism $D_{d-2}^{(3,4),(i)}$ is from an object $D_{d-1}^{(3)}$ to an object $D_{d-1}^{(4)}$. Then, the 2-morphism

$$D_{d-3}^{(1,2),(1,2)} \otimes D_{d-3}^{(3,4),(1,2)} \tag{15}$$

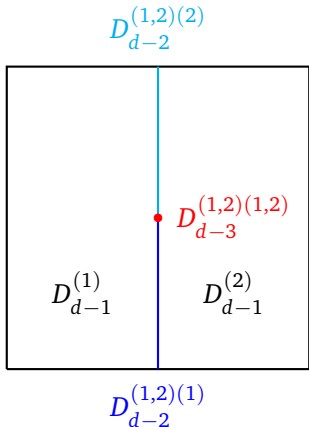

Figure 8: A 2-morphism $D_{d-3}^{(1,2)(1,2)}$ between 1-morphisms $D_{d-2}^{(1,2),(1)}$ and $D_{d-2}^{(1,2),(2)}$ (both from $D_{d-1}^{(1)}$ to $D_{d-1}^{(2)}$).

is from the 1-morphism $D_{d-2}^{(1,2),(1)} \otimes D_{d-2}^{(3,4),(1)}$ to the 1-morphism $D_{d-2}^{(1,2),(2)} \otimes D_{d-2}^{(3,4),(2)}$, where each 1-morphism $D_{d-2}^{(1,2),(i)} \otimes D_{d-2}^{(3,4),(i)}$ is from the object $D_{d-1}^{(1)} \otimes D_{d-1}^{(3)}$ to the object $D_{d-1}^{(2)} \otimes D_{d-1}^{(4)}$.

Similarly, for any arbitrary $\mathcal{C}_{\mathfrak{T}}$ containing 2-morphisms, i.e. for any $\mathfrak{T}$ having $d \geqslant 3$, we have another fusion structure on 2-morphisms which is parametrized by objects of $\mathcal{C}_{\mathfrak{T}}$. Consider a 2-morphism $D_{d-3}^{(1,2),(1,2)}$ from a 1-morphism $D_{d-2}^{(1,2),(1)}$ to a 1-morphism $D_{d-2}^{(1,2),(2)}$, where each 1-morphism $D_{d-2}^{(1,2),(i)}$ is from an object $D_{d-1}^{(1)}$ to an object $D_{d-1}^{(2)}$. Similarly, consider another 2-morphism $D_{d-3}^{(2,3),(1,2)}$ from a 1-morphism $D_{d-2}^{(2,3),(1)}$ to a 1-morphism $D_{d-2}^{(2,3),(2)}$, where each 1-morphism $D_{d-2}^{(2,3),(i)}$ is from the object $D_{d-1}^{(2)}$ to an object $D_{d-1}^{(3)}$. Then, the 2-morphism

$$D_{d-3}^{(1,2),(1,2)} \otimes_{D_{d-1}^{(2)}} D_{d-3}^{(2,3),(1,2)} \tag{16}$$

is from the 1-morphism $D_{d-2}^{(1,2),(1)} \otimes_{D_{d-1}^{(2)}} D_{d-2}^{(2,3),(1)}$ to the 1-morphism $D_{d-2}^{(1,2),(2)} \otimes_{D_{d-1}^{(2)}} D_{d-2}^{(2,3),(2)}$, where each 1-morphism $D_{d-2}^{(1,2),(i)} \otimes_{D_{d-1}^{(2)}} D_{d-2}^{(2,3),(i)}$ is from the object $D_{d-1}^{(1)}$ to the object $D_{d-1}^{(3)}$.

Now, if $\mathcal{C}_{\mathfrak{T}}$ contains 3-morphisms, i.e. if $\mathfrak{T}$ is a theory in $d \geqslant 4$, then we have a third fusion structure, which is parametrized by 1-morphisms of $\mathcal{C}_{\mathfrak{T}}$. Consider a 2-morphism $D_{d-3}^{(1,2),(1,2)}$ from a 1-morphism $D_{d-2}^{(1,2),(1)}$ to a 1-morphism $D_{d-2}^{(1,2),(2)}$, where each 1-morphism $D_{d-2}^{(1,2),(i)}$ is from an object $D_{d-1}^{(1)}$ to an object $D_{d-1}^{(2)}$. Similarly, consider another 2-morphism $D_{d-3}^{(1,2),(2,3)}$ from the 1-morphism $D_{d-2}^{(1,2),(2)}$ to another 1-morphism $D_{d-2}^{(1,2),(3)}$, where $D_{d-2}^{(1,2),(3)}$ is also from the object $D_{d-1}^{(1)}$ to the object $D_{d-1}^{(2)}$. Then, the 2-morphism

$$D_{d-3}^{(1,2),(1,2)} \otimes_{D_{d-2}^{(1,2),(2)}} D_{d-3}^{(1,2),(2,3)} \tag{17}$$

is from the 1-morphism $D_{d-2}^{(1,2),(1)}$ to the 1-morphism $D_{d-2}^{(1,2),(3)}$. See figure 9. It should be noted that

$$D_{d-3}^{(1,2),(1,2)} \otimes_{D_{d-2}^{(1,2),(2)}} D_{d-3}^{(1,2),(2,3)} = D_{d-3}^{(1,2),(2,3)} \circ D_{d-3}^{(1,2),(1,2)} . \tag{18}$$

**Higher-morphisms.** Continuing inductively, we define $p$-morphisms from $p-1$-morphism $D_{d-p}$ to $p-1$-morphism $D'_{d-p}$ of $\mathcal{C}_{\mathfrak{T}}$ as topological codimension-$(p+1)$ defects $D_{d-p-1}$ that

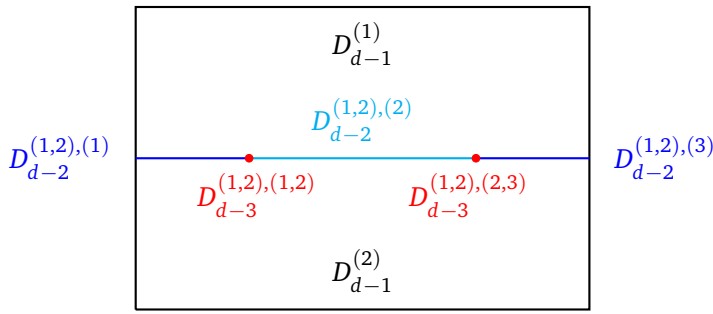

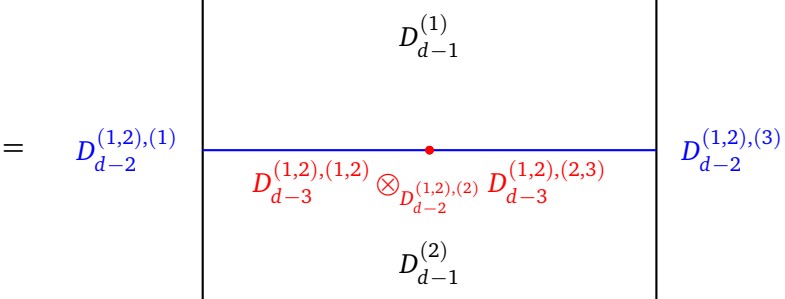

Figure 9: A 2-morphism fusion in $D_{d-2}^{(1,2),(2)}$: $D_{d-3}^{(1,2),(1,2)} \otimes_{D_{d-2}^{(1,2),(2)}} D_{d-3}^{(1,2),(2,3)}$.

live at the intersection of topological codimension-$p$ defects $D_{d-p}$ and $D'_{d-p}$ (with appropriate choices of orientation). There is an additive structure and composition on $p$-morphisms. For an arbitrary $\mathcal{C}_{\mathfrak{T}}$ admitting $p$-morphisms, i.e. for any theory $\mathfrak{T}$ of $d \geqslant p+1$, we can define many kinds of fusion structures on $p$-morphisms: a fusion structure $\otimes$, fusion structures $\otimes_{D_{d-1}}$ parametrized by objects of $\mathcal{C}_{\mathfrak{T}}$, fusion structures $\otimes_{D_{d-2}}$ parametrized by 1-morphisms of $\mathcal{C}_{\mathfrak{T}}$, and so on upto fusion structures $\otimes_{D_{d-p+1}}$ parametrized by $(p-2)$-morphisms of $\mathcal{C}_{\mathfrak{T}}$. If $\mathcal{C}_{\mathfrak{T}}$ admits $(p+1)$-morphisms, i.e. if $\mathfrak{T}$ has dimension $d \geqslant p+2$, then we can also define a fusion structure on $p$-morphisms parametrized by $(p-1)$-morphisms of $\mathcal{C}_{\mathfrak{T}}$, which is the same as composition of $p$-morphisms.

## 3  Localized Symmetries and Condensations

Suppose we are provided two topological defects $D_p^{(1)}$ and $D_p^{(2)}$ with a topological junction $D_{p-1}^{(1,2)}$ between them, such that wrapping the junction $D_{p-1}^{(1,2)}$ on a sphere $S^{p-1}$ is proportional to not wrapping it, as shown in figure 10. Then, we say that $D_p^{(1)}$ and $D_p^{(2)}$ are related by a *condensation*. See [23] for a general discussion of condensations.

This lets us define equivalence classes of topological defects[2] that are related to each other by condensations. Pick a representative $D_p^{(1)}$ of such an equivalence class. Then any other defect $D_p^{(2)}$ lying in the equivalence class can be obtained by performing a generalized gauging operation on the worldvolume of $D_p^{(1)}$. Moreover, all the topological sub-defects of $D_p^{(2)}$ can be obtained from topological sub-defects of $D_p^{(1)}$. The purpose of this section is to explain this generalized gauging construction.

---

[2]These are also known as 'Schur components' [51].

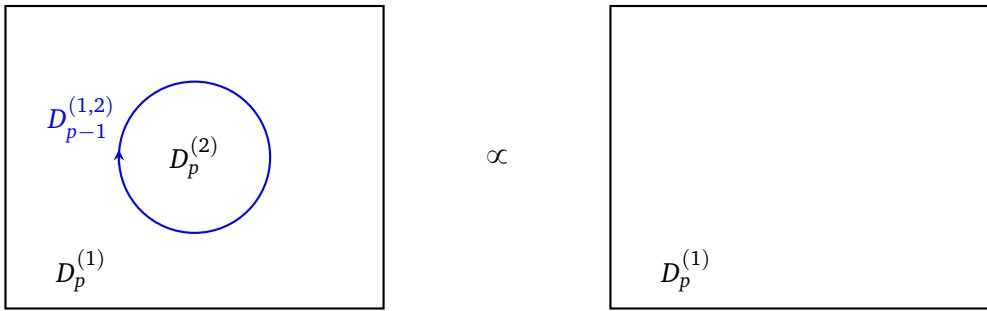

Figure 10: Two topological defects $D_p^{(1)}$ and $D_p^{(2)}$ are related by condensation if there exists a topological junction $D_{p-1}^{(1,2)}$ which can be bubbled out of nothing at the cost of changing the correlation function by an overall constant non-zero number.

## 3.1 Symmetries Localized Along Topological Defects

To describe the generalized gauging operation, we need to first begin with a discussion of the (higher-)category $\mathcal{C}_{\mathfrak{T},D_p}$ of symmetries localized along the worldvolume of a topological defect $D_p$ (which may be genuine or non-genuine). $\mathcal{C}_{\mathfrak{T},D_p}$ is a $(p-1)$-category describing topological defects that are constrained to live inside $D_p$, and we refer to it as the *symmetry category of the defect $D_p$*.

In fact $\mathcal{C}_{\mathfrak{T},D_p}$ can be recognized as a subcategory of the symmetry $(d-1)$-category $\mathcal{C}_{\mathfrak{T}}$ of the theory $\mathfrak{T}$. The defect $D_p$ is itself a $(d-p-1)$-morphism of $\mathcal{C}_{\mathfrak{T}}$. The objects of $\mathcal{C}_{\mathfrak{T},D_p}$ are $(d-p)$-morphisms of $\mathcal{C}_{\mathfrak{T}}$ from $D_p$ to itself. The 1-morphisms of $\mathcal{C}_{\mathfrak{T},D_p}$ are $(d-p+1)$-morphisms of $\mathcal{C}_{\mathfrak{T}}$ going between $(d-p)$-morphisms of $\mathcal{C}_{\mathfrak{T}}$ that are objects of $\mathcal{C}_{\mathfrak{T},D_p}$. Proceeding inductively, the $q$-morphisms of $\mathcal{C}_{\mathfrak{T},D_p}$ are $(d-p+q)$-morphisms of $\mathcal{C}_{\mathfrak{T}}$ going between $(d-p+q-1)$-morphisms of $\mathcal{C}_{\mathfrak{T}}$ that are $(q-1)$-morphisms of $\mathcal{C}_{\mathfrak{T},D_p}$. The additive and composition structures on $\mathcal{C}_{\mathfrak{T},D_p}$ descend from those on $\mathcal{C}_{\mathfrak{T}}$.

The fusion structure $\otimes$ on $\mathcal{C}_{\mathfrak{T},D_p}$ descends from the fusion structure $\otimes_{D_p}$ on $\mathcal{C}_{\mathfrak{T}}$. The fusion structures on $\mathcal{C}_{\mathfrak{T},D_p}$ parametrized by $q$-morphisms (where $q \geqslant 0$) of $\mathcal{C}_{\mathfrak{T},D_p}$ descend from fusion structures on $\mathcal{C}_{\mathfrak{T}}$ parametrized by the $(d-p+q)$-morphisms of $\mathcal{C}_{\mathfrak{T}}$ that are associated to $q$-morphisms of $\mathcal{C}_{\mathfrak{T},D_p}$.

## 3.2 Generalized Gauging: $p = 2$

Let us now describe the construction of $D_p^{(2)}$ in terms of $D_p^{(1)}$, when the two defects are related by condensation. We will first discuss the case of $p = 2$, where we can be quite concrete. Later we will sketch the case of general $p$, where we will not be so concrete.

$D_2^{(2)}$ can be obtained from $D_2^{(1)}$ by performing a generalized gauging [8, 52, 53] of the symmetry $\mathcal{C}_{\mathfrak{T},D_2^{(1)}}$ of $D_2^{(1)}$. The gauging is described by what is known as an algebra inside the 1-category $\mathcal{C}_{\mathfrak{T},D_2^{(1)}}$. The algebra is comprised of the following data:

- First of all, we have an object $A_1^{(1,2)}$ inside $\mathcal{C}_{\mathfrak{T},D_2^{(1)}}$, which can be constructed as

$$A_1^{(1,2)} = D_1^{(1,2)} \otimes D_1^{(2,1)}, \tag{19}$$

where $D_1^{(1,2)}$ is the junction lines between $D_2^{(1)}$ and $D_2^{(2)}$ discussed above that is responsible for condensation, and $D_1^{(2,1)}$ is the line obtained by reversing the orientation of $D_1^{(1,2)}$. See figure 11.

- Additionally we have the following canonical morphisms

$$
\begin{aligned}
A_0^{(1,2;p)} &: A_1^{(1,2)} \otimes A_1^{(1,2)} \to A_1^{(1,2)}, \\
A_0^{(1,2;cp)} &: A_1^{(1,2)} \to A_1^{(1,2)} \otimes A_1^{(1,2)}, \\
A_0^{(1,2;ev)} &: A_1^{(1,2)} \to 1_{D_2^{(1)}}, \\
A_0^{(1,2;cev)} &: 1_{D_2^{(1)}} \to A_1^{(1,2)},
\end{aligned}
\tag{20}
$$

which are constructed from $D_1^{(1,2)}$ and $D_1^{(2,1)}$ as shown in figure 12, and satisfy the properties shown in figure 13.

The gauging of $\mathcal{C}_{\mathfrak{T},D_2^{(1)}}$ by the algebra

$$
A^{(1,2)} = \left\{ A_1^{(1,2)}, A_0^{(1,2;p)}, A_0^{(1,2;cp)}, A_0^{(1,2;ev)}, A_0^{(1,2;cev)} \right\},
\tag{21}
$$

is performed by inserting a mesh of topological defects comprised out of algebra along the full locus of $D_1^{(1,2)}$, as shown in figure 14. We denote the defect with algebra $A^{(1,2)}$ condensed by

$$
D_2^{(2)} = \frac{D_2^{(1)}}{A^{(1,2)}}.
\tag{22}
$$

Above, we used $D_1^{(1,2)}$ and $D_1^{(2,1)}$ to construct the algebra $A^{(1,2)}$. Conversely, we can construct $D_1^{(1,2)}$ and $D_1^{(2,1)}$ using the algebra $A^{(1,2)}$, by inserting a mesh of topological defects comprised out of algebra along half-of the locus of $D_1^{(1)}$, as shown in figure 15.

**Category of lines after condensation.** The symmetry category capturing localized symmetries on $D_2^{(2)}$ can be recognized as

$$
\mathcal{C}_{\mathfrak{T},D_2^{(2)}} = \mathrm{Bimod}_{A^{(1,2)}} \left( \mathcal{C}_{\mathfrak{T},D_2^{(1)}} \right),
\tag{23}
$$

which is the category of $A^{(1,2)}$ *bimodules* in $\mathcal{C}_{\mathfrak{T},D_2^{(1)}}$. That is, the topological line operators living on $D_2^{(2)}$ are bimodules of the algebra $A^{(1,2)}$. Such a bimodule $B^{D_2^{(1)}}$ comprises of the following data

$$
B^{D_2^{(1)}} = \left\{ B_1^{D_2^{(1)}}, B_0^{D_2^{(1)};lp}, B_0^{D_2^{(1)};rp}, B_0^{D_2^{(1)};lcp}, B_0^{D_2^{(1)};rcp} \right\},
\tag{24}
$$

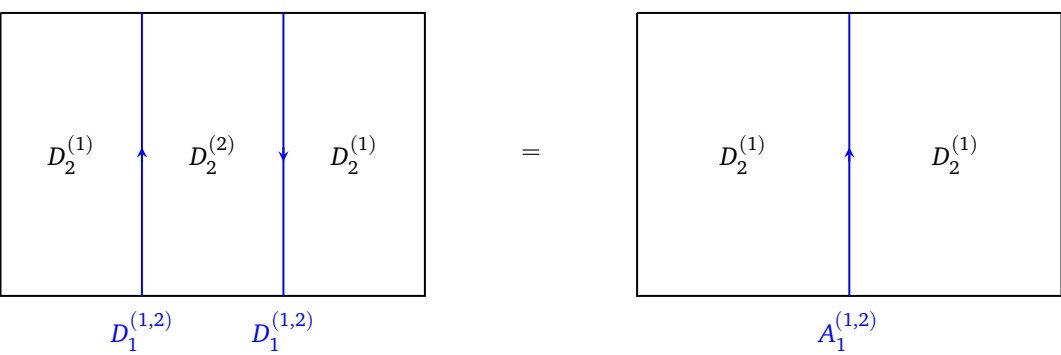

Figure 11: The construction of the object comprising the algebra implementing the gauging procedure to go from $D_2^{(1)}$ to $D_2^{(2)}$.

Figure 12: Construction of various morphisms comprising the algebra $A^{(1,2)}$.

where $B_1^{D_2^{(1)}}$ is an object of $\mathcal{C}_{\mathfrak{T},D_2^{(1)}}$, and the other four are morphisms

$$
\begin{aligned}
B_0^{D_2^{(1)};lp} &: A_1^{(1,2)} \otimes B_1^{D_2^{(1)}} \to B_1^{D_2^{(1)}}, \\
B_0^{D_2^{(1)};rp} &: B_1^{D_2^{(1)}} \otimes A_1^{(1,2)} \to B_1^{D_2^{(1)}}, \\
B_0^{D_2^{(1)};lcp} &: B_1^{D_2^{(1)}} \to A_1^{(1,2)} \otimes B_1^{D_2^{(1)}}, \\
B_0^{D_2^{(1)};rcp} &: B_1^{D_2^{(1)}} \to B_1^{D_2^{(1)}} \otimes A_1^{(1,2)},
\end{aligned}
$$

$$(25)$$

**Figure 13:** Conditions specified by the morphisms comprising the algebra $A^{(1,2)}$. These conditions follow simply from topological moves performed on the topological defects $D_1^{(1,2)}$ participating in the definition of these morphisms.

such that these satisfy the properties shown in figure 16. A morphism in the category $\mathrm{Bimod}_{A^{(1,2)}}\left(\mathcal{C}_{\mathfrak{T},D_2^{(1)}}\right)$ between bimodules $B^{D_2^{(1)},(1)}$ and $B^{D_2^{(1)},(2)}$ is a morphism between objects $B_1^{D_2^{(1)},(1)}$ and $B_1^{D_2^{(1)},(2)}$ in category $\mathcal{C}_{\mathfrak{T},D_2^{(1)}}$ satisfying the relationships shown in figure 17 with the morphisms defining the corresponding bimodules.

The topological line $L_1^{D_2^{(2)}}$ on $D_2^{(2)}$ associated to a bimodule $B^{D_2^{(1)}}$ has the property that

$$B_1^{D_2^{(1)}} = D_1^{(1,2)} \otimes L_1^{D_2^{(2)}} \otimes D_1^{(2,1)}. \tag{26}$$

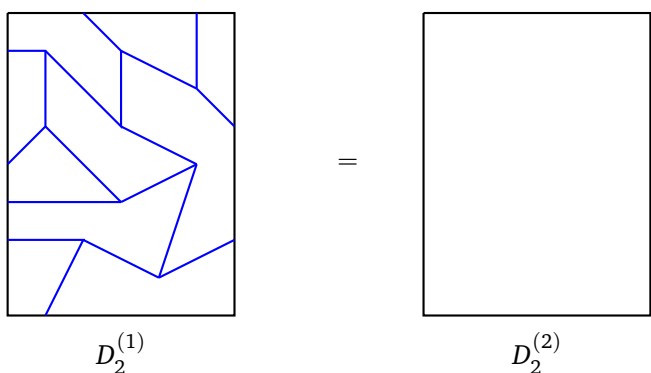

Figure 14: The construction of the topological defect $D_2^{(2)}$ by gauging algebra $A^{(1,2)}$ on $D_2^{(1)}$. The blue lines on the left hand side are algebra objects $A_1^{(1,2)}$, while the tri-junctions are morphisms comprising the algebra.

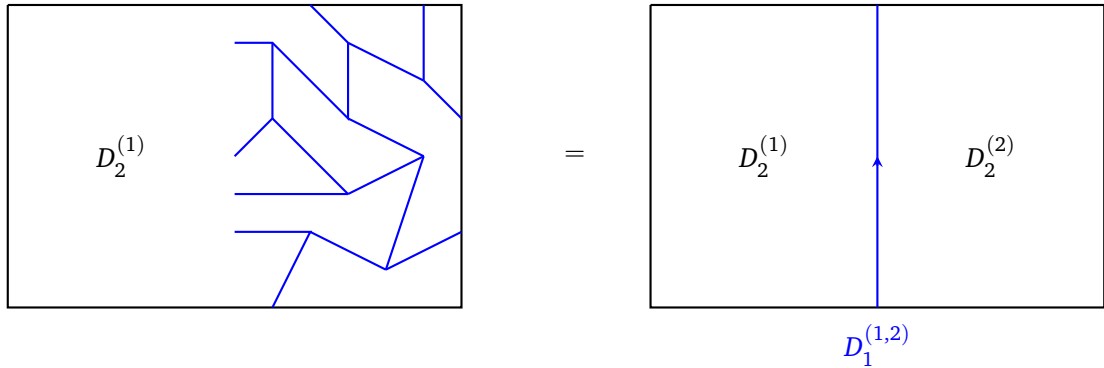

Figure 15: The construction of the interface $D_1^{(1,2)}$ using the algebra $A^{(1,2)}$. The blue lines on the left hand side are algebra objects $A_1^{(1,2)}$, while the tri-junctions and ends of the lines are morphisms comprising the algebra.

See figure 18. The morphisms $B_0^{D_2^{(1)};lp}$, $B_0^{D_2^{(1)};rp}$, $B_0^{D_2^{(1)};lcp}$ and $B_0^{D_2^{(1)};rcp}$ are defined in terms of $L_1^{D_2^{(2)}}$, $D_1^{(1,2)}$ and $D_1^{(2,1)}$ as shown in figure 19.

### 3.3 Generalized Gauging: General $p$

The above description for $p = 2$ is expected to generalize to general $p$. $D_p^{(2)}$ can be obtained from $D_p^{(1)}$ by performing a generalized gauging of the symmetry $\mathcal{C}_{\mathfrak{T},D_p^{(1)}}$. The gauging is expected to be described by what we call a $(p-1)$-algebra $A^{(1,2)}$ in the $(p-1)$-category $\mathcal{C}_{\mathfrak{T},D_p^{(1)}}$. The $(p-1)$-algebra $A^{(1,2)}$ is comprised of an object $A_{p-1}^{(1,2)}$ and multiple $i$-morphisms for $1 \leqslant i \leqslant p-1$ describing various ways in which $A_{p-1}^{(1,2)}$ objects can join, split, be annihilated and created. The object $A_{p-1}^{(1,2)}$ can again be described as

$$A_p^{(1,2)} = D_p^{(1,2)} \otimes D_p^{(2,1)}, \tag{27}$$

while the various $i$-morphisms are described in terms of different configurations of $D_p^{(1,2)}$ and $D_p^{(2,1)}$.

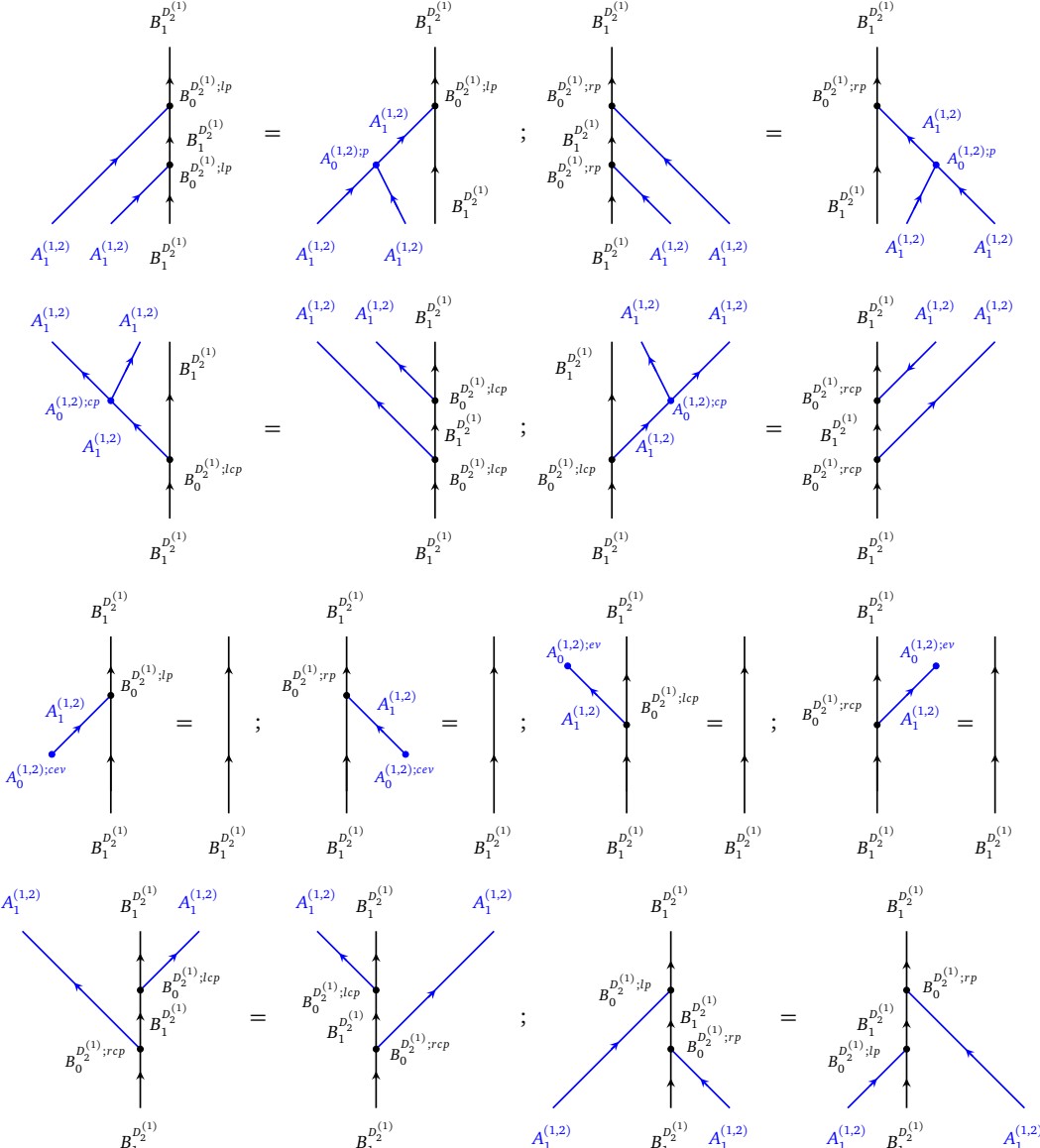

Figure 16: Conditions that a bimodule $B^{D_2^{(1)}}$ has to satisfy.

$D_p^{(2)}$ is then obtained by placing a mesh of the $(p-1)$-algebra $A^{(1,2)}$ along $D_p^{(1)}$. The topological sub-defects of $D_p^{(2)}$ describing the symmetry $(p-1)$-category $\mathcal{C}_{\mathfrak{T}, D_p^{(1)}}$ are obtained in terms of appropriate bimodules of the $(p-1)$-algebra $A^{(1,2)}$.

In this paper, we only consider the case of $p = 2$, and hence do not need to develop the theory of generalized gauging for general $p$ expanding the sketch discussed in this subsection. See [24, 25, 54–56] for prior work in this direction.

## 4  0-Form Gauging of Higher-Categorical Symmetries

In this section, we study a sub-symmetry of $\mathfrak{T}$ given by a $(d-2)$-category $\mathcal{C}_{\mathrm{id},\mathfrak{T}}$ which is a subcategory of the $(d-1)$-category $\mathcal{C}_{\mathfrak{T}}$ capturing the full symmetry of $\mathfrak{T}$. We have a group action on $\mathcal{C}_{\mathrm{id},\mathfrak{T}}$ given by a finite group $G$, which may be non-abelian. The group action corresponds

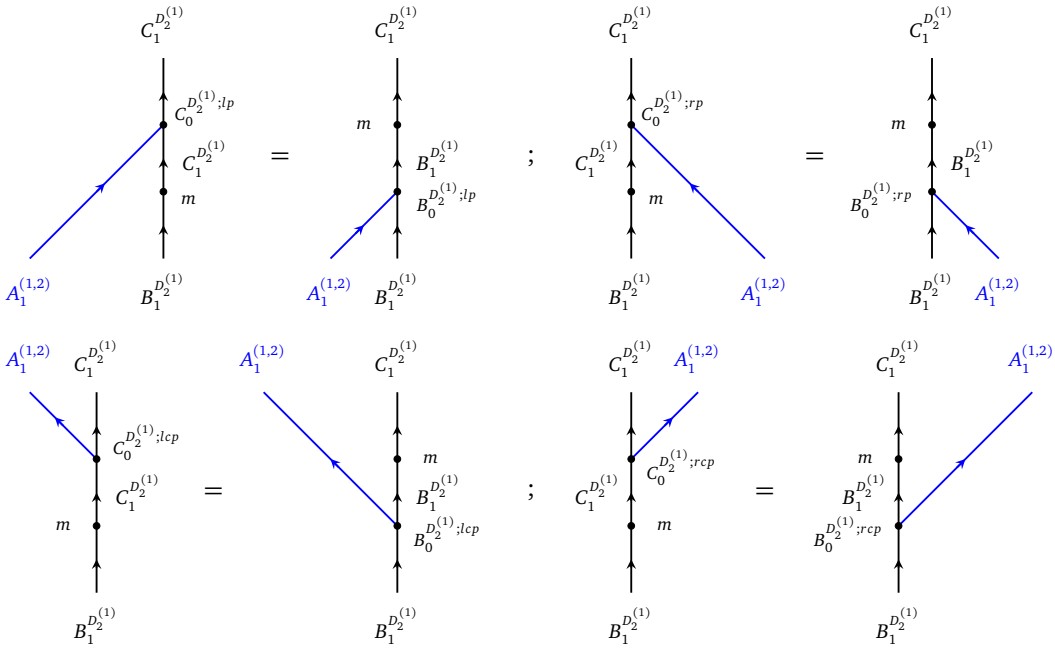

Figure 17: Conditions satisfied by a bimodule morphism $m$ from bimodule $B^{D_2^{(1)}}$ to bimodule $C^{D_2^{(1)}}$.

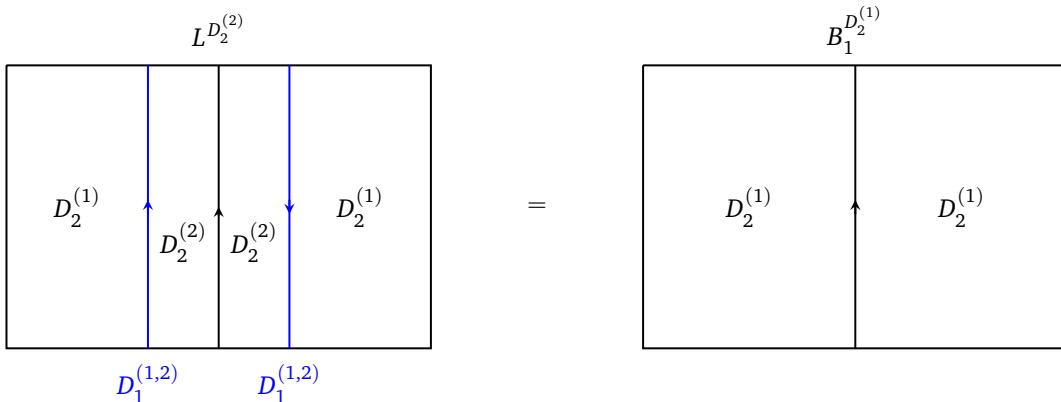

Figure 18: The relationship between a topological line defect $L_1^{D_2^{(2)}}$ living on $D_2^{(2)}$ and the associated bimodule object $B_1^{D_2^{(1)}}$ living on $D_2^{(1)}$.

to a 0-form symmetry of $\mathfrak{T}$ which can be gauged, resulting in the theory $\mathfrak{T}/G$. We describe a construction of the corresponding $(d-2)$-category $\mathcal{C}_{\mathrm{id},\mathfrak{T}/G}$ of the full symmetry $(d-1)$-category $\mathcal{C}_{\mathfrak{T}/G}$ of the gauged theory $\mathfrak{T}/G$ in terms of the data of $\mathcal{C}_{\mathrm{id},\mathfrak{T}}$ and the action of $G$ on it.

The classes of $G$ that we consider are restricted to be of the form[3]

$$G = \Gamma_1 \rtimes \Gamma_2 \rtimes \cdots \rtimes \Gamma_k, \tag{28}$$

where $\Gamma_i$ are abelian groups. This is because we describe the effect of gauging a finite abelian group $\Gamma$, and the effect of gauging $G$ can be deduced by sequentially gauging the finite abelian groups $\Gamma_i$.

---

[3]These do not exhaust all non-abelian finite $G$. An example of a group that cannot be written in this fashion is the group (of order 8) formed by quaternions.

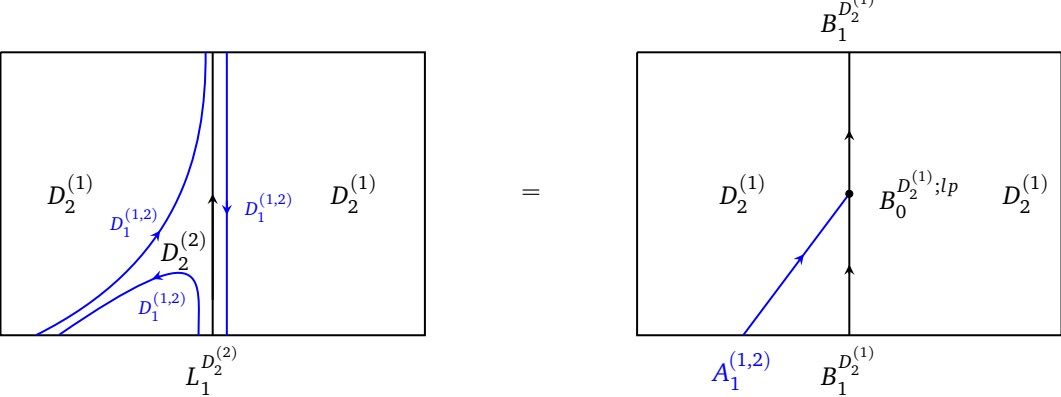

Figure 19: The definition of morphism $B_0^{D_2^{(1)};lp}$ in terms of $L_1^{D_2^{(2)}}$ and $D_1^{(1,2)}$. Other morphisms $B_0^{D_2^{(1)};rp}$, $B_0^{D_2^{(1)};lcp}$ and $B_0^{D_2^{(1)};rcp}$ are defined similarly.

A particularly interesting application would be the construction of higher-categories corresponding to non-invertible symmetries starting from a group action on higher-categories corresponding to invertible symmetries.

## 4.1 Setup

The category $\mathcal{C}_{\mathrm{id},\mathfrak{T}}$ is the category describing symmetries localized along the codimension-1 identity defect of $\mathfrak{T}$. In other words, $\mathcal{C}_{\mathrm{id},\mathfrak{T}}$ is obtained from the full symmetry category $\mathcal{C}_{\mathfrak{T}}$ by forgetting about non-trivial codimension-1 topological defects.

On the other hand, $G$ is a 0-form symmetry of $\mathfrak{T}$. This means that $\mathcal{C}_{\mathfrak{T}}$ contains objects $D_{d-1}^{(g)}$ parametrized by the elements $g$ of $G$. We further assume that there are no 1-morphisms between $D_{d-1}^{(g)}$ and $D_{d-1}^{(g')}$ for $g \neq g'$. This condition is equivalent to requiring that there are no topological defects in the twisted sector of the $G$ 0-form symmetry. If this condition is violated, one obtains extra codimension-two (and also higher codimensional) defects in $\mathcal{C}_{\mathrm{id},\mathfrak{T}/G}$ that are not accounted by our procedure discussed below. These extra defects are also known as topological defects lying in non-trivial flux sector, or as topological Gukov-Witten operators. Our procedure can be applied to such cases, but in such cases we only construct a subcategory of the full category $\mathcal{C}_{\mathrm{id},\mathfrak{T}/G}$, which can be understood as the subcategory formed by defects lying in the trivial flux sector.

The tensor product of these objects follows the group operation on $G$:

$$D_{d-1}^{(g)} \otimes D_{d-1}^{(g')} = D_{d-1}^{(gg')}. \tag{29}$$

We assume that $G$ does not participate in any higher-group structures and 't Hooft anomalies.

The action of $G$ on $\mathcal{C}_{\mathrm{id},\mathfrak{T}}$ is realized as follows:
Consider an object $D_{d-2}$ of $\mathcal{C}_{\mathrm{id},\mathfrak{T}}$. An element $g \in G$ sends $D_{d-2}$ to an element $g \cdot D_{d-2}$ of $\mathcal{C}_{\mathrm{id},\mathfrak{T}}$, which can be computed as

$$g \cdot D_{d-2} = D_{d-2}^{(g)} \otimes D_{d-2} \otimes D_{d-2}^{(g^{-1})}, \tag{30}$$

using the fusion structure on 1-morphisms of the category $\mathcal{C}_{\mathfrak{T}}$, where $D_{d-2}^{(g)}$ is the identity $(d-2)$-dimensional defect on $D_{d-1}^{(g)}$. See figure 20.

Now, consider a 1-morphism $D_{d-3}$ of $\mathcal{C}_{\mathrm{id},\mathfrak{T}}$ from an object $D_{d-2}$ of $\mathcal{C}_{\mathrm{id},\mathfrak{T}}$ to another object $D'_{d-2}$ of $\mathcal{C}_{\mathrm{id},\mathfrak{T}}$. An element $g \in G$ sends $D_{d-3}$ to a 1-morphism $g \cdot D_{d-3}$ of $\mathcal{C}_{\mathrm{id},\mathfrak{T}}$ from the object

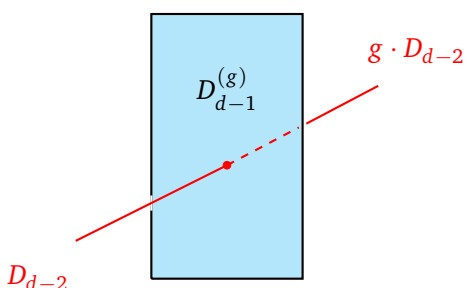

Figure 20: Action of the symmetry $g \in G$ on the defects $D_{d-2}$ as in equation (30).

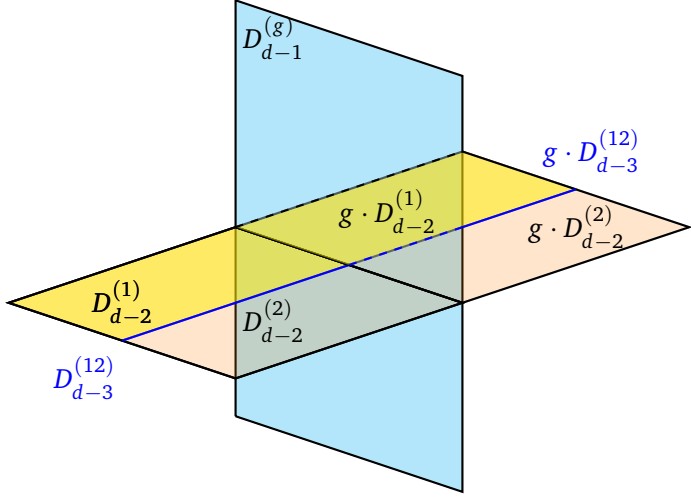

Figure 21: Action of the symmetry $g \in G$ on the defects $D_{d-3}$ as in equation (31).

$g \cdot D_{d-2}$ of $\mathcal{C}_{\mathrm{id},\mathfrak{T}}$ to the object $g \cdot D'_{d-2}$ of $\mathcal{C}_{\mathrm{id},\mathfrak{T}}$, which can be computed as

$$g \cdot D_{d-3} = D_{d-3}^{(g)} \otimes D_{d-3} \otimes D_{d-3}^{(g^{-1})}, \tag{31}$$

using the fusion structure on 2-morphisms of the category $\mathcal{C}_{\mathfrak{T}}$, where $D_{d-3}^{(g)}$ is the identity $(d-3)$-dimensional defect on $D_{d-1}^{(g)}$. See figure 21.

Continuing inductively, consider a $p$-morphism $D_{d-p-2}$ of $\mathcal{C}_{\mathrm{id},\mathfrak{T}}$ from a $(p-1)$-morphism $D_{d-p-1}$ of $\mathcal{C}_{\mathrm{id},\mathfrak{T}}$ to another $(p-1)$-morphism $D'_{d-p-1}$ of $\mathcal{C}_{\mathrm{id},\mathfrak{T}}$. An element $g \in G$ sends $D_{d-p-2}$ to a $p$-morphism $g \cdot D_{d-p-2}$ of $\mathcal{C}_{\mathrm{id},\mathfrak{T}}$ from the $(p-1)$-morphism $g \cdot D_{d-p-1}$ of $\mathcal{C}_{\mathrm{id},\mathfrak{T}}$ to the $(p-1)$-morphism $g \cdot D'_{d-p-1}$ of $\mathcal{C}_{\mathrm{id},\mathfrak{T}}$, which can be computed as

$$g \cdot D_{d-p-2} = D_{d-p-2}^{(g)} \otimes D_{d-p-2} \otimes D_{d-p-2}^{(g^{-1})}, \tag{32}$$

using the fusion structure on $(p+1)$-morphisms of the category $\mathcal{C}_{\mathfrak{T}}$, where $D_{d-p-2}^{(g)}$ is the identity $(d-p-2)$-dimensional defect on $D_{d-1}^{(g)}$. We will restrict $G$ to be an abelian group from this point on.

## 4.2 Gauging in 3d

We begin the discussion of gauging finite, abelian $G$ from the special case of $d = 3$. This has been considered in the literature earlier [30, 31] in the context of 3d TQFTs, and our discussion is mostly a review of these works but now applied to general 3d QFTs that need not

be topological. We formulate the discussion such that it is amenable to generalization to higher dimensions. The category $\mathcal{C}_{\mathrm{id},\mathfrak{T}}$ is a standard 1-category describing the genuine topological line defects and the topological local operators living at their junctions. Our task is to determine the 1-category $\mathcal{C}_{\mathrm{id},\mathfrak{T}/G}$ of the 3d theory $\mathfrak{T}/G$ obtained after gauging the 0-form symmetry $G$ of the 3d theory $\mathfrak{T}$.

Let us begin by discussing the objects of $\mathcal{C}_{\mathrm{id},\mathfrak{T}/G}$, i.e. the genuine line defects in the theory $\mathfrak{T}/G$. First of all, gauging $G$ produces Wilson line defects for the gauge group $G$, which are topological as $G$ is finite and abelian. We will discuss a non-abelian example in section 5.3. These line defects form a sub-category

$$\mathrm{Rep}(G) = \mathrm{Vec}_{\widehat{G}} \tag{33}$$

of $\mathcal{C}_{\mathrm{id},\mathfrak{T}/G}$, where $\mathrm{Vec}_{\widehat{G}}$ is the category of vector spaces graded by elements of the Pontryagin dual $\widehat{G}$ of $G$. Recall that $\widehat{G}$ is the group formed by irreducible representations of the finite group $G$ (which are all 1-dimensional) under tensor product operation. This subcategory provides objects in $\mathcal{C}_{\mathrm{id},\mathfrak{T}/G}$ labeled by representations of $G$. The irreducible representations of $G$, i.e. elements of $\widehat{G}$, give rise to simple objects of $\mathcal{C}_{\mathrm{id},\mathfrak{T}/G}$.

In addition to the $G$ representations, there are objects of $\mathcal{C}_{\mathrm{id},\mathfrak{T}/G}$ arising from the objects of $\mathcal{C}_{\mathrm{id},\mathfrak{T}}$. However, not every object of $\mathcal{C}_{\mathrm{id},\mathfrak{T}}$ descends to an object of $\mathcal{C}_{\mathrm{id},\mathfrak{T}/G}$. This is because only those objects of $\mathcal{C}_{\mathrm{id},\mathfrak{T}}$ that are left invariant by the action of $G$ are gauge invariant in the theory $\mathfrak{T}/G$, so only those objects survive as objects of $\mathcal{C}_{\mathrm{id},\mathfrak{T}/G}$. A simple object $D_1^{(O)}$ of $\mathcal{C}_{\mathrm{id},\mathfrak{T}/G}$ arising this way can be described as the object

$$D_1^{(O)} \equiv \bigoplus_{i \in O} D_1^{(i)} \tag{34}$$

in the category $\mathcal{C}_{\mathrm{id},\mathfrak{T}}$, where $D_1^{(i)}$ are distinct simple objects of $\mathcal{C}_{\mathrm{id},\mathfrak{T}}$ lying in an orbit $O$ of the $G$ action.

Finally, there are simple objects of $\mathcal{C}_{\mathrm{id},\mathfrak{T}/G}$ which are mixtures of the two above kinds of simple objects, which can be thought of as objects $D_1^{(O)}$ dressed by Wilson line defects. Concretely, to a simple object $D_1^{(O)}$, we can associate a subgroup $G_O$ of $G$, which is the stabilizer of any object $D_1^{(i)}$ for $i \in O$. Such an object $D_1^{(O)}$ can be dressed by representations of the stabilizer $G_O$. Thus the simple objects corresponding to $D_1^{(O)}$ can be represented as

$$D_1^{(O,R_O)}, \tag{35}$$

where $R_O$ is an irreducible representation of $G_O$, or in other words, an element of the Pontryagin dual group $\widehat{G}_O$ of $G_O$. The bare object $D_1^{(O)}$ is obtained by choosing $R_O$ to be the trivial representation. The simple objects of the subcategory $\mathrm{Rep}(G)$ of $\mathcal{C}_{\mathrm{id},\mathfrak{T}/G}$ are obtained as special cases of $D_1^{(O,R_O)}$ by taking $O$ to be the orbit formed by the identity object of $\mathcal{C}_{\mathrm{id},\mathfrak{T}}$, for which the stabilizer is the whole group $G$. We represent these simple objects as

$$D_1^{(R)}, \tag{36}$$

where $R$ is an irreducible representation of $G$. The identity object of the category $\mathcal{C}_{\mathrm{id},\mathfrak{T}/G}$ is denoted as

$$D_1^{(\mathrm{id})}, \tag{37}$$

which is obtained by choosing $R$ to be the trivial representation of $G$.

Let us now discuss the fusion operation on the objects of $\mathcal{C}_{\mathrm{id},\mathfrak{T}/G}$. First of all, we have

$$D_1^{(R)} \otimes D_1^{(R')} = D_1^{(RR')}, \tag{38}$$

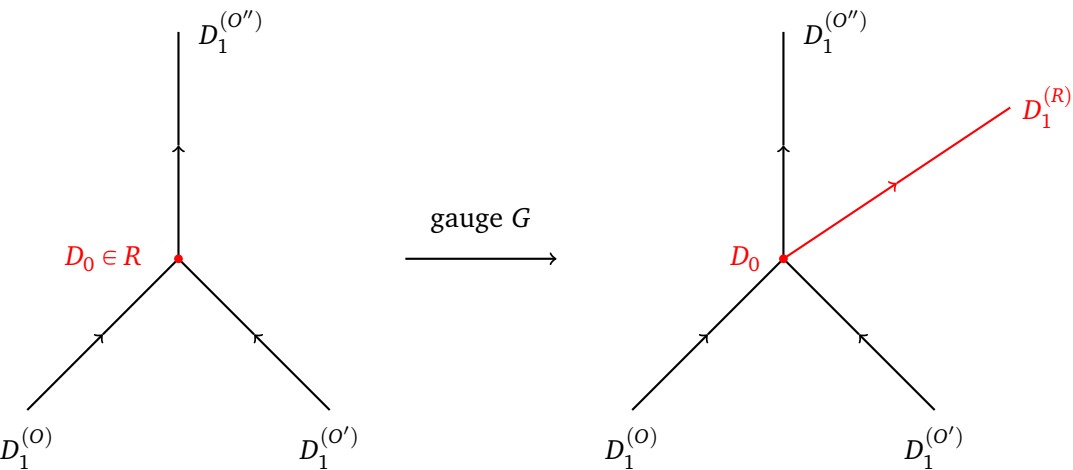

Figure 22: A local operator transforming in representation $R$ of $G$ before gauging $G$, is attached to a Wilson line in representation $R$ of $G$ after gauging $G$.

where $R, R' \in \widehat{G}$, and $RR' \in \widehat{G}$ is the product of $R$ and $R'$ in $\widehat{G}$. Next, we have

$$D_1^{(O,R_O)} \otimes D_1^{(R')} = D_1^{(O,R_O R'_O)}, \tag{39}$$

where $R'_O \in \widehat{G}_O$ is the image of $R \in \widehat{G}$ under the surjective homomorphism

$$\pi_O : \widehat{G} \to \widehat{G}_O, \tag{40}$$

dual to the injective homomorphism

$$i_O : G_O \to G, \tag{41}$$

descending from the fact that $G_O$ is a subgroup of $G$.

The fusions $D_1^{(O,R_O)} \otimes D_1^{(O',R'_{O'})}$ are more complicated. For this purpose, we consider the fate of local operators, i.e. morphisms of $\mathcal{C}_{\mathrm{id},\mathfrak{T}}$ under gauging. Because we have an action of $G$ on $\mathcal{C}_{\mathrm{id},\mathfrak{T}}$, the morphisms (between objects left invariant by $G$ action) can be decomposed into representations of $G$. After gauging $G$, a morphism transforming in representation $R$ of $G$ is not gauge invariant, but can be made gauge invariant by attaching a Wilson line in representation $R$ of $G$. See figure 22. This phenomenon provides information about the morphisms of $\mathcal{C}_{\mathrm{id},\mathfrak{T}/G}$, and hence in particular the tensor product structure on objects of $\mathcal{C}_{\mathrm{id},\mathfrak{T}/G}$.

Let us begin by verifying, from this point of view, the fusion relation (39) for the case when $R_O$ is trivial, in which case the fusion relation becomes

$$D_1^{(O)} \otimes D_1^{(R)} = D_1^{(O,R_O)}. \tag{42}$$

The verification amounts to showing that there is a 1-dimensional space of morphisms from $D_1^{(O)}$ to $D_1^{(O)}$ (which can also be referred to as endomorphisms of the object $D_1^{(O)}$) in the category $\mathcal{C}_{\mathrm{id},\mathfrak{T}}$ that form representation $R^{-1} \in \widehat{G}$ under the $G$ action. The endomorphism space of

$$D_1^{(O)} = \bigoplus_{i \in O} D_1^{(i)} \tag{43}$$

in $\mathcal{C}_{\mathrm{id},\mathfrak{T}}$ has dimension $|O|$, where $|O|$ denotes the number of elements $i$ in the orbit $O$. This is because $D_1^{(i)}$ are simple objects, so morphism space from $D_1^{(i)}$ to $D_1^{(j)}$ is $\delta_{ij}$ dimensional. There action of the stabilizer group $G_O$ on the 1-dimensional endomorphism space of each $D_1^{(i)}$ has

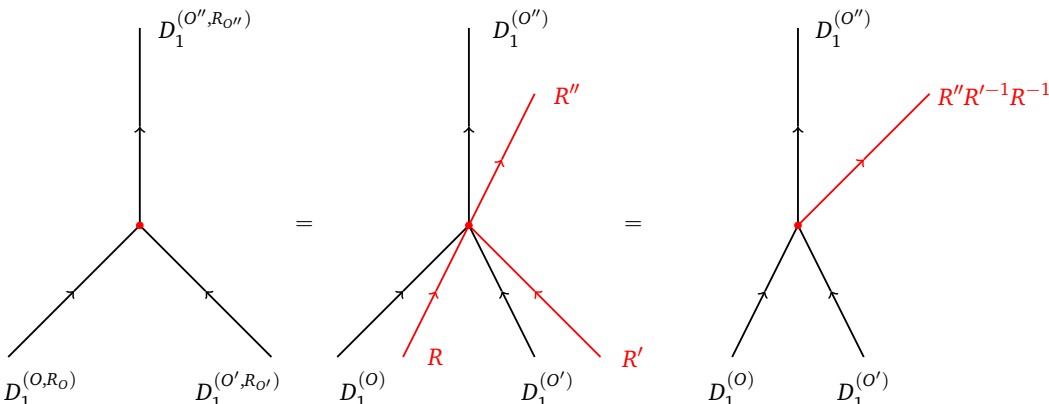

Figure 23: We can relate the morphism space $D_1^{(O,R_O)} \otimes D_1^{(O',R_{O'})} \to D_1^{(O'',R_{O''})}$ after gauging $G$ to the sub-space of the morphism space $D_1^{(O)} \otimes D_1^{(O')} \to D_1^{(O'')}$ before gauging $G$ transforming in a specific representation of $G$ determined by the representations $R_O, R_{O'}, R_{O''}$.

to be trivial for consistency. Using this fact, it can be easily shown that the endomorphism space of $D_1^{(O)}$ decomposes as

$$\bigoplus_{R \in \ker(\pi_O)} R, \tag{44}$$

where $\ker(\pi_O) \subseteq \widehat{G}$ is the kernel of the map $\pi_O$ defined in (40). Thus, we have

$$D_1^{(O)} \otimes D_1^{(R)} = D_1^{(O)}, \tag{45}$$

only if $R \in \ker(\pi_O)$, which agrees with (39).

Now let us say we want to deduce morphisms of $\mathcal{C}_{\mathrm{id}, \mathfrak{T}/G}$ from object $D_1^{(O)} \otimes D_1^{(O')}$ to $D_1^{(O'')}$. These can be recognized as the subspace of morphisms between same objects in the category $\mathcal{C}_{\mathrm{id}, \mathfrak{T}}$ which is left invariant by the action of $G$. Let $V_{(O),(O')}^{(O'')}$ be this invariant subspace. Then, $D_1^{(O)} \otimes D_1^{(O')}$ contains $\dim\left(V_{(O),(O')}^{(O'')}\right)$ copies of $D_1^{(O'')}$ in $\mathcal{C}_{\mathrm{id}, \mathfrak{T}/G}$, where $\dim\left(V_{(O),(O')}^{(O'')}\right)$ is the dimension of the vector space $V_{(O),(O')}^{(O'')}$.

Generalizing this, the morphisms of $\mathcal{C}_{\mathrm{id}, \mathfrak{T}/G}$ from object $D_1^{(O,R_O)} \otimes D_1^{(O',R_{O'})}$ to $D_1^{(O'',R_{O''})}$ are deduced from the subspace of morphisms from $D_1^{(O)} \otimes D_1^{(O')}$ to $D_1^{(O'')}$ of the category $\mathcal{C}_{\mathrm{id}, \mathfrak{T}}$ that transform in representation

$$R''' := R'' R'^{-1} R^{-1} \in \widehat{G}, \tag{46}$$

for any choices of elements $R \in \pi_O^{-1}(R_O) \subseteq \widehat{G}$, $R' \in \pi_{O'}^{-1}(R_{O'}) \subseteq \widehat{G}$ and $R'' \in \pi_{O''}^{-1}(R_{O''}) \subseteq \widehat{G}$. See figure 23 for an explanation. Let $V_{(O,R_O),(O',R_{O'})}^{(O'',R_{O''})}$ be this space of morphisms in $\mathcal{C}_{\mathrm{id}, \mathfrak{T}/G}$ from $D_1^{(O,R_O)} \otimes D_1^{(O',R_{O'})}$ to $D_1^{(O'',R_{O''})}$. Then, $D_1^{(O,R_O)} \otimes D_1^{(O',R_{O'})}$ contains

$$\dim\left(V_{(O,R_O),(O',R_{O'})}^{(O'',R_{O''})}\right) \tag{47}$$

copies of $D_1^{(O'',R_{O''})}$.

## 4.3 Gauging in Higher $d$ and Fusion

We now extend the discussion of the previous subsection to arbitrary $d \geqslant 4$. The category $\mathcal{C}_{\mathrm{id}, \mathfrak{T}}$ before gauging is now a $(d-2)$-category. Our task is to determine the $(d-2)$-category $\mathcal{C}_{\mathrm{id}, \mathfrak{T}/G}$

obtained after gauging the 0-form symmetry $G$ of the theory $\mathfrak{T}$.

The objects of $\mathcal{C}_{\mathrm{id},\mathfrak{T}/G}$ are objects of $\mathcal{C}_{\mathrm{id},\mathfrak{T}}$ left invariant by the $G$ action, and objects related to such gauge invariant objects by condensation. If $d \geqslant 5$, then 1-morphisms of $\mathcal{C}_{\mathrm{id},\mathfrak{T}/G}$ from object $D_{d-2}$ to object $D'_{d-2}$ of $\mathcal{C}_{\mathrm{id},\mathfrak{T}/G}$ are obtained as the 1-morphisms from object $D_{d-2}$ to object $D'_{d-2}$ of $\mathcal{C}_{\mathrm{id},\mathfrak{T}}$ that are left invariant by $G$-action, and 1-morphisms related to such gauge invariant 1-morphisms by condensation. Continuing inductively, if $d \geqslant 4 + p$, then $p$-morphisms of $\mathcal{C}_{\mathrm{id},\mathfrak{T}/G}$ from $(p-1)$-morphism $D_{d-p-1}$ to $(p-1)$-morphism $D'_{d-p-1}$ of $\mathcal{C}_{\mathrm{id},\mathfrak{T}/G}$ are obtained as the $p$-morphisms from $(p-1)$-morphism $D_{d-p-1}$ to $(p-1)$-morphism $D'_{d-p-1}$ of $\mathcal{C}_{\mathrm{id},\mathfrak{T}}$ that are left invariant by $G$-action, and $p$-morphisms related to such gauge invariant $p$-morphisms by condensation.

Fusion of two non-condensation defects can create condensation defects. That is, if we consider two $p$-morphisms of $\mathcal{C}_{\mathrm{id},\mathfrak{T}/G}$ obtained directly as gauge invariant $p$-morphisms of $\mathcal{C}_{\mathrm{id},\mathfrak{T}}$ without involving any condensation, then the product $p$-morphism is in general a $p$-morphism of $\mathcal{C}_{\mathrm{id},\mathfrak{T}/G}$ obtained as a gauge invariant $p$-morphism of $\mathcal{C}_{\mathrm{id},\mathfrak{T}}$ with an additional condensation on top of it. We will describe how the additional condensation can be determined for surface defects, while leaving the case of higher-dimensional defects to future works.

We still need to describe $(d-3)$-morphisms and $(d-2)$-morphisms of $\mathcal{C}_{\mathrm{id},\mathfrak{T}/G}$. These correspond respectively to (genuine and non-genuine) topological line defects and topological local operators of the gauged theory $\mathfrak{T}/G$. As in previous subsection, to describe them we need to also incorporate Wilson line defects created by the $G$ gauging. The analysis is a straightforward generalization of the analysis in the previous subsection.

First of all, the $(d-3)$-morphisms of $\mathcal{C}_{\mathrm{id},\mathfrak{T}}$ that are left invariant by $G$ action descend to $(d-3)$-morphisms of $\mathcal{C}_{\mathrm{id},\mathfrak{T}/G}$. Thus, a class of simple $(d-3)$-morphisms of $\mathcal{C}_{\mathrm{id},\mathfrak{T}/G}$ are $D_1^{(O)}$ which can be represented in the category $\mathcal{C}_{\mathrm{id},\mathfrak{T}}$ as

$$D_1^{(O)} = \bigoplus_{i \in O} D_1^{(i)}, \tag{48}$$

where $O$ is an orbit under $G$ action formed by simple $(d-3)$-morphisms $D_1^{(i)}$ of the category $\mathcal{C}_{\mathrm{id},\mathfrak{T}}$. Other simple $(d-3)$-morphisms of $\mathcal{C}_{\mathrm{id},\mathfrak{T}/G}$ are obtained by dressing $D_1^{(O)}$ by Wilson line defects valued in $\widehat{G}_O$, which is the Pontryagin dual of the stabilizer $G_O \subseteq G$ of the orbit $O$. These $(d-3)$-morphisms are represented as

$$D_1^{(O,R_O)}, \tag{49}$$

with $R_O \in \widehat{G}_O$.

The $(d-2)$-morphisms of $\mathcal{C}_{\mathrm{id},\mathfrak{T}/G}$ from $D_1^{(O,R_O)} \otimes D_1^{(O',R_{O'})}$ to $D_1^{(O'',R_{O''})}$ are obtained in terms of $(d-2)$-morphisms of $\mathcal{C}_{\mathrm{id},\mathfrak{T}}$ from $D_1^{(O)} \otimes D_1^{(O')}$ to $D_1^{(O'')}$ as described in the previous subsection. Similarly, the $(d-2)$-morphisms of $\mathcal{C}_{\mathrm{id},\mathfrak{T}/G}$ from $D_1^{(O,R_O)} \otimes_{D_q} D_1^{(O',R_{O'})}$ to $D_1^{(O'',R_{O''})}$ where $D_q$ with $q \geqslant 2$ is a higher-morphism of $\mathcal{C}_{\mathrm{id},\mathfrak{T}/G}$ are also obtained in terms of $(d-2)$-morphisms of $\mathcal{C}_{\mathrm{id},\mathfrak{T}}$ from $D_1^{(O)} \otimes_{D_q} D_1^{(O')}$ to $D_1^{(O'')}$, with the only difference in the procedure being that $\otimes$ is replaced by $\otimes_{D_q}$.

**Fusion of Surface Defects and Condensation.** We now provide the key to computing the fusion of topological surface defects in the symmetry category. For this we now describe how fusion of surfaces can create condensations – and provide numerous examples in the subsequent sections.

Consider two surfaces $D_2^{(O)}$ and $D_2^{(O')}$ described by orbits $O$ and $O'$. In the theory $\mathfrak{T}$ before gauging, they have a fusion rule

$$D_2^{(O)} \otimes D_2^{(O')} = \oplus_i D_2^{(i)}, \tag{50}$$



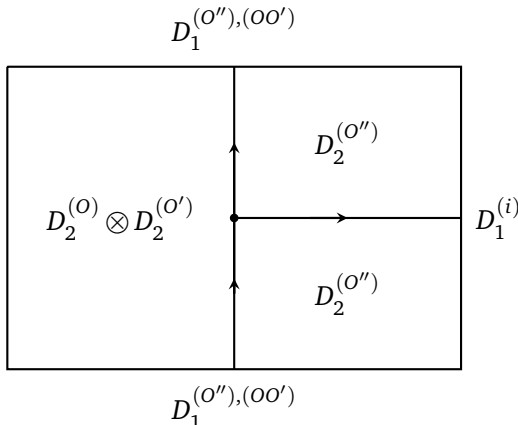

Figure 24: The coefficient $n_i$ appearing in (52) is the dimension of the vector space of local operators shown in the figure.

where $D_2^{(i)}$ are simple surfaces. We have line operators describing the 1-morphisms $D_2^{(O)} \otimes D_2^{(O')} \to D_2^{(i)}$ in the category $\mathcal{C}_{\mathrm{id}, \mathfrak{T}}$. These line operators organize themselves into orbits under the $G$ action such that we can write the above equation as

$$D_2^{(O)} \otimes D_2^{(O')} = \oplus_{O''} D_2^{(O'')}, \tag{51}$$

where $O''$ are orbits of surfaces. The right hand side $\oplus_{O''} D_2^{(O'')}$ of the above fusion is a representative of the equivalence class under condensation of $D_2^{(O)} \otimes D_2^{(O')}$ in the theory $\mathfrak{T}/G$.

Our task is now to describe the generalized gauging on top of each $D_2^{(O'')}$. This is captured in terms of an algebra $A^{(O''),(OO')}$ in the 1-category of localized symmetries $\mathcal{C}_{\mathrm{id}, \mathfrak{T}/G, D_2^{O''}}$. Let us first describe the object $A_1^{(O''),(OO')}$ comprising the algebra. From the gauging procedure, we obtain a line operator $D_1^{(O''),(OO')}$ describing a 1-morphism $D_2^{(O)} \otimes D_2^{(O')} \to D_2^{(O'')}$ in the category $\mathcal{C}_{\mathrm{id}, \mathfrak{T}/G}$. The algebra object can be expressed as

$$A_1^{(O''),(OO')} = \oplus_i n_i D_1^{(i)}, \tag{52}$$

where $D_1^{(i)}$ are the line operators living on $D_2^{(O'')}$ and $n_i$ is the dimension of the vector space formed by local operators living at the end of $D_1^{(i)}$ along $D_1^{(O''),(OO')}$ as shown in figure 24. This follows from (19) applied to the current situation in e.g. figure 24.

Thus, the algebra object $A_1^{(O''),(OO')}$ is described by local operators living on $D_1^{(O''),(OO')}$.

The morphisms comprising the algebra can be roughly determined as follows. We leave a full description to a future work. In many of the examples encountered in this paper, the algebra object would uniquely fix the algebra morphisms, so we will not require this machinery. The algebra product $A_0^{(O''),(OO');p}$ comes from fusing these local operators along the line $D_1^{(O''),(OO')}$. The algebra co-product $A_0^{(O''),(OO');cp}$ is the adjoint of the algebra product. The algebra evaluation and co-evaluation morphisms $A_0^{(O''),(OO');ev}$ and $A_0^{(O''),(OO');cev}$ are also easy to determine as follows. There is a one-dimensional space of local operators living along $D_1^{(O''),(OO')}$ that are not attached to any other line operator. The morphism $A_0^{(O''),(OO');ev}$ is the projection map from all local operators to this one-dimensional space, and the $A_0^{(O''),(OO');cev}$ is the inclusion map from this one-dimensional space to the space of all local operators.

In summary the fusion of surfaces is computed as follows:

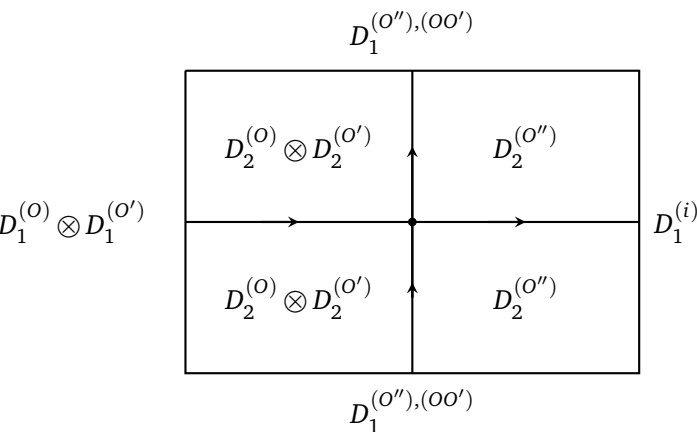

Figure 25: The coefficient $n_i$ appearing in (54) is the dimension of the vector space of local operators shown in the figure.

1. Determine the orbit decomposition of the surface fusion as in (51).

2. Compute the algebra object $A_1^{(O''),(OO')}$ which characterizes the gauging of the localized symmetry on $D_2^{O''}$. This is computed as in (52) in terms of certain 2-morphisms capturing various kinds of local operators living on the line defect describing a 1-morphism $D_2^{(O)} \otimes D_2^{(O')} \to D_2^{(O'')}$.

3. The morphisms comprising the algebra are determined as described in the previous paragraph.

4. Then the fusion of $D_2^{(O)} \otimes D_2^{(O')}$ will contain a term

$$\frac{D_2^{(O'')}}{A^{(O''),(OO')}}, \tag{53}$$

which describes the condensation appearing on top of $D_2^{(O'')}$ in terms of the algebra $A^{(O''),(OO')}$ using which one performs generalized gauging associated to the condensation. The fusion of condensation defects, which are a central part of the symmetry category, will not be discussed in this paper, as it requires developing further technology. The fusion of condensation defects have appeared in [47–50].

In a similar way, we can describe the fusion of two arbitrary lines $D_1^{(O)}$ and $D_1^{(O')}$ living respectively in $D_2^{(O)}$ and $D_2^{(O')}$. This is described as a collection of bimodules $B^{(O'')}$ in the symmetry categories associated to $D_2^{(O'')}$ appearing in (51). The bimodule object $B_1^{(O'')}$ is described as

$$B_1^{(O'')} = \oplus_i n_i D_1^{(i)}, \tag{54}$$

where $D_1^{(i)}$ are the line operators living on $D_2^{(O'')}$ and $n_i$ is the dimension of the vector space formed by local operators living at the end of $D_1^{(i)}$, $D_1^{(O)}$ and $D_1^{(O')}$ along $D_1^{(O''),(OO')}$ as shown in figure 25.

The morphisms $B_0^{(O'');lp}$, $B_0^{(O'');rp}$, $B_0^{(O'');lcp}$ and $B_0^{(O'');rcp}$ comprising the bimodule are obtained as follows. The morphisms $B_0^{(O'');lp}$ and $B_0^{(O'');rp}$ are obtained by fusing the above local

operators describing the bimodule object with the local operators describing the algebra object, and the morphisms $B_0^{(O'');lcp}$ and $B_0^{(O'');rcp}$ are obtained as adjoints of the above product operations. We leave a precise analysis to future work.

It should be possible to generalize the above procedure to describe condensations included in the fusion of higher-dimensional topological defects, and would be an interesting future direction to develop.

# 5 Examples: Non-Invertible Categorical Symmetries in 3d

In this section we provide examples of 3d theories whose topological line defects and local operators form a non-invertible symmetry described by a standard 1-category. All the examples we discuss can be obtained by gauging an invertible 0-form symmetry of 3d theories containing invertible 1-form symmetries upon which the 0-form symmetry acts non-trivially. The main example is the gauging of $\mathbb{Z}_2$ on pure Spin($2N$) gauge theories, which is carried out in section 5.1 and appendix A.1. We also provide an example of gauging of a non-abelian finite symmetry $S_3$ in section 5.3.

In subsection 5.2, we connect with established literature [30, 31] pertaining to gauging 0-form global symmetries in 3d TQFTs by reviewing the paradigmatic example of gauging $\mathbb{Z}_2$ electromagnetic duality symmetry in the topological $\mathbb{Z}_2$ gauge theory [57]. In this example we obtain the sub-category of the symmetry category of the gauged theory corresponding to the zero flux sector[4], which agrees with the known result [31, 58].

In most of the examples presented, the symmetry categories describe discrete symmetries, but we also provide an example in subsection 5.4 of a symmetry category describing continuous symmetries, or in other words a *continuous symmetry category*.

## 5.1 Pure Pin$^+(4N)$ Gauge Theory in 3d

In this subsection, we begin with pure Spin($4N$) Yang-Mills theory in 3d, which has a 1-form symmetry group $\mathbb{Z}_2 \times \mathbb{Z}_2$. The outer-automorphism of the gauge algebra $\mathfrak{so}(4N)$ provides a $\mathbb{Z}_2$ 0-form symmetry of the theory that exchanges the two $\mathbb{Z}_2$ factors of the 1-form symmetry group. Gauging this 0-form symmetry results in the pure Pin$^+(4N)$ Yang-Mills theory in 3d, which we show to contain a non-invertible categorical symmetry descending from the 1-form symmetry of Spin($4N$) Yang-Mills theory.

We label the two $\mathbb{Z}_2$ factors in the 1-form symmetry group of Spin($4N$) theory as $\mathbb{Z}_2^{(S)}$ and $\mathbb{Z}_2^{(C)}$ depending on whether the $\mathbb{Z}_2$ leaves the spinor irrep $S$ invariant, or the cospinor irrep $C$ invariant. The diagonal $\mathbb{Z}_2$ factor can be represented as $\mathbb{Z}_2^{(V)}$ as it leaves the vector irrep invariant. Thus, we express the 1-form symmetry group $\Gamma^{(1)}$ as

$$\Gamma^{(1)} = \mathbb{Z}_2^{(S)} \times \mathbb{Z}_2^{(C)}. \tag{55}$$

The outer-automorphism provides a 0-form symmetry group

$$\Gamma^{(0)} = \mathbb{Z}_2^{(0)}, \tag{56}$$

which exchanges $\mathbb{Z}_2^{(S)}$ and $\mathbb{Z}_2^{(C)}$, while leaving the $\mathbb{Z}_2^{(V)}$ invariant.

The data of 1-form symmetry can be converted into the data of a 1-category $\mathcal{C}_{\text{Spin}(4N)}$ as follows. The simple objects of $\mathcal{C}_{\text{Spin}(4N)}$ correspond to topological line operators implementing

---

[4]See the discussion regarding restriction of our analysis to zero-flux sector at the beginning of section 4.1.

the 1-form symmetry $\Gamma^{(1)}$, and we write the set of simple objects as

$$\mathcal{C}^{\text{ob}}_{\text{Spin}(4N)} = \left\{ D_1^{(\text{id})}, D_1^{(S)}, D_1^{(C)}, D_1^{(V)} \right\}, \tag{57}$$

where $D_1^{(\text{id})}$ is the identity line, and $D_1^{(i)}$ are the topological line operators corresponding respectively to generators of $\mathbb{Z}_2^{(i)}$ 1-form symmetries. The $\mathbb{Z}_2$ 0-form symmetry acts on the 1-form symmetry generators as

$$D_1^{(S)} \longleftrightarrow D_1^{(C)}, \tag{58}$$

and leaves $D_1^{(\text{id})}, D_1^{(V)}$ invariant. The tensor product of these objects follows the group law of $\Gamma^{(1)}$.

Now we gauge $\mathbb{Z}_2$ to obtain a category $\mathcal{C}_{\text{Pin}^+(4N)}$ describing topological line defects and local operators of the $\text{Pin}^+(4N)$ theory. A subset of simple objects of $\mathcal{C}_{\text{Pin}^+(4N)}$ arise as objects of $\mathcal{C}_{\text{Spin}(4N)}$ left invariant by the $\mathbb{Z}_2$ outer automorphism action. These are $D_1^{(\text{id})}, D_1^{(V)}$ and

$$D_1^{(SC)} := \left( D_1^{(S)} \oplus D_1^{(C)} \right)_{\mathcal{C}_{\text{Spin}}}, \tag{59}$$

where the subscript $\mathcal{C}_{\text{Spin}(4N)}$ on the RHS reflects that the object $D_1^{(SC)}$ is decomposed as this direct sum only in the category $\mathcal{C}_{\text{Spin}(4N)}$, but it is a simple object in the category $\mathcal{C}_{\text{Pin}^+(4N)}$.

Other simple objects of $\mathcal{C}_{\text{Pin}^+(4N)}$ are obtained by dressing with Wilson line defects. Note that the stabilizer for $D_1^{(\text{id})}, D_1^{(V)}$ is the whole 0-form symmetry group $\mathbb{Z}_2$, while the stabilizer for $D_1^{(SC)}$ is trivial. Thus, we obtain new simple objects of $\mathcal{C}_{\text{Pin}^+(4N)}$ by dressing $D_1^{(\text{id})}, D_1^{(V)}$ with the non-trivial irrep of $\mathbb{Z}_2$. We call the resulting simple objects as $D_1^{(-)}, D_1^{(V_-)}$ respectively. Thus, the full set of simple objects of $\mathcal{C}_{\text{Pin}^+(4N)}$ is

$$\mathcal{C}^{\text{ob}}_{\text{Pin}^+(4N)} = \left\{ D_1^{(\text{id})}, D_1^{(-)}, D_1^{(SC)}, D_1^{(V)}, D_1^{(V_-)} \right\}. \tag{60}$$

The topological line defects

$$\left\{ D_1^{(\text{id})}, D_1^{(-)} \right\} \tag{61}$$

are the Wilson line defects for the gauged $\mathbb{Z}_2$ 0-form symmetry, and hence generate the dual $\mathbb{Z}_2$ 1-form symmetry arising as a result of $\mathbb{Z}_2$ 0-form gauging. Their fusion rules are (38)

$$\begin{aligned}
D_1^{(\text{id})} \otimes D_1^{(\text{id})} &= D_1^{(\text{id})}, \\
D_1^{(\text{id})} \otimes D_1^{(-)} &= D_1^{(-)}, \\
D_1^{(-)} \otimes D_1^{(-)} &= D_1^{(\text{id})},
\end{aligned} \tag{62}$$

following the representation theory of $\mathbb{Z}_2$.

Moreover, from (39) we have

$$\begin{aligned}
D_1^{(SC)} \otimes D_1^{(\text{id})} &= D_1^{(SC)}, \\
D_1^{(SC)} \otimes D_1^{(-)} &= D_1^{(SC)}, \\
D_1^{(V)} \otimes D_1^{(\text{id})} &= D_1^{(V)}, \\
D_1^{(V)} \otimes D_1^{(-)} &= D_1^{(V_-)}, \\
D_1^{(V_-)} \otimes D_1^{(\text{id})} &= D_1^{(V_-)}, \\
D_1^{(V_-)} \otimes D_1^{(-)} &= D_1^{(V)}
\end{aligned} \tag{63}$$

in the category $\mathcal{C}_{\text{Pin}^+(4N)}$.

Let us now determine the fusions $D_1^{(SC)} \otimes D_1^{(V)}$ and $D_1^{(SC)} \otimes D_1^{(V_-)}$ in $\mathcal{C}_{\text{Pin}^+(4N)}$. Notice that in $\mathcal{C}_{\text{Spin}(4N)}$ we have the fusion

$$
\begin{aligned}
\left( D_1^{(SC)} \otimes D_1^{(V)} \right)_{\mathcal{C}_{\text{Spin}(4N)}} &= \left( (D_1^{(S)} \oplus D_1^{(C)}) \otimes D_1^{(V)} \right)_{\mathcal{C}_{\text{Spin}(4N)}} = \left( D_1^{(C)} \oplus D_1^{(S)} \right)_{\mathcal{C}_{\text{Spin}(4N)}} \\
&= \left( D_1^{(SC)} \right)_{\mathcal{C}_{\text{Spin}(4N)}},
\end{aligned}
\tag{64}
$$

which implies that, in the category $\mathcal{C}_{\text{Pin}^+(4N)}$, only at most a single copy of $D_1^{(SC)}$ can appear in the fusions $D_1^{(SC)} \otimes D_1^{(V)}$ and $D_1^{(SC)} \otimes D_1^{(V_-)}$. To determine whether or not $D_1^{(SC)}$ appears in these fusions, we need to study the $\mathbb{Z}_2$ representations formed by morphisms from $D_1^{(SC)} \otimes D_1^{(V)}$ to $D_1^{(SC)}$ in $\mathcal{C}_{\text{Spin}(4N)}$. There is a 2-dimensional vector space of such morphisms of $\mathcal{C}_{\text{Spin}(4N)}$ spanned by a morphism $D_0^{(S \otimes C, V)}$ from $D_1^{(S)} \otimes D_1^{(C)}$ to $D_1^{(V)}$, and a morphism $D_0^{(C \otimes S, V)}$ from $D_1^{(C)} \otimes D_1^{(S)}$ to $D_1^{(V)}$. It is clear that $\mathbb{Z}_2$ outer automorphism acts as the exchange

$$
D_0^{(S \otimes C, V)} \longleftrightarrow D_0^{(C \otimes S, V)}.
\tag{65}
$$

Thus the morphism space decomposes as $1 \oplus 1_{-1}$ under the $\mathbb{Z}_2$ 0-form symmetry, where $1$ denotes the trivial $\mathbb{Z}_2$ rep and $1_{-1}$ denotes the non-trivial $\mathbb{Z}_2$ irrep. Since both $\mathbb{Z}_2$ representations are present, we learn that

$$
\begin{aligned}
D_1^{(SC)} \otimes D_1^{(V)} &= D_1^{(SC)}, \\
D_1^{(SC)} \otimes D_1^{(V_-)} &= D_1^{(SC)}
\end{aligned}
\tag{66}
$$

are the descending fusion rules in the category $\mathcal{C}_{\text{Pin}^+(4N)}$.

Finally, we consider the tensor product $D_1^{(SC)} \otimes D_1^{(SC)}$ in the category $\mathcal{C}_{\text{Pin}^+(4N)}$. Notice that in $\mathcal{C}_{\text{Spin}(4N)}$ we have the fusion

$$
\left( D_1^{(SC)} \otimes D_1^{(SC)} \right)_{\mathcal{C}_{\text{Spin}(4N)}} = \left( 2D_1^{(\text{id})} \oplus 2D_1^{(V)} \right)_{\mathcal{C}_{\text{Spin}(4N)}},
\tag{67}
$$

which implies that, in the category $\mathcal{C}_{\text{Pin}^+(4N)}$, only $D_1^{(\text{id})}$, $D_1^{(-)}$, $D_1^{(V)}$ and $D_1^{(V_-)}$ can appear in the fusion $D_1^{(SC)} \otimes D_1^{(SC)}$. To determine the precise multiplicity of these objects, we need to study the $\mathbb{Z}_2$ representations formed by morphisms from $D_1^{(SC)} \otimes D_1^{(SC)}$ to $D_1^{(\text{id})}$ in $\mathcal{C}_{\text{Spin}(4N)}$ and the $\mathbb{Z}_2$ representations formed by morphisms from $D_1^{(SC)} \otimes D_1^{(SC)}$ to $D_1^{(V)}$ in $\mathcal{C}_{\text{Spin}(4N)}$.

Let us first consider the $\mathcal{C}_{\text{Spin}(4N)}$ morphisms from $D_1^{(SC)} \otimes D_1^{(SC)}$ to $D_1^{(\text{id})}$. There is a 2-dimensional vector space of such morphisms of $\mathcal{C}_{\text{Spin}(4N)}$ spanned by a morphism $D_0^{(S \otimes S, \text{id})}$ from $D_1^{(S)} \otimes D_1^{(S)}$ to $D_1^{(\text{id})}$, and a morphism $D_0^{(C \otimes C, \text{id})}$ from $D_1^{(C)} \otimes D_1^{(C)}$ to $D_1^{(\text{id})}$. It is clear that $\mathbb{Z}_2$ outer automorphism acts as the exchange

$$
D_0^{(S \otimes S, \text{id})} \longleftrightarrow D_0^{(C \otimes C, \text{id})}.
\tag{68}
$$

Thus the morphism space decomposes as $1 \oplus 1_-$ under the $\mathbb{Z}_2$ 0-form symmetry. Since there is a single copy of both $\mathbb{Z}_2$ representations, we learn that $D_1^{(SC)} \otimes D_1^{(SC)}$ contains a single copy of $D_1^{(\text{id})}$ and a single copy of $D_1^{(-)}$ in $\mathcal{C}_{\text{Pin}^+(4N)}$.

Now consider the $\mathcal{C}_{\text{Spin}(4N)}$ morphisms from $D_1^{(SC)} \otimes D_1^{(SC)}$ to $D_1^{(V)}$. There is a 2-dimensional vector space of such morphisms of $\mathcal{C}_{\text{Spin}(4N)}$ spanned by a morphism $D_0^{(S \otimes C, V)}$ from $D_1^{(S)} \otimes D_1^{(C)}$

to $D_1^{(V)}$, and a morphism $D_0^{(C \otimes S, V)}$ from $D_1^{(C)} \otimes D_1^{(S)}$ to $D_1^{(V)}$. It is clear that $\mathbb{Z}_2$ outer automorphism acts as the exchange

$$D_0^{(S \otimes C, V)} \longleftrightarrow D_0^{(C \otimes S, V)}. \tag{69}$$

Thus the morphism space decomposes as $1 \oplus 1_-$ under the $\mathbb{Z}_2$ 0-form symmetry. Since there is a single copy of both $\mathbb{Z}_2$ representations, we learn that $D_1^{(SC)} \otimes D_1^{(SC)}$ contains a single copy of $D_1^{(V)}$ and a single copy of $D_1^{(V_-)}$ in $\mathcal{C}_{\mathrm{Pin}^+(4N)}$.

Combining everything, we learn the fusion rule

$$D_1^{(SC)} \otimes D_1^{(SC)} = D_1^{(\mathrm{id})} \oplus D_1^{(-)} \oplus D_1^{(V)} \oplus D_1^{(V_-)} \tag{70}$$

of $\mathcal{C}_{\mathrm{Pin}^+(4N)}$. Since the RHS contains objects other than $D_1^{(\mathrm{id})}$, we find that $D_1^{(SC)}$ is a non-invertible topological line defect. Thus, the category $\mathcal{C}_{\mathrm{Pin}^+(4N)}$ describes non-invertible symmetries of the $\mathrm{Pin}^+(4N)$ gauge theory.

From the fusion rules, we observe that the resulting category $\mathcal{C}_{\mathrm{Pin}^+(4N)}$ can be recognized as one of the Tambara-Yamagami categories based on the abelian group $\mathbb{Z}_2 \times \mathbb{Z}_2$. There are four such categories [59] (see also Section 5.5 of [8]), and it is a natural question to ask which one is $\mathcal{C}_{\mathrm{Pin}^+(4N)}$. The difference between the four categories is captured in the data of the associators. We can compute the associators of $\mathcal{C}_{\mathrm{Pin}^+(4N)}$ from the associators of $\mathcal{C}_{\mathrm{Spin}(4N)}$, where the latter associators are trivial as $\mathcal{C}_{\mathrm{Spin}(4N)}$ describes a non-anomalous 1-form symmetry. From this computation, we find that

$$\mathcal{C}_{\mathrm{Pin}^+(4N)} = \mathrm{Rep}(D_8), \tag{71}$$

i.e. $\mathcal{C}_{\mathrm{Pin}^+(4N)}$ is the category formed by representations of the dihedral group $D_8$, which is one of the four Tambara-Yamagami categories.

There is an alternate derivation of $\mathcal{C}_{\mathrm{Pin}^+(4N)}$ which makes it manifest that it has to be $\mathrm{Rep}(D_8)$. We first gauge the $\mathbb{Z}_2^S \times \mathbb{Z}_2^C$ 1-form symmetry to go to the pure $PSO(4N)$ Yang-Mills theory in 3d. After gauging, we obtain a dual $\mathbb{Z}_2^S \times \mathbb{Z}_2^C$ 0-form symmetry. The $\mathbb{Z}_2$ outer-automorphism is still a 0-form symmetry, and it now acts on the $\mathbb{Z}_2^S \times \mathbb{Z}_2^C$ 0-form symmetry by exchanging $\mathbb{Z}_2^S$ and $\mathbb{Z}_2^C$. Thus, the total 0-form symmetry group of the $PSO(4N)$ theory is

$$\Gamma^{(0)} = \left( \mathbb{Z}_2^S \times \mathbb{Z}_2^C \right) \rtimes \mathbb{Z}_2 = D_8, \tag{72}$$

which is the dihedral group $D_8$. On the other hand, the 1-form symmetry group of the $PSO(4N)$ theory is trivial. Now, the pure $\mathrm{Pin}^+(4N)$ Yang-Mills theory is obtained from the $PSO(4N)$ Yang-Mills theory by gauging both the $\mathbb{Z}_2^S \times \mathbb{Z}_2^C$ 0-form symmetry and the $\mathbb{Z}_2$ outer-automorphism 0-form symmetry, or in other words, by gauging the full $D_8$ 0-form symmetry. Since there is no 1-form symmetry, the category $\mathcal{C}_{PSO(4N)}$ associated to the $PSO(4N)$ theory is the trivial category

$$\mathcal{C}_{PSO(4N)} = \mathrm{Vec}, \tag{73}$$

which is the category of ungraded vector spaces, and the action of $D_8$ on this category is trivial. After gauging $D_8$, we obtain non-trivial topological line defects arising as the Wilson line defects forming representations of $D_8$, and thus we find that

$$\mathcal{C}_{\mathrm{Pin}^+(4N)} = \mathrm{Rep}(D_8). \tag{74}$$

In appendix A.1 we compute in a similar fashion the symmetry category for $\mathrm{Pin}^+(4N+2)$ and find it to be also

$$\mathcal{C}_{\mathrm{Pin}^+(4N+2)} = \mathrm{Rep}(D_8). \tag{75}$$

## 5.2 Ising $\times$ $\overline{\text{Ising}}$ from $\mathbb{Z}_2$ Gauge Theory in 3d

In this subsection, we discuss the well-known example of gauging $\mathbb{Z}_2$ electromagnetic duality in the $\mathbb{Z}_2$ topological gauge theory. It is known that the resulting TQFT obtained after such a gauging corresponds to the Drinfeld double of the Ising fusion category. Within our approach, we access the untwisted or zero flux sector of this resulting (gauged) fusion category.

Consider the topological $\mathbb{Z}_2$ gauge theory described by the action

$$S = i\pi \int_M b_1 \cup \delta a_1 \,, \tag{76}$$

where $b_1, a_1 \in C^1(M, \mathbb{Z}_2)$. The model has a 1-form symmetry group $\mathbb{Z}_2 \times \mathbb{Z}_2$ generated by the electric and magnetic lines $D_1^{(e)}(\gamma) = \exp\left\{ i\pi \oint_\gamma a_1 \right\}$ and $D_1^{(m)}(\gamma) = \exp\left\{ i\pi \oint_\gamma b_1 \right\}$ respectively. We denote the corresponding $\mathbb{Z}_2$ subgroups as $\mathbb{Z}_2^e$ and $\mathbb{Z}_2^m$ and thus the 1-form symmetry as

$$\Gamma^{(1)} = \mathbb{Z}_2^e \times \mathbb{Z}_2^m \,. \tag{77}$$

The diagonal subgroup of $\Gamma^{(1)}$ is generated by a fermionic line $D_1^{(f)}(\gamma) = D_1^{(e)}(\gamma) D_1^{(m)}(\gamma)$. The topological lines form a 1-category $\mathcal{C}_{\mathbb{Z}_2}$ whose simple objects are

$$\mathcal{C}_{\mathbb{Z}_2}^{\text{ob}} = \left\{ D_1^{(\text{id})}, D_1^{(e)}, D_1^{(m)}, D_1^{(f)} \right\} \,. \tag{78}$$

Additionally, there is 0-form symmetry

$$\Gamma^{(0)} = \mathbb{Z}_2^{em} \,, \tag{79}$$

which acts on the objects of the 1-category by exchanging as

$$D_1^{(e)} \longleftrightarrow D_1^{(m)} \,, \tag{80}$$

and leaves the remaining objects $D_1^{(\text{id})}$ and $D_1^{(f)}$ invariant. Note that this is precisely the same symmetry structure as that obtained in pure $\text{Spin}(4N)$ gauge theory described in section 5.1. Upon gauging the 0-form symmetry $\mathbb{Z}_2^{em}$, the same analysis as in section 5.1 goes through i.e. the objects $D_1^{(e)}$ and $D_1^{(m)}$ combine into a single object $D_1^{(e,m)} := (D_1^{(e)} \oplus D_1^{(m)})_{\mathcal{C}_{\mathbb{Z}_2}}$ as they form a single orbit under the $\mathbb{Z}_2^{em}$ action. Meanwhile, the objects $D_1^{(\text{id})}$ and $D_1^{(f)}$, each split into two objects which carry an additional $\mathbb{Z}_2$ representation label, i.e.,

$$\left( D_1^{(\text{id})}, D_1^{(f)} \right)_{\mathcal{C}_{\mathbb{Z}_2}} \longmapsto \left( D_1^{(\text{id})}, D_1^{(-)}, D_1^{(f)}, D_1^{(f-)} \right)_{\mathcal{C}_{\mathbb{Z}_2/\Gamma^{(0)}}} \,. \tag{81}$$

There is a crucial difference between the symmetry category obtained after gauging $\mathbb{Z}_2^{em}$ in the topological $\mathbb{Z}_2$ gauge theory and the analagous symmetry category obtained in the $\text{Pin}_+(4N)$ theory upon gauging the outer-automorphism 0-form $\mathbb{Z}_2$ symmetry. Since the $\mathbb{Z}_2$ gauge theory is topological, the resulting theory obtained after gauging $\mathbb{Z}_2^{em}$ contains additional topological line operators corresponding to the fluxes or twisted sector operators of $\mathbb{Z}_2^{em}$. In contrast, since $\text{Spin}(4N)$ is not topological, the $\mathbb{Z}_2$ flux lines are not topological and hence do not contribute to the symmetry category of the $\text{Pin}_+(4N)$ gauge theory. Let us recall the fusion rules computed in section 5.1, with a relabelling of objects $(S, C, V) \to (e, m, f)$. Notably, these form a subcategory of the fusion category Ising $\times$ $\overline{\text{Ising}}$. The objects of which are

$$\begin{aligned} \mathcal{C}_{\text{Ising}}^{\text{ob}} &= \left\{ D_1^{(1)}, D_1^{(\psi)}, D_1^{(\sigma)} \right\} \,, \\ \mathcal{C}_{\overline{\text{Ising}}}^{\text{ob}} &= \left\{ D_1^{(1)}, D_1^{(\bar{\psi})}, D_1^{(\bar{\sigma})} \right\} \,. \end{aligned} \tag{82}$$

The objects in the zero flux sector of the category $\mathcal{C}_{\mathbb{Z}_2/\Gamma^{(0)}}$ can be identified with a sub-category of Ising $\times$ $\overline{\text{Ising}}$ by identifying the labels $(\text{id}) \sim (1)$, $(f) \sim (\psi)$, $(f-) \sim (\bar{\psi})$, $(-) \sim (\psi\bar{\psi})$ and $(e,m) \sim (\sigma\bar{\sigma})$. Similarly, the remaining objects in Ising $\times$ $\overline{\text{Ising}}$ can be identified with the flux lines of the gauged category $\mathcal{C}_{\mathbb{Z}_2/\Gamma^{(0)}}$ [30,31]. The $\mathbb{Z}_2$ gauge theory can be obtained by gauging a dual 1-form $\mathbb{Z}_2$ symmetry in the Ising $\times$ $\overline{\text{Ising}}$ theory, generated by $(-) \sim (\psi\bar{\psi})$.

## 5.3 Pure Spin$(8) \rtimes S_3$ Gauge Theory in 3d

Consider Spin$(8)$ gauge theory, which has a

$$\Gamma^{(0)} = S_3 \tag{83}$$

0-form outer-automorphism symmetry, which acts on 1-form symmetry

$$\Gamma^{(1)} = \mathbb{Z}_2^S \times \mathbb{Z}_2^C, \tag{84}$$

by permuting the generators of $\mathbb{Z}_2^S$, $\mathbb{Z}_2^C$ and $\mathbb{Z}_2^V$. We will now gauge this 0-form symmetry and obtain a 3d Spin$(8) \rtimes S_3$ gauge theory. To do this we first gauge a $\mathbb{Z}_3$ subgroup and then the full $S_3$.

### 5.3.1 Gauging $\mathbb{Z}_3$ Spin$(8)$ in 3d

Let us gauge the $\mathbb{Z}_3$ subgroup of $S_3$ 0-form symmetry, which acts as cyclic permutations

$$\mathbb{Z}_2^S \to \mathbb{Z}_2^C \to \mathbb{Z}_2^V \to \mathbb{Z}_2^S. \tag{85}$$

This produces pure Yang-Mills theory in 3d with gauge group

$$\text{Spin}(8) \rtimes \mathbb{Z}_3. \tag{86}$$

The category $\mathcal{C}_{\text{Spin}}$ describing 1-form symmetries of Spin$(8)$ theory descends to a category $\mathcal{C}_{\mathbb{Z}_3}$ of topological lines and local operators in the Spin$(8) \rtimes \mathbb{Z}_3$ theory. Its simple objects are

$$\mathcal{C}_{\mathbb{Z}_3}^{\text{ob}} = \{D_1^{(\text{id})}, D_1^{(\omega)}, D_1^{(\omega^2)}, D_1^{(SCV)}\}, \tag{87}$$

where $D_1^{(\text{id})}$, $D_1^{(\omega)}$ and $D_1^{(\omega^2)}$ are the $\mathbb{Z}_3$ Wilson lines whose fusion obeys group law of $\mathbb{Z}_3$, and

$$D_1^{(SCV)} = \left(D_1^{(S)} \oplus D_1^{(C)} \oplus D_1^{(V)}\right)_{\mathcal{C}_{\text{Spin}}}, \tag{88}$$

as a $\mathbb{Z}_3$ invariant object of the category $\mathcal{C}_{\text{Spin}}$ associated to the symmetries of the Spin$(8)$ theory. Since the stabilizer of $D_1^{(S)}$ is trivial, there are no Wilson line dressings of $D_1^{(SCV)}$ in $\mathcal{C}_{\mathbb{Z}_3}$.

The fusion rules of $D_1^{(SCV)}$ with the Wilson lines are simply

$$D_1^{(SCV)} \otimes D_1^{(i)} = D_1^{(SCV)}, \tag{89}$$

with $i \in \{\text{id}, \omega, \omega^2\}$.

To determine the fusion $D_1^{(SCV)} \otimes D_1^{(SCV)}$, we need to study morphism spaces $D_1^{(SCV)} \otimes D_1^{(SCV)} \to D_1^{(\text{id})}$ and $D_1^{(SCV)} \otimes D_1^{(SCV)} \to D_1^{(SCV)}$ in the category $\mathcal{C}_{\text{Spin}}$. The morphism space $D_1^{(SCV)} \otimes D_1^{(SCV)} \to D_1^{(\text{id})}$ is three-dimensional, being generated by morphisms

$$\begin{aligned}
D_0^{(S,S;\text{id})} &: D_1^{(S)} \otimes D_1^{(S)} \to D_1^{(\text{id})}, \\
D_0^{(C,C;\text{id})} &: D_1^{(C)} \otimes D_1^{(C)} \to D_1^{(\text{id})}, \\
D_0^{(V,V;\text{id})} &: D_1^{(V)} \otimes D_1^{(V)} \to D_1^{(\text{id})},
\end{aligned} \tag{90}$$

which decomposes in terms of representations of $\mathbb{Z}_3$ as

$$3 = 1 \oplus 1_\omega \oplus 1_{\omega^2} \,, \tag{91}$$

where $1_\omega$ is the representation in which the generator of $\mathbb{Z}_3$ acts by multiplication by $\omega$, and $1_\omega^2$ is the representation in which the generator of $\mathbb{Z}_3$ acts by multiplication by $\omega^2$. On the other hand, the morphism space $D_1^{(SCV)} \otimes D_1^{(SCV)} \to D_1^{(SCV)}$ is six-dimensional, being generated by morphisms

$$\begin{aligned}
D_0^{(S,C;V)} &: D_1^{(S)} \otimes D_1^{(C)} \to D_1^{(V)} \,, \\
D_0^{(C,V;S)} &: D_1^{(C)} \otimes D_1^{(V)} \to D_1^{(S)} \,, \\
D_0^{(V,S;C)} &: D_1^{(V)} \otimes D_1^{(S)} \to D_1^{(C)} \,, \\
D_0^{(C,S;V)} &: D_1^{(C)} \otimes D_1^{(S)} \to D_1^{(V)} \,, \\
D_0^{(V,C;S)} &: D_1^{(V)} \otimes D_1^{(C)} \to D_1^{(S)} \,, \\
D_0^{(S,V;C)} &: D_1^{(S)} \otimes D_1^{(V)} \to D_1^{(C)} \,,
\end{aligned} \tag{92}$$

which decomposes in terms of representations of $\mathbb{Z}_3$ as

$$6 = 2 \left( 1 \oplus 1_\omega \oplus 1_{\omega^2} \right) \,. \tag{93}$$

From this, we learn that

$$D_1^{(SCV)} \otimes D_1^{(SCV)} = D_1^{(\mathrm{id})} \oplus D_1^{(\omega)} \oplus D_1^{(\omega^2)} \oplus 2 D_1^{(SCV)} \,. \tag{94}$$

In fact, we can recognize the full category $\mathcal{C}_{\mathbb{Z}_3}$ as

$$\mathcal{C}_{\mathbb{Z}_3} = \mathrm{Rep}(A_4) \,, \tag{95}$$

where $A_4$ is the order 12 alternating group permuting 4 elements. This follows from the fact that

$$A_4 = (\mathbb{Z}_2 \times \mathbb{Z}_2) \rtimes \mathbb{Z}_3 \tag{96}$$

is part of the 0-form symmetry of the $PSO(8)$ theory, which is being gauged to construct the $\mathrm{Spin}(8) \rtimes \mathbb{Z}_3$ theory.

### 5.3.2 Gauging $S_3$ in $\mathrm{Spin}(8)$ Gauge Theory in 3d

Now consider gauging the full $S_3$ 0-form symmetry of the $\mathrm{Spin}(8)$ theory to construct pure Yang-Mills theory in 3d with gauge group $\mathrm{Spin}(8) \rtimes S_3$. This can be obtained by gauging the $\mathbb{Z}_2$ 0-form symmetry of the $\mathrm{Spin}(8) \rtimes \mathbb{Z}_3$ theory. Let us call the category associated to the $\mathrm{Spin}(8) \rtimes S_3$ theory descending from $\mathcal{C}_{\mathbb{Z}_3}$ as $\mathcal{C}_{S_3}$.

Let us deduce simple objects of $\mathcal{C}_{S_3}$. First of all, we obtain Wilson lines

$$\left\{ D_1^{(\mathrm{id})}, D_1^{(-)} \right\} \,, \tag{97}$$

associated to the gauged $\mathbb{Z}_2$. Then, we obtain simple objects of $\mathcal{C}_{S_3}$ from orbits formed by simple objects of $\mathcal{C}_{\mathbb{Z}_3}$ under the $\mathbb{Z}_2$ action. From this we obtain

$$D_1^{(\omega\omega^2)} = \left( D_1^{(\omega)} \oplus D_1^{(\omega^2)} \right)_{\mathcal{C}_{\mathbb{Z}_3}} \,, \tag{98}$$

and

$$D_1^{(SCV)} = \left( D_1^{(SCV)} \right)_{\mathcal{C}_{\mathbb{Z}_3}} \,. \tag{99}$$

Since the stabilizer of $D_1^{(SCV)}$ is non-trivial, we can dress it with a $\mathbb{Z}_2$ Wilson line to obtain another simple object

$$D_1^{(SCV_-)}. \tag{100}$$

In total, the simple objects of $\mathcal{C}_{S_3}$ are

$$\mathcal{C}_{S_3}^{\text{ob}} = \left\{ D_1^{(\text{id})}, D_1^{(-)}, D_1^{(\omega\omega^2)}, D_1^{(SCV)}, D_1^{(SCV_-)} \right\}. \tag{101}$$

Some of the straightforward fusion rules are

$$\begin{aligned}
D_1^{(-)} \otimes D_1^{(\omega\omega^2)} &= D_1^{(\omega\omega^2)}, \\
D_1^{(-)} \otimes D_1^{(SCV)} &= D_1^{(SCV_-)}, \\
D_1^{(-)} \otimes D_1^{(SCV_-)} &= D_1^{(SCV)}.
\end{aligned} \tag{102}$$

To determine $D_1^{(\omega\omega^2)} \otimes D_1^{(SCV)}$ and $D_1^{(\omega\omega^2)} \otimes D_1^{(SCV_-)}$, we note that the morphism space $D_1^{(\omega\omega^2)} \otimes D_1^{(SCV)} \to D_1^{(SCV)}$ in $\mathcal{C}_{\mathbb{Z}_3}$ is two-dimensional, spanned by the morphisms

$$\begin{aligned}
D_0^{(\omega,SCV;SCV)} &: \ D_1^{(\omega)} \otimes D_1^{(SCV)} \to D_1^{(SCV)}, \\
D_0^{(\omega^2,SCV;SCV)} &: \ D_1^{(\omega^2)} \otimes D_1^{(SCV)} \to D_1^{(SCV)},
\end{aligned} \tag{103}$$

which decomposes under the $\mathbb{Z}_2$ action as

$$2 = 1 \oplus 1_-, \tag{104}$$

leading to the fusion rules

$$\begin{aligned}
D_1^{(\omega\omega^2)} \otimes D_1^{(SCV)} &= D_1^{(SCV)} \oplus D_1^{(SCV_-)}, \\
D_1^{(\omega\omega^2)} \otimes D_1^{(SCV_-)} &= D_1^{(SCV)} \oplus D_1^{(SCV_-)}.
\end{aligned} \tag{105}$$

To determine $D_1^{(\omega\omega^2)} \otimes D_1^{(\omega\omega^2)}$, we note that the morphism space $D_1^{(\omega\omega^2)} \otimes D_1^{(\omega\omega^2)} \to D_1^{(\text{id})}$ in $\mathcal{C}_{\mathbb{Z}_3}$ is two-dimensional, spanned by the morphisms

$$\begin{aligned}
D_0^{(\omega,\omega^2;\text{id})} &: \ D_1^{(\omega)} \otimes D_1^{(\omega^2)} \to D_1^{(\text{id})}, \\
D_0^{(\omega^2,\omega;\text{id})} &: \ D_1^{(\omega^2)} \otimes D_1^{(\omega)} \to D_1^{(\text{id})},
\end{aligned} \tag{106}$$

which decomposes under the $\mathbb{Z}_2$ action as

$$2 = 1 \oplus 1_-. \tag{107}$$

On the other hand, the morphism space $D_1^{(\omega\omega^2)} \otimes D_1^{(\omega\omega^2)} \to D_1^{(\omega\omega^2)}$ in $\mathcal{C}_{\mathbb{Z}_3}$ is also two-dimensional, spanned by the morphisms

$$\begin{aligned}
D_0^{(\omega,\omega;\omega^2)} &: \ D_1^{(\omega)} \otimes D_1^{(\omega)} \to D_1^{(\omega^2)}, \\
D_0^{(\omega^2,\omega^2;\omega)} &: \ D_1^{(\omega^2)} \otimes D_1^{(\omega^2)} \to D_1^{(\omega)},
\end{aligned} \tag{108}$$

which decomposes under the $\mathbb{Z}_2$ action as

$$2 = 1 \oplus 1_-. \tag{109}$$

Combining the above, we are lead to the fusion rule

$$D_1^{(\omega\omega^2)} \otimes D_1^{(\omega\omega^2)} = D_1^{(\mathrm{id})} \oplus D_1^{(-)} \oplus D_1^{(\omega\omega^2)} \,. \tag{110}$$

Finally, to determine fusion rules $D_1^{(SCV)} \otimes D_1^{(SCV)}$, $D_1^{(SCV_-)} \otimes D_1^{(SCV_-)}$ and $D_1^{(SCV)} \otimes D_1^{(SCV_-)}$, we note that the morphism space $D_1^{(SCV)} \otimes D_1^{(SCV)} \to D_1^{(\mathrm{id})}$ in $\mathcal{C}_{\mathbb{Z}_3}$ is one-dimensional which is left invariant by the $\mathbb{Z}_2$ action. The morphism space $D_1^{(SCV)} \otimes D_1^{(SCV)} \to D_1^{(\omega\omega^2)}$ in $\mathcal{C}_{\mathbb{Z}_3}$ is two-dimensional, spanned by the morphisms

$$\begin{aligned} D_0^{(SCV,SCV;\omega)} &: \ D_1^{(SCV)} \otimes D_1^{(SCV)} \to D_1^{(\omega)} \,, \\ D_0^{(SCV,SCV;\omega^2)} &: \ D_1^{(SCV)} \otimes D_1^{(SCV)} \to D_1^{(\omega^2)} \,, \end{aligned} \tag{111}$$

which decomposes under the $\mathbb{Z}_2$ action as

$$2 = 1 \oplus 1_- \,. \tag{112}$$

The morphism space $D_1^{(SCV)} \otimes D_1^{(SCV)} \to D_1^{(SCV)}$ in $\mathcal{C}_{\mathbb{Z}_3}$ is also two-dimensional, spanned by the morphisms

$$\begin{aligned} D_0^{(SCV,SCV;SCV)(a)} &= \left( D_0^{(S,C;V)} + D_0^{(C,V;S)} + D_0^{(V,S;C)} \right)_{\mathcal{C}_{\mathrm{Spin}}} \,, \\ D_0^{(SCV,SCV;SCV)(b)} &= \left( D_0^{(C,S;V)} + D_0^{(V,C;S)} + D_0^{(S,V;C)} \right)_{\mathcal{C}_{\mathrm{Spin}}} \,, \end{aligned} \tag{113}$$

where we have expressed them as morphisms of $\mathcal{C}_{\mathrm{Spin}}$ to make the action of $\mathbb{Z}_2$ on them manifest. The $\mathbb{Z}_2$ acts as the exchange

$$D_0^{(SCV,SCV;SCV)(a)} \longleftrightarrow D_0^{(SCV,SCV;SCV)(b)} \,. \tag{114}$$

Thus, the morphism space $D_1^{(SCV)} \otimes D_1^{(SCV)} \to D_1^{(SCV)}$ in $\mathcal{C}_{\mathbb{Z}_3}$ decomposes as

$$2 = 1 \oplus 1_- \tag{115}$$

under $\mathbb{Z}_2$. Combining the above, we are lead to the fusion rules

$$\begin{aligned} D_1^{(SCV)} \otimes D_1^{(SCV)} &= D_1^{(\mathrm{id})} \oplus D_1^{(\omega\omega^2)} \oplus D_1^{(SCV)} \oplus D_1^{(SCV_-)} \,, \\ D_1^{(SCV_-)} \otimes D_1^{(SCV_-)} &= D_1^{(\mathrm{id})} \oplus D_1^{(\omega\omega^2)} \oplus D_1^{(SCV)} \oplus D_1^{(SCV_-)} \,, \\ D_1^{(SCV)} \otimes D_1^{(SCV_-)} &= D_1^{(-)} \oplus D_1^{(\omega\omega^2)} \oplus D_1^{(SCV)} \oplus D_1^{(SCV_-)} \,. \end{aligned} \tag{116}$$

We can in fact recognize the full category $\mathcal{C}_{S_3}$ as

$$\mathcal{C}_{S_3} = \mathrm{Rep}(S_4) \,. \tag{117}$$

This follows from the fact that

$$S_4 = (\mathbb{Z}_2 \times \mathbb{Z}_2) \rtimes S_3 \tag{118}$$

is the 0-form symmetry group of the $PSO(8)$ theory, which is being gauged to construct the $\mathrm{Spin}(8) \rtimes S_3$ theory.

## 5.4 Pure $O(2)$ Gauge Theory in 3d

Pure Yang-Mills theory in 3d with gauge group $O(2)$ can be constructed from pure Yang-Mills theory in 3d with gauge group $U(1)$ by gauging the charge conjugation $\mathbb{Z}_2$ 0-form symmetry. The $U(1)$ theory has a

$$\Gamma^{(1)} = U(1) \tag{119}$$

1-form symmetry. The

$$\Gamma^{(0)} = \mathbb{Z}_2 \tag{120}$$

charge conjugation symmetry acts by complex conjugation on $\Gamma^{(1)}$. This 1-form symmetry descends to a *continuous* non-invertible categorical symmetry in the $O(2)$ theory.

The symmetry defects that implement flat $U(1)$ backgrounds can be organized into a 1-category $\mathcal{C}_{U(1)}$, which has simple objects

$$\mathcal{C}_{U(1)}^{\mathrm{ob}} = \left\{ D_1^{(\theta)} \,\middle|\, \theta \in \mathbb{R}/\mathbb{Z} \right\}. \tag{121}$$

Physically, inserting a single simple object $D_1^{(\theta)}$ along a 1-cycle corresponds to turning on a symmetry background with holonomy $\theta$ around the linking 1-cycle. $\mathbb{Z}_2$ leaves invariant $D_1^{(0)}$ and $D_1^{(1/2)}$, while acting as the exchange

$$D_1^{(\theta)} \longleftrightarrow D_1^{(-\theta)}, \tag{122}$$

for $\theta \neq 0, 1/2$. The fusion of simple objects follows the additive group law of $\mathbb{R}/\mathbb{Z}$.

Now we gauge $\mathbb{Z}_2$ charge conjugation to obtain the category $\mathcal{C}_{O(2)}$ descending from $\mathcal{C}_{U(1)}$. The simple objects of $\mathcal{C}_{O(2)}$ are

$$\mathcal{C}_{O(2)}^{\mathrm{ob}} = \left\{ D_1^{(0)}, D_1^{(-)}, D_1^{(1/2)}, D_1^{(1/2,-)}, L_1^{(\theta)} \,\middle|\, 0 < \theta < 1/2 \right\}, \tag{123}$$

where

$$L_1^{(\theta)} = \left( D_1^{(\theta)} \oplus D_1^{(-\theta)} \right)_{\mathcal{C}_{U(1)}}, \tag{124}$$

as object in $\mathcal{C}_{U(1)}$. Since $D_1^{(0)}$ and $D_1^{(1/2)}$ have $\mathbb{Z}_2$ stabilizers, they lead to the simple objects $D_1^{(-)}$ and $D_1^{(1/2,-)}$ by dressing with $\mathbb{Z}_2$ Wilson line. Let us now discuss the fusion rules of these objects. The fusion rules of $D_1^{(0)}, D_1^{(-)}, D_1^{(1/2)}, D_1^{(1/2,-)}$ follow the group law of $\mathbb{Z}_2 \times \mathbb{Z}_2$. To compute $D_1^{(-)} \otimes L_1^{(\theta)}$, we note that the endomorphism space of $L_1^{(\theta)}$ in $\mathcal{C}_{U(1)}$ is two-dimensional, which decomposes as $2 = 1 \oplus 1_-$ under $\mathbb{Z}_2$, implying

$$D_1^{(-)} \otimes L_1^{(\theta)} = L_1^{(\theta)} \tag{125}$$

in $\mathcal{C}_{O(2)}$.

To compute $D_1^{(1/2)} \otimes L_1^{(\theta)}$ and $D_1^{(1/2,-)} \otimes L_1^{(\theta)}$, we note that the only non-trivial morphism space is $D_1^{(1/2)} \otimes L_1^{(\theta)} \to L_1^{(1/2-\theta)}$ in $\mathcal{C}_{U(1)}$, which decomposes as $2 = 1 \oplus 1_-$ under $\mathbb{Z}_2$, implying

$$\begin{aligned}
D_1^{(1/2)} \otimes L_1^{(\theta)} &= L_1^{(1/2-\theta)}, \\
D_1^{(1/2,-)} \otimes L_1^{(\theta)} &= L_1^{(1/2-\theta)}
\end{aligned} \tag{126}$$

in $\mathcal{C}_{O(2)}$.

Let us now turn to the fusion rules $L_1^{(\theta)} \otimes L_1^{(\theta')}$. For $\theta' \neq 1/2 - \theta$ and $\theta' \neq \theta$, only $L_1^{\left(|\theta+\theta'|_{1/2}\right)}$ and $L_1^{\left(|\theta-\theta'|_{1/2}\right)}$ can appear, where

$$|\alpha|_{1/2} = \alpha + n, \tag{127}$$

if $0 < \alpha + n < 1/2$ for some $n \in \mathbb{Z}$, and

$$|\alpha|_{1/2} = -\alpha + n, \tag{128}$$

if $0 < -\alpha + n < 1/2$ for some $n \in \mathbb{Z}$. The reader can check that there is a single $\mathbb{Z}_2$ invariant morphism for both possibilities $L_1^{\left(|\theta+\theta'|_{1/2}\right)}$ and $L_1^{\left(|\theta-\theta'|_{1/2}\right)}$, implying the fusion rule

$$L_1^{(\theta)} \otimes L_1^{(\theta')} = L_1^{\left(|\theta+\theta'|_{1/2}\right)} \oplus L_1^{\left(|\theta-\theta'|_{1/2}\right)} \tag{129}$$

in $\mathcal{C}_{O(2)}$.

Now let us consider $L_1^{(\theta)} \otimes L_1^{(1/2-\theta)}$. For $\theta \neq 1/4$, the possible simple objects that can appear in this fusion are $D_1^{(1/2)}$, $D_1^{(1/2,-)}$ and $L_1^{\left(|2\theta-1/2|_{1/2}\right)}$. The morphism space $L_1^{(\theta)} \otimes L_1^{(1/2-\theta)} \to D_1^{(1/2)}$ in $\mathcal{C}_{U(1)}$ decomposes as $2 = 1 \oplus 1_-$ under $\mathbb{Z}_2$, implying that both $D_1^{(1/2)}, D_1^{(1/2,-)}$ appear in the corresponding fusion in $\mathcal{C}_{O(2)}$ with multiplicities 1. On the other hand, the morphism space $L_1^{(\theta)} \otimes L_1^{(1/2-\theta)} \to L_1^{\left(|2\theta-1/2|_{1/2}\right)}$ in $\mathcal{C}_{U(1)}$ has a single morphism invariant under $\mathbb{Z}_2$, implying that the total fusion rule is

$$L_1^{(\theta)} \otimes L_1^{(1/2-\theta)} = D_1^{(1/2)} \oplus D_1^{(1/2,-)} \oplus L_1^{\left(|2\theta-1/2|_{1/2}\right)} \tag{130}$$

in $\mathcal{C}_{O(2)}$.

In a similar fashion, we can compute $L_1^{(\theta)} \otimes L_1^{(\theta)}$ for $\theta \neq 1/4$, which is found to be

$$L_1^{(\theta)} \otimes L_1^{(\theta)} = D_1^{(0)} \oplus D_1^{(-)} \oplus L_1^{\left(|2\theta|_{1/2}\right)} \tag{131}$$

in $\mathcal{C}_{O(2)}$.

Finally, we are left with the computation of $L_1^{(1/4)} \otimes L_1^{(1/4)}$. The possible simple objects in the fusion are $D_1^{(1/2)}, D_1^{(1/2,-)}, D_1^{(0)}$ and $D_1^{(-)}$. The reader can check that all these appear with multiplicity one, leading to the fusion rule

$$L_1^{(1/4)} \otimes L_1^{(1/4)} = D_1^{(0)} \oplus D_1^{(-)} \oplus D_1^{(1/2)} \oplus D_1^{(1/2,-)} \tag{132}$$

in $\mathcal{C}_{O(2)}$.

# 6 Examples: Non-Invertible 2-Categorical Symmetries in 4d

In this section we provide examples of 4d theories whose topological surface defects, line defects and local operators form a non-invertible symmetry described by a 2-category. All the examples we discuss can be obtained by gauging an invertible 0-form symmetry of 4d theories containing invertible 1-form symmetries upon which the 0-form symmetry acts non-trivially. Most of the symmetry 2-categories we discuss describe discrete symmetries, but we also provide an example in subsection 6.3, of a symmetry 2-category describing continuous symmetries, or in other words a *continuous symmetry 2-category*.

We discuss the $\text{Pin}^+(4N)$ 4d Yang-Mills theory in section 6.1 and the non-abelian gauging by $S_3$ of 4d $\text{Spin}(8)$ Yang-Mills. The $O(2)$ theory is discusssed in section 6.3, and the closely related principle extension of $SU(N)$, $\widetilde{SU(N)}$, in appendix A.2. To illustrate that such non-invertible symmetries also arise beyond pure gauge theories, we provide a quiver example in appendix A.3.

In all examples we find that fusions of surfaces lead to surfaces with condensation on top of them. In most of the examples, the condensation is described by gauging of an invertible 0-form symmetry living on the surface defect. However, we also find an example where we have to gauge a non-invertible categorical symmetry of the surface defect. See equation (193) and discussion around it.

The most non-trivial example in terms of fusions is the discrete gauge theory in section 6.4, which has two layers of non-invertibles in the higher category, i.e. we have both non-invertible genuine line operators and non-invertible genuine surface operators.

## 6.1 Pure $\text{Pin}^+(4N)$ Gauge Theory in 4d

We start with 4d $\text{Spin}(4N)$ pure Yang-Mills theory which has a $\mathbb{Z}_2 \times \mathbb{Z}_2$ 1-form symmetry on which a $\mathbb{Z}_2$ 0-form symmetry acts non-trivially. Gauging the $\mathbb{Z}_2$ 0-form symmetry leads to the 4d $\text{Pin}^+(4N)$ pure Yang-Mills theory. The $\mathbb{Z}_2 \times \mathbb{Z}_2$ 1-form symmetry of the $\text{Spin}(4N)$ theory descends to a non-invertible 2-categorical symmetry in the $\text{Pin}^+(4N)$ theory. We discuss the topological defects in the two theories before and after gauging, including their fusion algebra. In a later section, we derive these fusion rules using a different approach and find agreement.

The 1-form symmetry of the $\text{Spin}(4N)$ theory is

$$\Gamma^{(1)} = \mathbb{Z}_2^S \times \mathbb{Z}_2^C \,. \tag{133}$$

As before, we represent by $\mathbb{Z}_2^V$ the diagonal $\mathbb{Z}_2$ of $\mathbb{Z}_2^S$ and $\mathbb{Z}_2^C$. The theory has a

$$\Gamma^{(0)} = \mathbb{Z}_2 \tag{134}$$

outer-automorphism 0-form symmetry which exchanges $\mathbb{Z}_2^S$ and $\mathbb{Z}_2^C$, while leaving $\mathbb{Z}_2^V$ invariant.

The 1-form symmetry $\Gamma^{(1)}$ corresponds to a rather trivial 2-category $\mathcal{C}_{\text{Spin}(4N)}$, whose simple objects are

$$\mathcal{C}_{\text{Spin}(4N)}^{\text{ob}} = \left\{ D_2^{(\text{id})}, D_2^{(S)}, D_2^{(C)}, D_2^{(V)} \right\} \,, \tag{135}$$

where $D_2^{(\text{id})}$ is the identity surface defect, while $D_2^{(i)}$ for $i \in \{S, C, V\}$ is the topological surface defect corresponding to the generator of $\mathbb{Z}_2^i$. The fusion of these surface defects follows the group law of $\Gamma^{(1)}$

$$D_2^{(i)} \otimes D_2^{(j)} = D_2^{(ij)} \,, \tag{136}$$

with $ij \in \Gamma^{(1)}$.

There is a single simple 1-endomorphism for each simple object, which we denote as $D_1^{(i)}$ for $i \in \{\text{id}, S, C, V\}$. It can be understood as the identity line defect living on each simple surface defect $D_2^{(i)}$. There are no 1-morphisms between two distinct simple objects. Thus, the full set of simple 1-endomorphisms of simple objects is

$$\mathcal{C}_{\text{Spin}(4N)}^{\text{1-endo}} = \left\{ D_1^{(\text{id})}, D_1^{(S)}, D_1^{(C)}, D_1^{(V)} \right\} \,. \tag{137}$$

The fusion $\otimes$ for 1-endomorphisms follows the group law of $\Gamma^{(1)}$

$$D_1^{(i)} \otimes D_1^{(j)} = D_1^{(ij)} \,, \tag{138}$$

and the fusion $\otimes_{D_2^{(i)}}$ is trivially

$$D_1^{(i)} \otimes_{D_2^{(i)}} D_1^{(i)} = D_1^{(i)} . \tag{139}$$

The $\Gamma^{(0)} = \mathbb{Z}_2$ outer-automorphism 0-form symmetry acts on $\mathcal{C}_{\mathrm{Spin}(4N)}$ as

$$D_i^{(S)} \longleftrightarrow D_i^{(C)} , \tag{140}$$

for each $i \in \{1,2\}$, while leaving invariant $D_i^{(\mathrm{id})}$ and $D_i^{(V)}$.

We now gauge the outer automorphism $\mathbb{Z}_2$, which results in the $\mathrm{Pin}^+(4N)$ gauge theory. Let us call the resulting symmetry 2-category as $\mathcal{C}_{\mathrm{Pin}^+(4N)}$. The objects of $\mathcal{C}_{\mathrm{Pin}^+(4N)}$ modulo condensations are the objects of $\mathcal{C}_{\mathrm{Spin}(4N)}$ left invariant by the $\mathbb{Z}_2$ action. Thus, the simple objects of $\mathcal{C}_{\mathrm{Pin}^+(4N)}$ modulo condensations are[5]

$$\mathcal{C}_{\mathrm{Pin}^+(4N)}^{\mathrm{ob}} = \left\{ D_2^{(\mathrm{id})}, D_2^{(SC)}, D_2^{(V)} \right\} , \tag{141}$$

where

$$D_2^{(SC)} = \left( D_2^{(S)} \oplus D_2^{(C)} \right)_{\mathcal{C}_{\mathrm{Spin}(4N)}} , \tag{142}$$

as an object of the 2-category $\mathcal{C}_{\mathrm{Spin}(4N)}$.

Now let us discuss 1-morphisms between the objects appearing in the set $\mathcal{C}_{\mathrm{Pin}^+(4N)}^{\mathrm{ob}}$. Since there are no non-identity simple 1-endomorphisms of the identity object in $\mathcal{C}_{\mathrm{Spin}(4N)}$, the simple 1-endomorphisms of the identity object $D_2^{(\mathrm{id})}$ in $\mathcal{C}_{\mathrm{Pin}^+(4N)}$ are

$$\left\{ D_1^{(\mathrm{id})}, D_1^{(-)} \right\} , \tag{143}$$

which are the Wilson line defects for the gauged $\mathbb{Z}_2$. Similarly, since the simple 1-endomorphism $D_1^{(V)}$ in $\mathcal{C}_{\mathrm{Spin}(4N)}$ is left invariant by the $\mathbb{Z}_2$ action, the simple 1-endomorphisms of the object $D_2^{(V)}$ in the 2-category $\mathcal{C}_{\mathrm{Pin}^+(4N)}$ are

$$\left\{ D_1^{(V)}, D_1^{(V_-)} \right\} , \tag{144}$$

where $D_1^{(V)}$ is the identity 1-endomorphism on $D_2^{(V)}$, and $D_1^{(V_-)}$ is obtained by dressing $D_1^{(V)}$ with the non-trivial $\mathbb{Z}_2$ Wilson line. On the other hand, there is only the identity 1-endomorphism $D_1^{(SC)}$ of $D_2^{(SC)}$, which can be expressed as

$$D_1^{(SC)} = \left( D_1^{(S)} \oplus D_1^{(C)} \right)_{\mathcal{C}_{\mathrm{Spin}(4N)}} , \tag{145}$$

as a 1-morphism of $\mathcal{C}_{\mathrm{Spin}(4N)}$. Thus,

$$\mathcal{C}_{\mathrm{Pin}^+(4N)}^{\text{1-endo}} = \left\{ D_1^{(\mathrm{id})}, D_1^{(-)}, D_1^{(SC)}, D_1^{(V)}, D_1^{(V_-)} \right\} \tag{146}$$

are the simple 1-endomorphisms of simple objects $\mathcal{C}_{\mathrm{Pin}^+(4N)}^{\mathrm{ob}}$.

Let us deduce the fusion rules of the objects in $\mathcal{C}_{\mathrm{Pin}^+(4N)}^{\mathrm{ob}}$. First of all, we have

$$\begin{aligned} D_2^{(\mathrm{id})} \otimes D_2^{(V)} &= D_2^{(V)} , \\ D_2^{(V)} \otimes D_2^{(V)} &= D_2^{(\mathrm{id})} . \end{aligned} \tag{147}$$

---

[5]The simple objects of the symmetry category include the condensation defects. We use the notation $\mathcal{C}^{\mathrm{ob}}$ to denote however simple objects modulo condensation, since we will only discuss fusion of these objects, with the condensation defects being discussed elsewhere [48–50].

These are just the fusion rules of $\mathcal{C}_{\mathrm{Spin}(4N)}$ as there is no $\mathbb{Z}_2$ action on the involved objects. Thus, $D_2^{(\mathrm{id})}$ and $D_2^{(V)}$ are invertible surface defects which can be recognized as generating the $\mathbb{Z}_2$ center 1-form symmetry of the $\mathrm{Pin}^+(4N)$ theory.

To determine the fusion rule $D_2^{(i)} \otimes D_2^{(SC)}$ for $i \in \{\mathrm{id}, V\}$, notice that there are two simple 1-morphisms from the object $D_2^{(i)} \otimes D_2^{(SC)}$ to the object $D_2^{(SC)}$ in the 2-category $\mathcal{C}_{\mathrm{Spin}(4N)}$ and no 1-morphisms from the object $D_2^{(i)} \otimes D_2^{(SC)}$ to any other object. The two simple 1-morphisms can be recognized as the 1-morphisms

$$
\begin{aligned}
D_1^{(V,S;C)} &: \ D_2^{(V)} \otimes D_2^{(S)} \to D_2^{(C)}, \\
D_1^{(V,C;S)} &: \ D_2^{(V)} \otimes D_2^{(C)} \to D_2^{(S)},
\end{aligned}
\tag{148}
$$

for $i = V$, and the 1-morphisms

$$
\begin{aligned}
D_1^{(\mathrm{id},S;S)} &: \ D_2^{(\mathrm{id})} \otimes D_2^{(S)} \to D_2^{(S)}, \\
D_1^{(\mathrm{id},C;C)} &: \ D_2^{(\mathrm{id})} \otimes D_2^{(C)} \to D_2^{(C)},
\end{aligned}
\tag{149}
$$

for $i = \mathrm{id}$, where these 1-morphisms are in the 2-category $\mathcal{C}_{\mathrm{Spin}(4N)}$. The $\mathbb{Z}_2$ outer-automorphism acts as the exchange

$$
\begin{aligned}
D_1^{(V,S;C)} &\longleftrightarrow D_1^{(V,C;S)}, \\
D_1^{(\mathrm{id},S;S)} &\longleftrightarrow D_1^{(\mathrm{id},C;C)}.
\end{aligned}
\tag{150}
$$

Thus, there is a single simple 1-morphism for each $i$

$$
D_1^{(i,SC;SC)} : \ D_2^{(i)} \otimes D_2^{(SC)} \to D_2^{(SC)}
\tag{151}
$$

in the 2-category $\mathcal{C}_{\mathrm{Pin}^+(4N)}$, which can be expressed as

$$
\begin{aligned}
D_1^{(V,SC;SC)} &= \left( D_1^{(V,S;C)} \oplus D_1^{(V,C;S)} \right)_{\mathcal{C}_{\mathrm{Spin}(4N)}}, \\
D_1^{(\mathrm{id},SC;SC)} &= \left( D_1^{(\mathrm{id},S;S)} \oplus D_1^{(\mathrm{id},C;C)} \right)_{\mathcal{C}_{\mathrm{Spin}(4N)}}
\end{aligned}
\tag{152}
$$

1-morphisms in the category $\mathcal{C}_{\mathrm{Spin}(4N)}$, leading to the fusion rule

$$
D_2^{(i)} \otimes D_2^{(SC)} = D_2^{(SC)},
\tag{153}
$$

for $i \in \{\mathrm{id}, V\}$ in $\mathcal{C}_{\mathrm{Pin}^+(4N)}$. There is no possibility of condensations arising on the right hand side of the above equation, because there are no non-trivial lines living on $D_2^{(SC)}$ as discussed above.

Now let us discuss $D_2^{(SC)} \otimes D_2^{(SC)}$ in $\mathcal{C}_{\mathrm{Pin}^+(4N)}$. From the corresponding fusion in $\mathcal{C}_{\mathrm{Spin}(4N)}$, we see that only $D_2^{(\mathrm{id})}$ and $D_2^{(V)}$ can appear in this fusion. There are two 1-morphisms $D_2^{(SC)} \otimes D_2^{(SC)} \to D_2^{(\mathrm{id})}$ in $\mathcal{C}_{\mathrm{Spin}(4N)}$, which can be recognized as

$$
\begin{aligned}
D_1^{(S,S;\mathrm{id})} &: \ D_2^{(S)} \otimes D_2^{(S)} \to D_2^{(\mathrm{id})}, \\
D_1^{(C,C;\mathrm{id})} &: \ D_2^{(C)} \otimes D_2^{(C)} \to D_2^{(\mathrm{id})}.
\end{aligned}
\tag{154}
$$

Similarly, there are two 1-morphisms $D_2^{(SC)} \otimes D_2^{(SC)} \to D_2^{(V)}$ in $\mathcal{C}_{\mathrm{Spin}(4N)}$, which can be recognized as

$$
\begin{aligned}
D_1^{(S,C;V)} &: \ D_2^{(S)} \otimes D_2^{(C)} \to D_2^{(V)}, \\
D_1^{(C,S;V)} &: \ D_2^{(C)} \otimes D_2^{(S)} \to D_2^{(V)}.
\end{aligned}
\tag{155}
$$

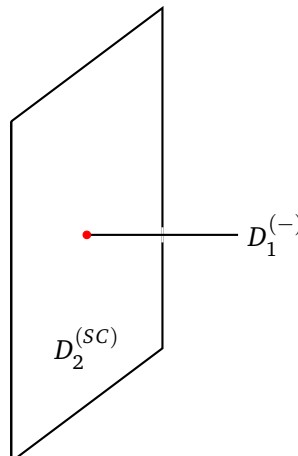

Figure 26: The defect configuration describing a 2-morphism from $D_1^{(SC)} \otimes D_1^{(SC)}$ to $D_1^{(-)}$, where $D_1^{(SC)}$ is the identity line living on the surface $D_2^{(SC)}$ shown in the figure.

These 1-morphisms are exchanged by the $\mathbb{Z}_2$ action as

$$
\begin{aligned}
D_1^{(S,S;\text{id})} &\longleftrightarrow D_1^{(C,C;\text{id})}, \\
D_1^{(C,S;V)} &\longleftrightarrow D_1^{(S,C;V)}.
\end{aligned}
\tag{156}
$$

Thus, we have single simple 1-morphisms

$$
\begin{aligned}
D_1^{(SC,SC;\text{id})} &: \ D_2^{(SC)} \otimes D_2^{(SC)} \to D_2^{(\text{id})}, \\
D_1^{(SC,SC;V)} &: \ D_2^{(SC)} \otimes D_2^{(SC)} \to D_2^{(V)}
\end{aligned}
\tag{157}
$$

in $\mathcal{C}_{\text{Pin}^+(4N)}$, and hence following the general analysis of section 4.3, we obtain that the fusion rule must take the form

$$
D_2^{(SC)} \otimes D_2^{(SC)} = \frac{D_2^{(\text{id})}}{A^{(\text{id})}} \oplus \frac{D_2^{(V)}}{A^{(V)}}
\tag{158}
$$

in $\mathcal{C}_{\text{Pin}^+(4N)}$, where we still need to determine the algebras $A^{(\text{id})}$ and $A^{(V)}$ describing the condensation/gauging on top of the surfaces $D_2^{(\text{id})}$ and $D_2^{(V)}$ respectively. The form of the above fusion rule means that $D_2^{(SC)}$ is a non-invertible surface defect, and hence the 2-category $\mathcal{C}_{\text{Pin}^+(4N)}$ describes a *non-invertible symmetry* of the $\text{Pin}^+(4N)$ theory.

To complete the description of above fusion, we need to determine local operators corresponding to 2-morphisms from $D_1^{(SC)} \otimes D_1^{(SC)}$ to lines $D_1^{(\text{id})}$, $D_1^{(-)}$, $D_1^{(V)}$ and $D_1^{(V-)}$. This is the same as the determination of local operators in the analogous 3d case we considered in the previous section. From the results obtained there, we learn that there is a single dimensional vector space of 2-morphisms from $D_1^{(SC)} \otimes D_1^{(SC)}$ to each of the lines $D_1^{(\text{id})}$, $D_1^{(-)}$, $D_1^{(V)}$ and $D_1^{(V-)}$. Let us make a side comment here in order to resolve some of the confusing statements found in previous literature: the fact that we have a 2-morphism

$$
D_1^{(SC)} \otimes D_1^{(SC)} \to D_1^{(-)},
\tag{159}
$$

means that there is a non-zero local operator lying at the intersection of the genuine line $D_1^{(-)}$ and the genuine surface $D_2^{(SC)}$, since $D_1^{(SC)}$ is the identity line on the surface $D_2^{(SC)}$. See figure 26. This looks like a configuration implying that a surface can fuse with itself to give line

defects, and could motivate one to introduce fusion rules taking two objects (i.e. surfaces) to a 1-morphism (i.e. a line). However, in the standard definitions of 2-category used in mathematics, the fusion of two objects is always an object. As we have described above, mathematically this figure is interpreted instead as a 2-morphism.

Returning back to our original problem, we have determined the algebra objects to be

$$
\begin{aligned}
A_1^{(\mathrm{id})} &= D_1^{(\mathrm{id})} \oplus D_1^{(-)}, \\
A_1^{(V)} &= D_1^{(V)} \oplus D_1^{(V_-)},
\end{aligned}
\tag{160}
$$

which means that we have to gauge the $\mathbb{Z}_2$ 0-form symmetry on $D_2^{(\mathrm{id})}$ generated by $D_1^{(-)}$ and the $\mathbb{Z}_2$ 0-form symmetry localized on $D_2^{(V)}$ generated by $D_1^{(V_-)}$. There is a unique way to perform this gauging as $H^2(B\mathbb{Z}_2, U(1)) = 0$. Consequently the morphisms comprising the algebras $A^{(\mathrm{id})}$ and $A^{(V)}$ are uniquely fixed, and the full fusion rule can be expressed as

$$
D_2^{(SC)} \otimes D_2^{(SC)} = \frac{D_2^{(\mathrm{id})}}{\mathbb{Z}_2} \oplus \frac{D_2^{(V)}}{\mathbb{Z}_2}.
\tag{161}
$$

Let us now discuss various fusion rules for lines. The fusion of 1-morphisms inside objects is straightforward to determine:

$$
\begin{aligned}
D_1^{(\mathrm{id})} \otimes_{D_2^{(\mathrm{id})}} D_1^{(-)} &= D_1^{(-)}, \\
D_1^{(-)} \otimes_{D_2^{(\mathrm{id})}} D_1^{(-)} &= D_1^{(\mathrm{id})}, \\
D_1^{(SC)} \otimes_{D_2^{(SC)}} D_1^{(SC)} &= D_1^{(SC)}, \\
D_1^{(V)} \otimes_{D_2^{(V)}} D_1^{(V)} &= D_1^{(V)}, \\
D_1^{(V)} \otimes_{D_2^{(V)}} D_1^{(V_-)} &= D_1^{(V_-)}, \\
D_1^{(V_-)} \otimes_{D_2^{(V)}} D_1^{(V_-)} &= D_1^{(V)}.
\end{aligned}
\tag{162}
$$

On the other hand, the fusion $\otimes$ of 1-morphisms is

$$
\begin{aligned}
D_1^{(\mathrm{id})} \otimes D_1^{(\mathrm{id})} &= D_1^{(\mathrm{id})}, \\
D_1^{(\mathrm{id})} \otimes D_1^{(-)} &= D_1^{(-)}, \\
D_1^{(-)} \otimes D_1^{(-)} &= D_1^{(\mathrm{id})}, \\
D_1^{(SC)} \otimes D_1^{(\mathrm{id})} &= D_1^{(SC)}, \\
D_1^{(SC)} \otimes D_1^{(-)} &= D_1^{(SC)}, \\
D_1^{(V)} \otimes D_1^{(\mathrm{id})} &= D_1^{(V)}, \\
D_1^{(V)} \otimes D_1^{(-)} &= D_1^{(V_-)}, \\
D_1^{(V_-)} \otimes D_1^{(\mathrm{id})} &= D_1^{(V_-)}, \\
D_1^{(V_-)} \otimes D_1^{(-)} &= D_1^{(V)}, \\
D_1^{(SC)} \otimes D_1^{(V)} &= D_1^{(SC)}, \\
D_1^{(SC)} \otimes D_1^{(V_-)} &= D_1^{(SC)}, \\
D_1^{(SC)} \otimes D_1^{(SC)} &= D_1^{\left(D_2^{(SC)} \otimes D_2^{(SC)}\right)},
\end{aligned}
\tag{163}
$$

where $D_1^{\left(D_2^{(SC)} \otimes D_2^{(SC)}\right)}$ denotes the identity line on the surface (161). The bimodule description of the line $D_1^{(SC)} \otimes D_1^{(SC)}$ in $D_2^{(\text{id})} \oplus D_2^{(V)}$ is provided simply by the algebra $A^{(\text{id})} \oplus A^{(V)}$ itself.

This example is of particular utility, as we have an alternate means of computing the fusions using the approach in [17]. We will do so in section 8 and find agreement with the above prescription. This reassures us to apply our approach to an example that is beyond the mixed anomaly approach, namely the $\mathbb{Z}_3$ and $S_3$ gaugings of Spin(8).

## 6.2 Pure Spin(8) ⋊ $S_3$ Gauge Theory in 4d

We consider the gauging, as in 3d, of the outer automorphism $S_3$ acting on pure Spin(8) Yang-Mills. Again we will perform this in two steps by first gauging a subgroup $\mathbb{Z}_3$.

### 6.2.1 Pure Spin(8) ⋊ $\mathbb{Z}_3$ Gauge Theory in 4d

Now consider gauging $\mathbb{Z}_3$ outer-automorphism symmetry of the 4d Spin(8) pure Yang-Mills theory. This constructs 4d Spin(8) ⋊ $\mathbb{Z}_3$ pure Yang-Mills theory.

The $\mathbb{Z}_3$ acts on $\mathcal{C}_{\text{Spin}(8)}$ as a cyclic permutation

$$D_i^{(S)} \to D_i^{(C)} \to D_i^{(V)} \to D_i^{(S)}, \tag{164}$$

for $i \in \{1, 2\}$.

Let us denote the 2-category for the Spin(8) ⋊ $\mathbb{Z}_3$ theory descending from $\mathcal{C}_{\text{Spin}(8)}$ as $\mathcal{C}_{\mathbb{Z}_3}$. Its simple objects modulo condensations are

$$\mathcal{C}_{\mathbb{Z}_3}^{\text{ob}} = \left\{ D_2^{(\text{id})}, D_2^{(SCV)} \right\}, \tag{165}$$

where

$$D_2^{(SCV)} = \left( D_2^{(S)} \oplus D_2^{(C)} \oplus D_2^{(V)} \right)_{\mathcal{C}_{\text{Spin}(8)}}, \tag{166}$$

as an object of the 2-category $\mathcal{C}_{\text{Spin}(8)}$.

Let us deduce the fusion rule $D_2^{(SCV)} \otimes D_2^{(SCV)}$. From the corresponding fusion in $\mathcal{C}_{\text{Spin}(8)}$, we see that both $D_2^{(\text{id})}$ and $D_2^{(SCV)}$ can appear in this fusion modulo condensations. There are three 1-morphisms $D_2^{(SCV)} \otimes D_2^{(SCV)} \to D_2^{(\text{id})}$ in $\mathcal{C}_{\text{Spin}(8)}$, which can be recognized as

$$\begin{aligned}
D_1^{(S,S;\text{id})} &: D_2^{(S)} \otimes D_2^{(S)} \to D_2^{(\text{id})}, \\
D_1^{(C,C;\text{id})} &: D_2^{(C)} \otimes D_2^{(C)} \to D_2^{(\text{id})}, \\
D_1^{(V,V;\text{id})} &: D_2^{(V)} \otimes D_2^{(V)} \to D_2^{(\text{id})}.
\end{aligned} \tag{167}$$

These three 1-morphisms are cyclically permuted by the $\mathbb{Z}_3$ action:

$$D_1^{(S,S;\text{id})} \to D_1^{(C,C;\text{id})} \to D_1^{(V,V;\text{id})} \to D_1^{(S,S;\text{id})}. \tag{168}$$

From this, we obtain a 1-morphism

$$D_1^{(SCV,SCV;\text{id})} : D_2^{(SCV)} \otimes D_2^{(SCV)} \to D_2^{(\text{id})} \tag{169}$$

of $\mathcal{C}_{\mathbb{Z}_3}$, which can be expressed as the following 1-morphism

$$D_1^{(SCV,SCV;\text{id})} = \left( D_1^{(S,S;\text{id})} \oplus D_1^{(C,C;\text{id})} \oplus D_1^{(V,V;\text{id})} \right)_{\mathcal{C}_{\text{Spin}(8)}} \tag{170}$$

of the 2-category $\mathcal{C}_{\text{Spin}(8)}$.

On the other hand, there are six 1-morphisms $D_2^{(SCV)} \otimes D_2^{(SCV)} \to D_2^{(SCV)}$ in $\mathcal{C}_{\text{Spin}(8)}$, which can be recognized as

$$
\begin{aligned}
D_1^{(S,C;V)} &: \ D_2^{(S)} \otimes D_2^{(C)} \to D_2^{(V)} \,, \\
D_1^{(C,V;S)} &: \ D_2^{(C)} \otimes D_2^{(V)} \to D_2^{(S)} \,, \\
D_1^{(V,S;C)} &: \ D_2^{(V)} \otimes D_2^{(S)} \to D_2^{(C)} \,, \\
D_1^{(C,S;V)} &: \ D_2^{(C)} \otimes D_2^{(S)} \to D_2^{(V)} \,, \\
D_1^{(V,C;S)} &: \ D_2^{(V)} \otimes D_2^{(C)} \to D_2^{(S)} \,, \\
D_1^{(S,V;C)} &: \ D_2^{(S)} \otimes D_2^{(V)} \to D_2^{(C)} \,.
\end{aligned}
\tag{171}
$$

These six 1-morphisms are cyclically permuted by the $\mathbb{Z}_3$ action in two orbits:

$$
\begin{aligned}
D_1^{(S,C;V)} &\to D_1^{(C,V;S)} \to D_1^{(V,S;C)} \to D_1^{(S,C;V)} \,, \\
D_1^{(C,S;V)} &\to D_1^{(V,C;S)} \to D_1^{(S,V;C)} \to D_1^{(C,S;V)} \,.
\end{aligned}
\tag{172}
$$

From this, we obtain two simple 1-morphisms

$$
D_1^{(SCV,SCV;SCV)(i)} : \ D_2^{(SCV)} \otimes D_2^{(SCV)} \to D_2^{(SCV)} \,,
\tag{173}
$$

for $i \in \{a, b\}$ of $\mathcal{C}_{\mathbb{Z}_3}$, which can be expressed as the following 1-morphisms

$$
\begin{aligned}
D_1^{(SCV,SCV;SCV)(a)} &= \left( D_1^{(S,C;V)} \oplus D_1^{(C,V;S)} \oplus D_1^{(V,S;C)} \right)_{\mathcal{C}_{\text{Spin}(8)}} \,, \\
D_1^{(SCV,SCV;SCV)(b)} &= \left( D_1^{(C,S;V)} \oplus D_1^{(V,C;S)} \oplus D_1^{(S,V;C)} \right)_{\mathcal{C}_{\text{Spin}(8)}}
\end{aligned}
\tag{174}
$$

of the 2-category $\mathcal{C}_{\text{Spin}(8)}$.

In total, we obtain using the general result in section 4.3

$$
D_2^{(SCV)} \otimes D_2^{(SCV)} = \frac{D_2^{(\text{id})}}{A^{(\text{id})}} \oplus \frac{D_2^{(SCV)}}{A^{(SCV,a)}} \oplus \frac{D_2^{(SCV)}}{A^{(SCV,b)}}
\tag{175}
$$

in the 2-category $\mathcal{C}_{\mathbb{Z}_3}$. Consequently, $D_2^{(SCV)}$ is a non-invertible surface defect, and hence the 2-category $\mathcal{C}_{\mathbb{Z}_3}$ describes a non-invertible symmetry of the $\text{Spin}(8) \rtimes \mathbb{Z}_3$ theory. We will finish the determination of the algebras $A^{(\text{id})}$, $A^{(SCV,a)}$ and $A^{(SCV,b)}$ later.

Let us now turn to a discussion of the 1-morphisms of $\mathcal{C}_{\mathbb{Z}_3}$. Since there are no non-identity simple 1-endomorphisms of the identity object in $\mathcal{C}_{\text{Spin}(8)}$, the simple 1-endomorphisms of the identity object $D_2^{(\text{id})}$ in $\mathcal{C}_{\mathbb{Z}_3}$ are

$$
\left\{ D_1^{(\text{id})}, D_1^{(\omega)}, D_1^{(\omega^2)} \right\} \,,
\tag{176}
$$

which are the Wilson line defects for the gauged $\mathbb{Z}_3$. Their fusion $\otimes$, which equals the fusion $\otimes_{D_2^{(\text{id})}}$ inside the identity object, follows the group law of $\mathbb{Z}_3$. On the other hand, there is only the identity 1-endomorphism $D_1^{(SCV)}$ of $D_2^{(SCV)}$ in $\mathcal{C}_{\mathbb{Z}_3}$, which can be expressed as

$$
D_1^{(SCV)} = \left( D_1^{(S)} \oplus D_1^{(C)} \oplus D_1^{(V)} \right)_{\mathcal{C}_{\text{Spin}(8)}} \,,
\tag{177}
$$

as a 1-morphism of $\mathcal{C}_{\text{Spin}(8)}$. Thus,

$$
\mathcal{C}_{\mathbb{Z}_3}^{\text{1-endo}} = \left\{ D_1^{(\text{id})}, D_1^{(\omega)}, D_1^{(\omega^2)}, D_1^{(SCV)} \right\}
\tag{178}
$$

is the list of simple 1-endomorphisms of simple objects in $\mathcal{C}^{\text{ob}}_{\mathbb{Z}_3}$. Since $D_1^{(SCV)}$ is the identity 1-endomorphism of $D_2^{(SCV)}$, we simply have

$$
D_1^{(SCV)} \otimes_{D_2^{(SCV)}} D_1^{(SCV)} = D_1^{(SCV)} . \tag{179}
$$

On the other hand, from computations simiar to as in the analogous 3d case, we learn that

$$
\begin{aligned}
D_1^{(SCV)} \otimes D_1^{(\omega)} &= D_1^{(SCV)} , \\
D_1^{(SCV)} \otimes D_1^{(\omega^2)} &= D_1^{(SCV)} .
\end{aligned} \tag{180}
$$

We also have a 2-morphism from $D_1^{(SCV)} \otimes D_1^{(SCV)}$ to $D_1^{(\text{id})}$, $D_1^{(\omega)}$ and $D_1^{(\omega^2)}$, and two 2-morphisms to $D_1^{(SCV)}$. One of these two 2-morphisms lives along the line $D_1^{(SCV,SCV;SCV)(a)}$, while the other lives along the line $D_1^{(SCV,SCV;SCV)(b)}$.

Thus the two algebras $A_1^{(SCV,i)}$ are trivial, whereas

$$
A_1^{(\text{id})} = D_1^{(\text{id})} \oplus D_1^{(\omega)} \oplus D_1^{(\omega^2)} . \tag{181}
$$

The above allows us to complete the fusion (175) to

$$
D_2^{(SCV)} \otimes D_2^{(SCV)} = \frac{D_2^{(\text{id})}}{\mathbb{Z}_3} \oplus 2 D_2^{(SCV)} , \tag{182}
$$

where

$$
\frac{D_2^{(\text{id})}}{\mathbb{Z}_3} \tag{183}
$$

is the condensation surface defect obtained by gauging $\mathbb{Z}_3$ 2-form symmetry of the $\text{Spin}(8) \rtimes \mathbb{Z}_3$ theory along a two-dimensional surface, and there is no gauging performed along the two $D_2^{(SCV)}$ surfaces.

### 6.2.2 Gauging $S_3$ in $\text{Spin}(8)$ Gauge Theory in 4d

To construct the 4d $\text{Spin}(8) \rtimes S_3$ pure Yang-Mills theory, we can begin with the 4d $\text{Spin}(8) \rtimes \mathbb{Z}_3$ pure Yang-Mills theory studied in the previous subsection, and gauge a $\mathbb{Z}_2$ 0-form symmetry of it. The symmetry 2-category $\mathcal{C}_{\mathbb{Z}_3}$ of the $\text{Spin}(8) \rtimes \mathbb{Z}_3$ theory descends to a symmetry 2-category $\mathcal{C}_{S_3}$ of the $\text{Spin}(8) \rtimes S_3$ theory under the gauging procedure.

The simple objects (modulo condensation) of $\mathcal{C}_{\mathbb{Z}_3}$ are left invariant by the $\mathbb{Z}_2$ action. Thus, the simple objects modulo condensation

$$
\mathcal{C}^{\text{ob}}_{S_3} = \left\{ D_2^{(\text{id})}, D_2^{(SCV)} \right\} , \tag{184}
$$

of $\mathcal{C}_{S_3}$ are the same as for $\mathcal{C}_{\mathbb{Z}_3}$.

Let us deduce the fusion rule $D_2^{(SCV)} \otimes D_2^{(SCV)}$ in $\mathcal{C}_{S_3}$. The $\mathbb{Z}_2$ action acts as

$$
D_1^{(SCV,SCV;SCV)(a)} \longleftrightarrow D_1^{(SCV,SCV;SCV)(b)} . \tag{185}
$$

Thus, there is a single 1-morphism

$$
D_1^{(SCV,SCV;SCV)} : D_2^{(SCV)} \otimes D_2^{(SCV)} \to D_2^{(SCV)} \tag{186}
$$

in $\mathcal{C}_{S_3}$, which can be expressed as

$$D_1^{(SCV,SCV;SCV)} = \left( D_1^{(SCV,SCV;SCV)(a)} \oplus D_1^{(SCV,SCV;SCV)(b)} \right)_{\mathcal{C}_{\mathbb{Z}_3}}, \tag{187}$$

as 1-morphism of $\mathcal{C}_{\mathbb{Z}_3}$. Thus, we are lead to conclude the fusion rule

$$D_2^{(SCV)} \otimes D_2^{(SCV)} = \frac{D_2^{(\mathrm{id})}}{A^{(\mathrm{id})}} \oplus \frac{D_2^{(SCV)}}{A^{(SCV)}} \tag{188}$$

in the 2-category $\mathcal{C}_{S_3}$. We will determine these algebras below.

There are three simple 1-endomorphisms $D_1^{(i)}$ for $i \in \{\mathrm{id}, \omega, \omega^2\}$ of the identity object in $\mathcal{C}_{\mathbb{Z}_3}$. Out of these, $D_1^{(\omega)}$ and $D_1^{(\omega^2)}$ are combined into a simple 1-endomorphism

$$D_1^{(\omega\omega^2)} = \left( D_1^{(\omega)} \oplus D_1^{(\omega^2)} \right)_{\mathcal{C}_{\mathbb{Z}_3}} \tag{189}$$

of the identity object of $\mathcal{C}_{S_3}$. On the other hand, the 1-endomorphism $D_1^{(\mathrm{id})}$ of $\mathcal{C}_{\mathbb{Z}_3}$ is left invariant by the $\mathbb{Z}_2$ action, so gives rise to two simple 1-endomorphisms $D_1^{(\mathrm{id})}, D_1^{(-)}$ of the identity object of $\mathcal{C}_{S_3}$. These can also be recognized as the Wilson lines created by the $\mathbb{Z}_2$ gauging. Similarly, there is only the identity 1-endomorphism $D_1^{(SCV)}$ of $D_2^{(SCV)}$ in $\mathcal{C}_{\mathbb{Z}_3}$, which is left invariant by the $\mathbb{Z}_2$ action. Consequently, we can dress $D_1^{(SCV)}$ of $\mathcal{C}_{\mathbb{Z}_3}$ by Wilson lines to obtain two simple 1-endomorphisms $D_1^{(SCV)}, D_1^{(SCV_-)}$ of the identity object of $\mathcal{C}_{S_3}$. In total,

$$\mathcal{C}_{S_3}^{\text{1-endo}} = \left\{ D_1^{(\mathrm{id})}, D_1^{(-)}, D_1^{(\omega\omega^2)}, D_1^{(SCV)}, D_1^{(SCV_-)} \right\} \tag{190}$$

describes simple 1-endomorphisms of simple objects in $\mathcal{C}_{S_3}^{\text{ob}}$.

The fusion rules for the fusion $\otimes_{D_2^{(SCV)}}$ parametrized by the object $D_2^{(SCV)}$ are

$$\begin{aligned}
D_1^{(SCV)} \otimes_{D_2^{(SCV)}} D_1^{(SCV)} &= D_1^{(SCV)}, \\
D_1^{(SCV)} \otimes_{D_2^{(SCV)}} D_1^{(SCV_-)} &= D_1^{(SCV_-)}, \\
D_1^{(SCV_-)} \otimes_{D_2^{(SCV)}} D_1^{(SCV)} &= D_1^{(SCV_-)}, \\
D_1^{(SCV_-)} \otimes_{D_2^{(SCV)}} D_1^{(SCV_-)} &= D_1^{(SCV)}.
\end{aligned} \tag{191}$$

The fusion rules for $D_1^{(\mathrm{id})}, D_1^{(-)}, D_1^{(\omega\omega^2)}$ under the fusion $\otimes_{D_2^{(\mathrm{id})}}$ parametrized by the identity object $D_2^{(\mathrm{id})}$ are the same as the fusion rules for these objects under the fusion $\otimes$ that we discuss below.

From computations similar to the ones for the analogous 3d case we considered in the previous section, we find the following fusion rules

$$\begin{aligned}
D_1^{(-)} \otimes D_1^{(\omega\omega^2)} &= D_1^{(\omega\omega^2)}, \\
D_1^{(-)} \otimes D_1^{(SCV)} &= D_1^{(SCV_-)}, \\
D_1^{(-)} \otimes D_1^{(SCV_-)} &= D_1^{(SCV)}, \\
D_1^{(\omega\omega^2)} \otimes D_1^{(SCV)} &= D_1^{(SCV)} \oplus D_1^{(SCV_-)}, \\
D_1^{(\omega\omega^2)} \otimes D_1^{(SCV_-)} &= D_1^{(SCV)} \oplus D_1^{(SCV_-)}, \\
D_1^{(\omega\omega^2)} \otimes D_1^{(\omega\omega^2)} &= D_1^{(\mathrm{id})} \oplus D_1^{(-)} \oplus D_1^{(\omega\omega^2)}.
\end{aligned} \tag{192}$$

We also have a single 2-morphism from $D_1^{(SCV)} \otimes D_1^{(SCV)}$ to each of $D_1^{(\text{id})}$, $D_1^{(\omega\omega^2)}$, $D_1^{(SCV)}$ and $D_1^{(SCV_-)}$; a single 2-morphism from $D_1^{(SCV_-)} \otimes D_1^{(SCV_-)}$ to each of $D_1^{(\text{id})}$, $D_1^{(\omega\omega^2)}$, $D_1^{(SCV)}$ and $D_1^{(SCV_-)}$; and a single 2-morphism from $D_1^{(SCV)} \otimes D_1^{(SCV_-)}$ to each of $D_1^{(-)}$, $D_1^{(\omega\omega^2)}$, $D_1^{(SCV)}$ and $D_1^{(SCV_-)}$.

The above 2-morphisms allow us to compute the algebras appearing in (188), which we can now complete as

$$D_2^{(SCV)} \otimes D_2^{(SCV)} = \frac{D_2^{(\text{id})}}{A_1^{(\text{id})} = D_1^{(\text{id})} \oplus D_1^{(\omega\omega^2)}} \oplus \frac{D_2^{(SCV)}}{\mathbb{Z}_2} \,, \tag{193}$$

where

$$\frac{D_2^{(SCV)}}{\mathbb{Z}_2} \tag{194}$$

is the defect obtained by gauging the $\mathbb{Z}_2$ localized symmetry of $D_2^{(SCV)}$ with the corresponding algebra being uniquely determined by the algebra object

$$A_1^{(SCV)} = D_1^{(SCV)} \oplus D_1^{(SCV_-)} \,. \tag{195}$$

and

$$\frac{D_2^{(\text{id})}}{A = D_1^{(\text{id})} \oplus D_1^{(\omega\omega^2)}} \tag{196}$$

is obtained from $D_2^{(\text{id})}$ by gauging the $\text{Rep}(S_3)$ localized symmetry of $D_2^{(\text{id})}$ using the algebra object

$$A_1^{(\text{id})} = D_1^{(\text{id})} \oplus D_1^{(\omega\omega^2)} \,. \tag{197}$$

We leave the precise determination of the morphisms comprising the algebra $A^{(\text{id})}$ to future work.

The fusions of lines living on $D_2^{(SCV)}$ are described in terms of bimodules of the categories describing symmetries localized on $D_2^{(\text{id})}$ and $D_2^{(SCV)}$. For the fusion $D_1^{(SCV)} \otimes D_1^{(SCV)}$, the corresponding bimodules are the algebras $A^{(\text{id})}$ and $A^{(SCV)}$ themselves, as the fused line is simply the identity line on fused surface $D_2^{(SCV)} \otimes D_2^{(SCV)}$. For the fusion $D_1^{(SCV_-)} \otimes D_1^{(SCV_-)}$, the objects underlying the bimodules are $A_1^{(\text{id})}$ and $A_1^{(SCV)}$, while we leave the determination of morphisms comprising the bimodules to future work. Finally, for the fusion $D_1^{(SCV)} \otimes D_1^{(SCV_-)}$, the objects underlying the bimodules are $D_1^{(-)} \oplus D_1^{(\omega\omega^2)}$ and $A_1^{(SCV)}$, while we leave the determination of morphisms comprising the bimodules to future work.

## 6.3 Pure $O(2)$ Gauge Theory in 4d

4d $O(2)$ pure Yang-Mills theory can be constructed from 4d $U(1)$ pure Yang-Mills theory by gauging the charge conjugation $\mathbb{Z}_2$ 0-form symmetry. The $U(1)$ theory has a

$$\Gamma^{(1)} = U(1)_e \times U(1)_m \tag{198}$$

1-form symmetry, where $U(1)_e$ is the electric 1-form symmetry acting on Wilson line defects, and $U(1)_m$ is the magnetic 1-form symmetry acting on 't Hooft line defects. The

$$\Gamma^{(0)} = \mathbb{Z}_2 \tag{199}$$

charge conjugation symmetry acts by complex conjugation on both $U(1)_e$ and $U(1)_m$. This 1-form symmetry descends to a *continuous* non-invertible 2-categorical symmetry. In what follows we ignore $U(1)_m$ and consider only the $U(1)_e$ symmetry (or vice versa). It is straightforward to extend the analysis to the full $\Gamma^{(1)}$ symmetry.

The 2-category $\mathcal{C}_{U(1)}$ associated to the 1-form symmetry $U(1)_e$ of the $U(1)$ theory has simple objects, the Gukov-Witten operators, which generate the 1-form symmetry

$$\mathcal{C}_{U(1)}^{\text{ob}} = \left\{ D_2^{(\theta)} \,\middle|\, \theta \in \mathbb{R}/\mathbb{Z} \right\}, \tag{200}$$

whose fusion follows the group law of $\mathbb{R}/\mathbb{Z}$. There is only an identity line $D_1^{(\theta)}$ on each such surface $D_2^{(\theta)}$. Thus the set of simple 1-endomorphisms of simple objects is

$$\mathcal{C}_{U(1)}^{1-\text{endo}} = \left\{ D_1^{(\theta)} \,\middle|\, \theta \in \mathbb{R}/\mathbb{Z} \right\}, \tag{201}$$

whose fusion $\otimes$ is also given by the group law of $\mathbb{R}/\mathbb{Z}$.

The $\mathbb{Z}_2$ charge conjugation leaves invariant $D_i^{(0)}$ and $D_i^{(1/2)}$, while exchanging

$$D_i^{(\theta)} \longleftrightarrow D_i^{(-\theta)}, \tag{202}$$

for $i \in \{1, 2\}$ and $\theta \neq 0, 1/2$. Gauging the $\mathbb{Z}_2$ charge conjugation leads to the $O(2)$ theory in which the 2-category $\mathcal{C}_{U(1)}$ descends to a 2-category $\mathcal{C}_{O(2)}$. The simple objects modulo condensations of $\mathcal{C}_{O(2)}$ are

$$\mathcal{C}_{O(2)}^{\text{ob}} = \left\{ D_2^{(0)}, D_2^{(1/2)}, S_2^{(\theta)} \,\middle|\, 0 < \theta < 1/2 \right\}, \tag{203}$$

where

$$S_2^{(\theta)} = \left( D_2^{(\theta)} \oplus D_2^{(-\theta)} \right)_{\mathcal{C}_{U(1)}}, \tag{204}$$

as object in $\mathcal{C}_{U(1)}$.

The simple 1-endomorphisms of simple objects in $\mathcal{C}_{O(2)}^{\text{ob}}$ are

$$\mathcal{C}_{O(2)}^{1\text{-endo}} = \left\{ D_1^{(0)}, D_1^{(-)}, D_1^{(1/2)}, D_1^{(1/2,-)}, L_1^{(\theta)} \,\middle|\, 0 < \theta < 1/2 \right\}, \tag{205}$$

where $D_1^{(0)}$, $D_1^{(1/2)}$ and $L_1^{(\theta)}$ are identity 1-endomorphisms of $D_2^{(0)}$, $D_2^{(1/2)}$ and $S_2^{(\theta)}$ respectively, and

$$L_1^{(\theta)} = \left( D_1^{(\theta)} \oplus D_1^{(-\theta)} \right)_{\mathcal{C}_{U(1)}}, \tag{206}$$

as 1-morphism in $\mathcal{C}_{U(1)}$. Since $D_1^{(0)}$ and $D_1^{(1/2)}$ have $\mathbb{Z}_2$ stabilizers, dressing them by the nontrivial $\mathbb{Z}_2$ Wilson line leads to the non-identity simple 1-endomorphisms $D_1^{(-)}$ and $D_1^{(1/2,-)}$ of $D_2^{(0)}$ and $D_2^{(1/2)}$ respectively.

The $\mathcal{C}_{O(2)}$ fusion rules of these objects can be deduced to be

$$\begin{aligned}
D_2^{(1/2)} \otimes D_2^{(1/2)} &= D_2^{(0)}, \\
D_2^{(1/2)} \otimes S_2^{(\theta)} &= S_2^{(1/2-\theta)}, \\
S_2^{(\theta)} \otimes S_2^{(\theta')} &= S_2^{\left(|\theta+\theta'|_{1/2}\right)} \oplus S_2^{\left(|\theta-\theta'|_{1/2}\right)}, \\
S_2^{(\theta)} \otimes S_2^{(1/2-\theta)} &= \frac{D_2^{(1/2)}}{\mathbb{Z}_2} \oplus S_2^{\left(|2\theta-1/2|_{1/2}\right)}; \quad \theta \neq 1/4, \\
S_2^{(\theta)} \otimes S_2^{(\theta)} &= \frac{D_2^{(0)}}{\mathbb{Z}_2} \oplus S_2^{\left(|2\theta|_{1/2}\right)}; \quad \theta \neq 1/4, \\
S_2^{(1/4)} \otimes S_2^{(1/4)} &= \frac{D_2^{(0)}}{\mathbb{Z}_2} \oplus \frac{D_2^{(1/2)}}{\mathbb{Z}_2},
\end{aligned} \tag{207}$$

where $\theta' \neq 1/2 - \theta$ and $\theta' \neq \theta$, and $|\alpha|_{1/2}$ is defined in (127) and (128). The fusion $\otimes$ of 1-morphisms are straightforward.

A closely related symmetry category is that of $\widetilde{SU}(N)$ Gauge Theory in 4d, which we discuss in appendix A.2.

## 6.4 $(\mathbb{Z}_2 \times \mathbb{Z}_2) \rtimes \mathbb{Z}_2$ Gauge Theory in 4d

Our general procedure is equally applicable to discrete gauge theories, for which we now discuss a 4d example. Consider two copies of the $\mathbb{Z}_2$ gauge theory in 4d. We denote the various topological defects in the two copies by $L$ and $R$ labels respectively. The action takes the form

$$S = i\pi \sum_{I=L,R} \int_{M_4} b^I \cup \delta a^I \,, \tag{208}$$

where $a^I \in C^1(M_4, \mathbb{Z}_2)$ and $b^I \in C^2(M_4, \mathbb{Z}_2)$. The model has topological ('t-Hooft) surface and (Wilson) line operators that generate 1-form and 2-form global symmetries respectively. The 1-form symmetry group is

$$\Gamma^{(1)} = \mathbb{Z}_2^L \times \mathbb{Z}_2^R \,, \tag{209}$$

generated by the topological surface operators $\exp\left\{i\pi \int_\Sigma b^I\right\}$. Similarly, the 2-form global symmetry is also

$$\Gamma^{(2)} = \mathbb{Z}_2^L \times Z_2^R \,, \tag{210}$$

generated by the topological line operators $\exp\left\{i\pi \int_\gamma a^I\right\}$. The data of topological operators can be recast as a fusion 2-category $\mathcal{C}_{\mathbb{Z}_2 \times \mathbb{Z}_2}$. The simple objects of the category are the topological surface operators

$$\mathcal{C}_{\mathbb{Z}_2 \times \mathbb{Z}_2}^{\text{ob}} = \left\{D_2^{(\text{id})}, D_2^{(L)}, D_2^{(R)}, D_2^{(LR)}\right\} \,. \tag{211}$$

The fusion of surfaces in $\mathcal{C}_{\mathbb{Z}_2 \times \mathbb{Z}_2}$ is read off from the group composition in $\mathbb{Z}_2^L \times \mathbb{Z}_2^R$, i.e,

$$D_2^{(g)} \otimes D_2^{(h)} = D_2^{(gh)} \,, \tag{212}$$

where $g, h \in \{\text{id}, L, R, LR\} = \mathbb{Z}_2^L \times \mathbb{Z}_2^R$. The endomorphism space of each of the simple surfaces is isomorphic as a set to the topological line operators in the theory. We denote the lines on a surface $D_2^{(g)}$ by the label $D_1^{(g),(h)}$. The set of 1-endomorphisms corresponding to a surface $D_2^{(g)}$ is

$$\mathcal{C}_{(g),\mathbb{Z}_2 \times \mathbb{Z}_2}^{\text{1-endo}} = \left\{D_1^{(g),(h)} \mid h \in \mathbb{Z}_2 \times \mathbb{Z}_2\right\} \,. \tag{213}$$

The fusion rules of lines inherit the group structure i.e

$$D_1^{(g_1),(h_1)} \otimes D_1^{(g_2),(h_2)} = D_1^{(g_1 g_2),(h_1 h_2)} \,. \tag{214}$$

Similarly, the fusion of lines within a surface is given by

$$D_1^{(g),(h_1)} \otimes_{D_2^{(g)}} D_1^{(g),(h_2)} = D_1^{(g),(h_1 h_2)} \,. \tag{215}$$

Furthermore each of the simple lines have a 1-dimensional endomorphism space associated to which are the 2-morphisms of the symmetry category $\mathcal{C}_{\mathbb{Z}_2 \times \mathbb{Z}_2}$.

$$\mathcal{C}_{\mathbb{Z}_2 \times \mathbb{Z}_2}^{\text{2-morph}} = \left\{D_0^{(g),(h)} \mid g, h \in \mathbb{Z}_2 \times \mathbb{Z}_2\right\} \,. \tag{216}$$

The 2-morphisms satisfy the fusion structure

$$
\begin{aligned}
D_0^{(g_1),(h_1)} \otimes D_0^{(g_2),(h_2)} &= D_0^{(g_1 g_2),(h_1 h_2)}, \\
D_0^{(g),(h_1)} \otimes_{D_2^{(g)}} D_0^{(g),(h_2)} &= D_0^{(g),(h_1 h_2)}.
\end{aligned}
\tag{217}
$$

Finally, the fusion structure of 2-morphisms within lines is trivial. The theory has a 0-form symmetry

$$
\Gamma^{(0)} = \mathbb{Z}_2,
\tag{218}
$$

which acts by exchanging $L \leftrightarrow R$. We are interested in the symmetry category $\mathcal{C}_{(\mathbb{Z}_2 \times \mathbb{Z}_2)/\Gamma^{(0)}}$ that arises upon gauging this 0-form global symmetry. More precisely, using the procedure developed in the previous sections, we can access the untwisted or identity flux sector which forms a sub-category $\mathcal{C}^{(\mathrm{id})}_{(\mathbb{Z}_2 \times \mathbb{Z}_2)/\Gamma^{(0)}} \subset \mathcal{C}_{(\mathbb{Z}_2 \times \mathbb{Z}_2)/\Gamma^{(0)}}$. Firstly, the objects of the gauged category are $\Gamma^{(0)}$ orbits within $\mathcal{C}^{\mathrm{ob}}_{\mathbb{Z}_2 \times \mathbb{Z}_2}$. More precisely

$$
\mathcal{C}^{(\mathrm{id}),\mathrm{ob}}_{(\mathbb{Z}_2 \times \mathbb{Z}_2)/\Gamma^{(0)}} = \left\{ D_2^{(\mathrm{id})}, D_2^{(L,R)}, D_2^{(LR)} \right\},
\tag{219}
$$

where

$$
D_2^{(L,R)} = \left( D_2^L \bigoplus D_2^R \right)_{\mathcal{C}_{\mathbb{Z}_2 \times \mathbb{Z}_2}},
\tag{220}
$$

as an object in the pre-gauged symmetry category $\mathcal{C}_{\mathbb{Z}_2 \times \mathbb{Z}_2}$. Next, we compute the fusion rules of surfaces. Firstly, we have

$$
\begin{aligned}
D_2^{(\mathrm{id})} \otimes D_2^{(\mathrm{id})} &= D_2^{(\mathrm{id})}, \\
D_2^{(\mathrm{id})} \otimes D_2^{(LR)} &= D_2^{(LR)}, \\
D_2^{(LR)} \otimes D_2^{(LR)} &= D_2^{(\mathrm{id})},
\end{aligned}
\tag{221}
$$

which are obtained from the fusion rules in $\mathcal{C}_{\mathbb{Z}_2 \times \mathbb{Z}_2}$ as each of the objects involved are $\Gamma^{(0)}$ invariant. The fusion rules of the $D_2^{(L,R)}$ surface with $D_2^{(i)}$ where $i \in \{\mathrm{id}, LR\}$, can be computed by lifting the surfaces to the pre-gauged category and restricting to $\Gamma^{(0)}$ invariant 1-morphisms. For instance, in the pre-gauged symmetry category $\mathcal{C}_{\mathbb{Z}_2 \times \mathbb{Z}_2}$ there are two 1-morphisms from the object $D_2^{(\mathrm{id})} \otimes D_2^{(L,R)}$ to $D_2^{(L,R)}$ and no 1-morphisms to any other object

$$
\begin{aligned}
D_1^{(\mathrm{id}),(L);(L)} &: D_2^{(\mathrm{id})} \otimes D_2^{(L)} \longrightarrow D_2^{(L)}, \\
D_1^{(\mathrm{id}),(R);(R)} &: D_2^{(\mathrm{id})} \otimes D_2^{(R)} \longrightarrow D_2^{(R)}.
\end{aligned}
\tag{222}
$$

Similarly, in $\mathcal{C}_{\mathbb{Z}_2 \times \mathbb{Z}_2}$, there are two simple 1-morphisms from $D_2^{(LR)} \otimes D_2^{(L,R)}$ to $D_2^{(L,R)}$ and no 1-morphisms to any other object

$$
\begin{aligned}
D_1^{(LR),(L);(R)} &: D_2^{(LR)} \otimes D_2^{(L)} \longrightarrow D_2^{(R)}, \\
D_1^{(LR),(R);(L)} &: D_2^{(LR)} \otimes D_2^{(R)} \longrightarrow D_2^{(L)}.
\end{aligned}
\tag{223}
$$

These two 1-morphisms in each set are exchanged under the action of $\Gamma^{(0)}$

$$
\begin{aligned}
D_1^{(\mathrm{id}),(L);(L)} &\longleftrightarrow D_1^{(\mathrm{id}),(R);(R)}, \\
D_1^{(LR),(L);(R)} &\longleftrightarrow D_1^{(LR),(R);(L)},
\end{aligned}
\tag{224}
$$

and therefore form a single simple 1-morphism in $\mathcal{C}^{(\mathrm{id})}_{(\mathbb{Z}_2 \times \mathbb{Z}_2)/\Gamma^{(0)}}$

$$
\begin{aligned}
D_1^{(\mathrm{id}),(L,R),(L,R)} &: D_2^{(\mathrm{id})} \otimes D_2^{(L,R)} \longrightarrow D_2^{(L,R)}, \\
D_1^{(LR),(L,R),(L,R)} &: D_2^{(LR)} \otimes D_2^{(L,R)} \longrightarrow D_2^{(L,R)}.
\end{aligned}
\tag{225}
$$

Furthermore the fusion of the identity line on $D_2^{(L,R)}$ with the identity line on $D_2^{(i)}$ gives

$$
D_1^{(L,R),(\mathrm{id})} \otimes D_1^{(i),(\mathrm{id})} = D_1^{(L,R),(\mathrm{id})},
\tag{226}
$$

hence there is no additional condensation in this fusion process. Next, the object $D_2^{(L,R)} \otimes D_2^{(L,R)}$, in the category $\mathcal{C}_{\mathbb{Z}_2 \times \mathbb{Z}_2}$, has two simple 1-morphisms each to the surfaces $D_2^{(\mathrm{id})}$ and $D_2^{(LR)}$. These are

$$
\begin{aligned}
D_1^{(L),(L);(\mathrm{id})} &: D_2^{(L)} \otimes D_2^{(L)} \longrightarrow D_2^{(\mathrm{id})}, \\
D_1^{(R),(R);(\mathrm{id})} &: D_2^{(R)} \otimes D_2^{(R)} \longrightarrow D_2^{(\mathrm{id})}, \\
D_1^{(R),(L);(LR)} &: D_2^{(R)} \otimes D_2^{(R)} \longrightarrow D_2^{(\mathrm{id})}, \\
D_1^{(L),(R);(LR)} &: D_2^{(L)} \otimes D_2^{(R)} \longrightarrow D_2^{(LR)}.
\end{aligned}
\tag{227}
$$

These 1-morphisms form two orbits under the action of $\Gamma^{(0)}$

$$
\begin{aligned}
D_1^{(L),(L);(\mathrm{id})} &\longleftrightarrow D_1^{(R),(R);(\mathrm{id})}, \\
D_1^{(L),(R);(LR)} &\longleftrightarrow D_1^{(R),(L);(LR)}.
\end{aligned}
\tag{228}
$$

The $\Gamma^{(0)}$ invariant orbits therefore become simple 1-morphisms in the symmetry category $\mathcal{C}_{(\mathbb{Z}_2 \times \mathbb{Z}_2)/\Gamma^{(0)}}$. There can be an additional condensation on the $D_2^{\mathrm{id}}$ and $D_2^{(LR)}$ surface, depending on the fusion of identity lines $D_1^{(L,R),(\mathrm{id})} \otimes D_1^{(L,R),(\mathrm{id})}$. Let us denote the algebras corresponding to these (potential) condensations as $A^{(\mathrm{id})}$ and $A^{(LR)}$. To summarize, the fusion of topological surface operators in the identity flux sector of the gauged theory is

$$
\begin{aligned}
D_2^{(\mathrm{id})} \otimes D_2^{(LR)} &= D_2^{(LR)}, \\
D_2^{(LR)} \otimes D_2^{(LR)} &= D_2^{(\mathrm{id})}, \\
D_2^{(\mathrm{id})} \otimes D_2^{(L,R)} &= D_2^{(L,R)}, \\
D_2^{(LR)} \otimes D_2^{(L,R)} &= D_2^{(L,R)}, \\
D_2^{(L,R)} \otimes D_2^{(L,R)} &= \frac{D_2^{(\mathrm{id})}}{A^{(\mathrm{id})}} \oplus \frac{D_2^{(LR)}}{A^{(LR)}}.
\end{aligned}
\tag{229}
$$

Next, we move onto the topological lines, i.e., 1-morphisms in the gauged sub-category $\mathcal{C}^{(\mathrm{id})}_{(\mathbb{Z}_2 \times \mathbb{Z}_2)/\Gamma^{(0)}}$. Firstly the identity line $D_1^{(\mathrm{id}),(\mathrm{id})}$ in $\mathcal{C}_{\mathbb{Z}_2 \times \mathbb{Z}_2}$ splits into two lines denoted as $D_1^{(\mathrm{id}),(\mathrm{id})}$ and $D_1^{(\mathrm{id}),(-)}$ where the latter is the non-trivial $\mathbb{Z}_2$ Wilson line and carries the sign representation of $\mathbb{Z}_2$. Importantly, this line generates a non-anomalous 2-form global symmetry which is dual to $\Gamma^{(0)}$. This 2-form $\mathbb{Z}_2$ symmetry can be gauged by proliferating the topological line $D_1^{(\mathrm{id}),(-)}$ in order to recover the original symmetry category $\mathcal{C}_{\mathbb{Z}_2 \times \mathbb{Z}_2}$. Similarly all the other $\Gamma^{(0)}$ invariant lines in $\mathcal{C}_{\mathbb{Z}_2 \times \mathbb{Z}_2}$ also split into two lines each, labelled by representations of $\Gamma^{(0)}$ such that the non-trivial representation, denoted with a minus sign, is obtained by dressing the original line operator in $\mathcal{C}_{\mathbb{Z}_2 \times \mathbb{Z}_2}$ with the $\mathbb{Z}_2$ Wilson line $D_1^{(\mathrm{id}),(-)}$. The remaining lines in the

$\mathcal{C}_{\mathbb{Z}_2 \times \mathbb{Z}_2}$, i.e., those that transform under $\Gamma^{(0)}$, combine into orbits that become simple lines in $\mathcal{C}^{(\mathrm{id})}_{(\mathbb{Z}_2 \times \mathbb{Z}_2)/\Gamma^{(0)}}$. It is straightforward to enumerate the lines in $\mathcal{C}^{(\mathrm{id})}_{(\mathbb{Z}_2 \times \mathbb{Z}_2)/\Gamma^{(0)}}$. The endomorphism lines in the surface $D_2^{(i)}$, where $i \in \{\mathrm{id}, LR\}$ are

$$\mathcal{C}^{\text{1-endo},(i)}_{(\mathbb{Z}_2 \times \mathbb{Z}_2)/\Gamma^{(0)}} = \left\{ D_1^{(i),(\mathrm{id})}, D_1^{(i),(-)}, D_1^{(i),(LR)}, D_1^{(i),(LR,-)}, D_1^{(i),(L,R)} \right\}, \tag{230}$$

where

$$D_1^{(i),(L,R)} = \left( D_1^{(i),(L)} \oplus D_1^{(i),(R)} \right)_{\mathcal{C}_{\mathbb{Z}_2 \times \mathbb{Z}_2}}. \tag{231}$$

Similarly the 1-endomorphisms within the surface $D_2^{(L,R)}$ are

$$\mathcal{C}^{\text{1-endo},(L,R)}_{(\mathbb{Z}_2 \times \mathbb{Z}_2)/\Gamma^{(0)}} = \left\{ D_1^{(L,R),(\mathrm{id})}, D_1^{(L,R),(LR)}, D_1^{(L,R),(L,R;1)}, D_1^{(L,R),(L,R;2)} \right\}, \tag{232}$$

where

$$\begin{aligned} D_1^{(L,R),(\mathrm{id})} &= \left( D_1^{(L),(\mathrm{id})} \oplus D_1^{(R),(\mathrm{id})} \right)_{\mathcal{C}_{\mathbb{Z}_2 \times \mathbb{Z}_2}}, \\ D_1^{(L,R),(LR)} &= \left( D_1^{(L),(LR)} \oplus D_1^{(R),(LR)} \right)_{\mathcal{C}_{\mathbb{Z}_2 \times \mathbb{Z}_2}}, \\ D_1^{(L,R),(L,R;1)} &= \left( D_1^{(L),(L)} \oplus D_1^{(R),(R)} \right)_{\mathcal{C}_{\mathbb{Z}_2 \times \mathbb{Z}_2}}, \\ D_1^{(L,R),(L,R;2)} &= \left( D_1^{(L),(R)} \oplus D_1^{(R),(L)} \right)_{\mathcal{C}_{\mathbb{Z}_2 \times \mathbb{Z}_2}}. \end{aligned} \tag{233}$$

Next, we move onto the fusion of lines within surfaces. Firstly, within the surface $D_2^{(i)}$ where $i \in \{\mathrm{id}, LR\}$, the fusion rules are

$$\begin{aligned} D_1^{(i),(\mathrm{id})} \otimes_{D_2^{(i)}} D_1^{(i),(-)} &= D_1^{(i),(-)}, \\ D_1^{(i),(LR)} \otimes_{D_2^{(i)}} D_1^{(i),(-)} &= D_1^{(i),(LR,-)}, \\ D_1^{(i),(LR)} \otimes_{D_2^{(i)}} D_1^{(i),(LR,-)} &= D_1^{(i),(-)}, \\ D_1^{(i),(LR,-)} \otimes_{D_2^{(i)}} D_1^{(i),(LR,-)} &= D_1^{(i),(\mathrm{id})}, \\ D_1^{(i),(\mathrm{id})} \otimes_{D_2^{(i)}} D_1^{(i),(L,R)} &= D_1^{(i),(L,R)}, \\ D_1^{(i),(-)} \otimes_{D_2^{(i)}} D_1^{(i),(L,R)} &= D_1^{(i),(L,R)}, \\ D_1^{(i),(LR)} \otimes_{D_2^{(i)}} D_1^{(i),(L,R)} &= D_1^{(i),(L,R)}, \\ D_1^{(i),(LR,-)} \otimes_{D_2^{(i)}} D_1^{(i),(L,R)} &= D_1^{(i),(L,R)}, \\ D_1^{(i),(L,R)} \otimes_{D_2^{(i)}} D_1^{(i),(L,R)} &= D_1^{(i),(\mathrm{id})} \oplus D_1^{(i),(-)} \oplus D_1^{(i),(LR)} \oplus D_1^{(i),(LR,-)}. \end{aligned} \tag{234}$$

Notice that the fusion 1-category of lines within the defect $D_2^{(i)}$ is isomorphic to the Tambara-Yamagami fusion-category $TY(\mathbb{Z}_2 \times \mathbb{Z}_2)$. The fusion of lines on distinct surfaces $D_2^i$ and $D_2^j$ with $j \in \{\mathrm{id}, LR\}$ is almost the same as in (234) except we need to account for the group

composition of the surface labels, i.e.,

$$
\begin{aligned}
D_1^{(i),(\mathrm{id})} \otimes D_1^{(j),(-)} &= D_1^{(ij),(-)}, \\
D_1^{(i),(LR)} \otimes D_1^{(j),(-)} &= D_1^{(ij),(LR,-)}, \\
D_1^{(i),(LR)} \otimes D_1^{(j),(LR,-)} &= D_1^{(ij),(-)}, \\
D_1^{(i),(LR,-)} \otimes D_1^{(j),(LR,-)} &= D_1^{(ij),(\mathrm{id})}, \\
D_1^{(i),(\mathrm{id})} \otimes D_1^{(j),(L,R)} &= D_1^{(ij),(L,R)}, \\
D_1^{(i),(-)} \otimes D_1^{(j),(L,R)} &= D_1^{(ij),(L,R)}, \\
D_1^{(i),(LR)} \otimes D_1^{(j),(L,R)} &= D_1^{(ij),(L,R)}, \\
D_1^{(i),(LR,-)} \otimes D_1^{(j),(L,R)} &= D_1^{(ij),(L,R)}, \\
D_1^{(i),(L,R)} \otimes D_1^{(j),(L,R)} &= D_1^{(ij),(\mathrm{id})} \oplus D_1^{(ij),(-)} \oplus D_1^{(ij),(LR)} \oplus D_1^{(ij),(LR,-)}.
\end{aligned}
\tag{235}
$$

In summary the lines $D_1^{(i),(\alpha)}$ with $\alpha = \mathrm{id}, -, LR$ or $LR, -$ are invertible lines while $D_1^{(i),(L,R)}$ is a non-invertible line and together these lines form the Tambara-Yamagami fusion-category $TY(\mathbb{Z}_2 \times \mathbb{Z}_2)$. Next, the fusion among lines in the surface $D_2^{(L,R)}$ and with other lines can be computed by lifting to the pre-gauged category and finding $\Gamma^{(0)}$ invariant morphisms. First consider the fusion rules between lines in the $D_2^{(L,R)}$ surface and lines in $D_2^i$ with $i \in \{\mathrm{id}, LR\}$. Fusion of invertible lines $D_1^{(i),(\alpha)}$ with $D_1^{(L,R),(\mathrm{id})}$ is

$$
\begin{aligned}
D_1^{(i),(\mathrm{id})} \otimes D_1^{(L,R),(\mathrm{id})} &= D_1^{(L,R),(\mathrm{id})}, \\
D_1^{(i),(-)} \otimes D_1^{(L,R),(\mathrm{id})} &= D_1^{(L,R),(\mathrm{id})}, \\
D_1^{(i),(LR)} \otimes D_1^{(L,R),(\mathrm{id})} &= D_1^{(L,R),(LR)}, \\
D_1^{(i),(LR,-)} \otimes D_1^{(L,R),(\mathrm{id})} &= D_1^{(L,R),(LR)}.
\end{aligned}
\tag{236}
$$

The fusion of the non-invertible line $D_1^{(i),(L,R)}$ with $D_1^{(L,R),(\mathrm{id})}$ is computed using the lift to the pre-gauged category as

$$
\begin{aligned}
D_1^{(i),(L,R)} \otimes D_1^{(L,R),(\mathrm{id})} &= \left( D_1^{(i),(L)} \oplus D_1^{(i),(R)} \right) \otimes \left( D_1^{(L),(\mathrm{id})} \oplus D_1^{(R),(\mathrm{id})} \right) \Big|_{\mathcal{C}_{\mathbb{Z}_2 \times \mathbb{Z}_2}} \\
&= \left( D_1^{(L),(L)} \oplus D_1^{(R),(R)} \oplus D_1^{(L),(R)} \oplus D_1^{(R),(L)} \right) \Big|_{\mathcal{C}_{\mathbb{Z}_2 \times \mathbb{Z}_2}} \\
&= D_1^{(L,R),(L,R;1)} \oplus D_1^{(L,R),(L,R;2)}.
\end{aligned}
\tag{237}
$$

Similarly, the fusion rules involving the invertible lines $D_1^{(i),(\alpha)}$ on $D_2^i$ with the line $D_1^{(L,R),(LR)}$ on $D_2^{(L,R)}$ are

$$
\begin{aligned}
D_1^{(i),(\mathrm{id})} \otimes D_1^{(L,R),(LR)} &= D_1^{(L,R),(LR)}, \\
D_1^{(i),(-)} \otimes D_1^{(L,R),(LR)} &= D_1^{(L,R),(LR)}, \\
D_1^{(i),(LR)} \otimes D_1^{(L,R),(LR)} &= D_1^{(L,R),(\mathrm{id})}, \\
D_1^{(i),(LR,-)} \otimes D_1^{(L,R),(LR)} &= D_1^{(L,R),(\mathrm{id})}.
\end{aligned}
\tag{238}
$$

While the fusion of the non-invertible line $D_1^{(i),(L,R)}$ with $D_1^{(L,R),(LR)}$ is computed as

$$
\begin{aligned}
D_1^{(i),(L,R)} \otimes D_1^{(L,R),(LR)} &= \left( D_1^{(i),(L)} \oplus D_1^{(i),(R)} \right) \otimes \left( D_1^{(L),(LR)} \oplus D_1^{(R),(LR)} \right) \Big|_{\mathcal{C}_{\mathbb{Z}_2 \times \mathbb{Z}_2}} \\
&= \left( D_1^{(L),(L)} \oplus D_1^{(R),(R)} \oplus D_1^{(L),(R)} \oplus D_1^{(R),(L)} \right) \Big|_{\mathcal{C}_{\mathbb{Z}_2 \times \mathbb{Z}_2}} \\
&= D_1^{(L,R),(L,R;1)} \oplus D_1^{(L,R),(L,R;2)} .
\end{aligned}
\tag{239}
$$

The fusion of the invertible lines $D_1^{(i),(\alpha)}$ on $D_2^i$ with the line $D_1^{(L,R),(L,R;\sigma)}$ with $\sigma = 1, 2$ on $D_2^{(L,R)}$ are

$$
\begin{aligned}
D_1^{(\text{id}),(\text{id})} \otimes D_1^{(L,R),(L,R;\sigma)} &= D_1^{(L,R),(L,R;\sigma)} , \\
D_1^{(\text{id}),(-)} \otimes D_1^{(L,R),(L,R;\sigma)} &= D_1^{(L,R),(L,R;\sigma)} , \\
D_1^{(\text{id}),(LR)} \otimes D_1^{(L,R),(L,R;\sigma)} &= D_1^{(L,R),(\sigma+1 \bmod 2)} , \\
D_1^{(\text{id}),(LR,-)} \otimes D_1^{(L,R),(L,R;\sigma)} &= D_1^{(L,R),(\sigma+1 \bmod 2)} , \\
D_1^{(LR),(\text{id})} \otimes D_1^{(L,R),(L,R;\sigma)} &= D_1^{(L,R),(L,R;\sigma+1 \bmod 2)} , \\
D_1^{(LR),(-)} \otimes D_1^{(L,R),(L,R;\sigma)} &= D_1^{(L,R),(L,R;\sigma+1 \bmod 2)} , \\
D_1^{(LR),(LR)} \otimes D_1^{(L,R),(L,R;\sigma)} &= D_1^{(L,R),(\sigma)} , \\
D_1^{(LR),(LR,-)} \otimes D_1^{(L,R),(L,R;\sigma)} &= D_1^{(L,R),(\sigma)} ,
\end{aligned}
\tag{240}
$$

and the fusion of the non-invertible line $D_1^{(i),(L,R)}$ with $D_1^{(L,R),(L,R;\sigma)}$ is

$$
\begin{aligned}
D_1^{(\text{id}),(L,R)} \otimes D_1^{(L,R),(L,R;\sigma)} &= D_1^{(L,R),(\text{id})} \oplus D_1^{(L,R),(LR)} , \\
D_1^{(LR),(L,R)} \otimes D_1^{(L,R),(L,R;\sigma)} &= D_1^{(L,R),(\text{id})} \oplus D_1^{(L,R),(LR)} .
\end{aligned}
\tag{241}
$$

Next, we are left with computing the fusion rules of lines within the $D_2^{(L,R)}$ surface. In particular, the fusion $D_1^{(L,R),(\text{id})} \otimes D_1^{(L,R),(\text{id})}$ has an important physical consequence—it directly encodes the algebra objects $A^{(\text{id})}$ and $A^{(LR)}$ that condense on the fusion outcome of $D_2^{(L,R)} \otimes D_2^{(L,R)}$

$$
D_1^{(L,R),(\text{id})} \otimes D_1^{(L,R),(\text{id})} = A^{(\text{id})} \oplus A^{LR} .
\tag{242}
$$

In the category $\mathcal{C}_{\mathbb{Z}_2 \times \mathbb{Z}_2}$, there is a two dimensional morphism space between $D_1^{(L,R),(\text{id})}$ and the lines $D_1^{(\text{id}),(\text{id})}$ and $D_1^{(LR),(\text{id})}$ and no other morphisms to any other lines. The morphism space decomposes into the two representations of $\Gamma^{(0)}$ in $\mathcal{C}_{(\mathbb{Z}_2 \times \mathbb{Z}_2)/\Gamma^{(0)}}$ and therefore one needs to attach the $\mathbb{Z}_2$ Wilson line to the non-trivial morphism to make it $\Gamma^{(0)}$ invariant. Consequently, the algebra objects can be read off to be

$$
\begin{aligned}
A_1^{(\text{id})} &= D_1^{(\text{id}),(\text{id})} \oplus D_1^{(\text{id}),(-)} , \\
A_1^{(LR)} &= D_1^{(LR),(\text{id})} \oplus D_1^{(LR),(-)} .
\end{aligned}
\tag{243}
$$

Finally, the fusion rules among the remaining lines in the $D_2^{(L,R)}$ surface can be computed. Since the fusion outcome of $D_2^{(L,R)} \otimes D_2^{(L,R)}$ involves extra condensations (see (229)), the fusion of lines on $D_2^{(L,R)}$ are described as algebra bimodules on defects before condensations (see section

3)

$$D_1^{(L,R),(LR)} \otimes D_1^{(L,R),(LR)} = B^{(D_2^{(\mathrm{id})}/A^{(\mathrm{id})}),(\mathrm{id};LR;LR)} \oplus B^{(D_2^{(LR)}/A^{(LR)}),(\mathrm{id};LR;LR)} ,$$

$$D_1^{(L,R),(\mathrm{id})} \otimes D_1^{(L,R),(LR)} = B^{(D_2^{(\mathrm{id})}/A^{(\mathrm{id})}),(LR;\mathrm{id};LR)} \oplus B^{(D_2^{(LR)}/A^{(LR)}),(LR;\mathrm{id};LR)} ,$$

$$D_1^{(L,R),(\mathrm{id})} \otimes D_1^{(L,R),(L,R,1)} = B^{(D_2^{(\mathrm{id})}/A^{(\mathrm{id})}),(L,R;\mathrm{id};L,R,1)} \oplus B^{(D_2^{(LR)}/A^{(LR)}),(L,R;\mathrm{id};L,R;1)}$$

$$D_1^{(L,R),(\mathrm{id})} \otimes D_1^{(L,R),(L,R,2)} = B^{(D_2^{(\mathrm{id})}/A^{(\mathrm{id})}),(L,R;\mathrm{id};L,R,2)} \oplus B^{(D_2^{(LR)}/A^{(LR)}),(L,R;\mathrm{id};L,R,2)} ,$$

$$D_1^{(L,R),(LR)} \otimes D_1^{(L,R),(L,R,1)} = B^{(D_2^{(\mathrm{id})}/A^{(\mathrm{id})}),(L,R;LR;L,R,1)} \oplus B^{(D_2^{(LR)}/A^{(LR)}),(L,R;LR;L,R,1)} , \qquad (244)$$

$$D_1^{(L,R),(LR)} \otimes D_1^{(L,R),(L,R,2)} = B^{(D_2^{(\mathrm{id})}/A^{(\mathrm{id})}),(L,R;LR;L,R,2)} \oplus B^{(D_2^{(LR)}/A^{(LR)}),(L,R;LR;L,R,2)} ,$$

$$D_1^{(L,R),(L,R,1)} \otimes D_1^{(L,R),(L,R,1)} = B^{(D_2^{(\mathrm{id})}/A^{(\mathrm{id})}),(\mathrm{id};L,R,1;L,R,1)} \oplus B^{(D_2^{(LR)}/A^{(LR)}),(LR;L,R1;L,R,1)} ,$$

$$D_1^{(L,R),(L,R,1)} \otimes D_1^{(L,R),(L,R,2)} = B^{(D_2^{(\mathrm{id})}/A^{(\mathrm{id})}),(\mathrm{id};L,R,1;L,R,2)} \oplus B^{(D_2^{(LR)}/A^{(LR)}),(LR;L,R1;L,R,2)} ,$$

$$D_1^{(L,R),(L,R,2)} \otimes D_1^{(L,R),(L,R,2)} = B^{(D_2^{(\mathrm{id})}/A^{(\mathrm{id})}),(\mathrm{id};L,R,2;L,R,2)} \oplus B^{(D_2^{(LR)}/A^{(LR)}),(LR;L,R,2;L,R,2)} .$$

The bimodules with a direct sum of invertible lines as objects are

$$B_1^{(D_2^{(\mathrm{id})}/A^{(\mathrm{id})}),(\mathrm{id};LR;LR)} = D_1^{(\mathrm{id})(\mathrm{id})} \oplus D_1^{(\mathrm{id})(-)} ,$$

$$B_1^{(D_2^{(LR)}/A^{(LR)}),(\mathrm{id};LR;LR)} = D_1^{(LR)(\mathrm{id})} \oplus D_1^{(LR)(-)} ,$$

$$B_1^{(D_2^{(\mathrm{id})}/A^{(\mathrm{id})}),(LR;\mathrm{id};LR)} = D_1^{(\mathrm{id})(LR)} \oplus D_1^{(\mathrm{id})(LR,-)} ,$$

$$B_1^{(D_2^{(LR)}/A^{(LR)}),(LR;\mathrm{id};LR)} = D_1^{(LR)(LR)} \oplus D_1^{(LR)(LR,-)} ,$$

$$B_1^{(D_2^{(\mathrm{id})}/A^{(\mathrm{id})}),(\mathrm{id};L,R,1;L,R,1)} = D_1^{(\mathrm{id})(\mathrm{id})} \oplus D_1^{(\mathrm{id})(-)} ,$$

$$B_1^{(D_2^{(LR)}/A^{(LR)}),(LR;L,R,1;L,R,1)} = D_1^{(LR)(LR)} \oplus D_1^{(LR)(LR,-)} , \qquad (245)$$

$$B_1^{(D_2^{(\mathrm{id})}/A^{(\mathrm{id})}),(\mathrm{id};L,R,1;L,R,2)} = D_1^{(\mathrm{id})(\mathrm{id})} \oplus D_1^{(\mathrm{id})(-)} ,$$

$$B_1^{(D_2^{(LR)}/A^{(LR)}),(LR;L,R,1;L,R,2)} = D_1^{(LR)(LR)} \oplus D_1^{(LR)(LR,-)} ,$$

$$B_1^{(D_2^{(\mathrm{id})}/A^{(\mathrm{id})}),(\mathrm{id};L,R,2;L,R,2)} = D_1^{(\mathrm{id})(\mathrm{id})} \oplus D_1^{(\mathrm{id})(-)} ,$$

$$B_1^{(D_2^{(LR)}/A^{(LR)}),(LR;L,R,2;L,R,2)} = D_1^{(LR)(LR)} \oplus D_1^{(LR)(LR,-)} .$$

While the remaining bimodules have non-invertible lines as objects. These are

$$B_1^{(D_2^{(\mathrm{id})}/A^{(\mathrm{id})}),(L,R;\mathrm{id};L,R,1)} = D_1^{(\mathrm{id})(L,R)} ,$$

$$B_1^{(D_2^{(LR)}/A^{(LR)}),(L,R;\mathrm{id};L,R;1)} = D_1^{(LR)(L,R)} ,$$

$$B_1^{(D_2^{(\mathrm{id})}/A^{(\mathrm{id})}),(L,R;\mathrm{id};L,R,2)} = D_1^{(\mathrm{id})(L,R)} ,$$

$$B_1^{(D_2^{(LR)}/A^{(LR)}),(L,R;\mathrm{id};L,R,2)} = D_1^{(LR)(L,R)} ,$$

$$B_1^{(D_2^{(\mathrm{id})}/A^{(\mathrm{id})}),(L,R;LR;L,R,1)} = D_1^{(\mathrm{id})(L,R)} , \qquad (246)$$

$$B_1^{(D_2^{(LR)}/A^{(LR)}),(L,R;LR;L,R,1)} = D_1^{(LR)(L,R)} ,$$

$$B_1^{(D_2^{(\mathrm{id})}/A^{(\mathrm{id})}),(L,R;LR;L,R,2)} = D_1^{(\mathrm{id})(L,R)} ,$$

$$B_1^{(D_2^{(LR)}/A^{(LR)}),(L,R;LR;L,R,2)} = D_1^{(LR)(L,R)} .$$

We leave the computation of the bimodule morphisms for future work.

# 7 Examples: Non-invertible 3-Categorical Symmetries in 5d and 6d

In this section, we discuss examples of UV complete 5d and 6d theories that carry non-invertible 3-categorical symmetries. So far we considered non-supersymmetric theories, however in 5d and 6d, the natural class of theories are supersymmetric.

The higher-category gauging is applicable again in cases where there is also a description in terms of higher-groups/mixed anomalies, and we will give a comparison in section 8.

## 7.1 5d $\mathcal{N} = 2$ $\mathrm{Pin}^+(4N)$ Super Yang-Mills Theory

It is clear from the example of $\mathrm{Pin}^+(4N)$ Yang-Mills Theory in the previous two sections that this theory contains a non-invertible $(d-2)$-categorical symmetry in $d$ spacetime dimensions. However it is not UV complete on its own in $d \geqslant 5$. But in $d = 5$, its analogue with 16 supercharges, namely the 5d $\mathcal{N} = 2$ $\mathrm{Pin}^+(4N)$ super Yang-Mills theory, is a 5d KK theory, i.e. it UV completes to a 6d SCFT compactified on a circle.

We construct it by gauging the $\mathbb{Z}_2$ outer automorphism symmetry of 5d $\mathcal{N} = 2$ $\mathrm{Spin}(4N)$ super Yang-Mills, which has a 3-category $\mathcal{C}_{\mathrm{Spin}(4N)}$ describing invertible $\mathbb{Z}_2 \times \mathbb{Z}_2$ 1-form symmetry. The key elements of the 3-category are

$$\left\{ D_i^{(\mathrm{id})}, D_i^{(S)}, D_i^{(C)}, D_i^{(V)} \right\} , \tag{247}$$

for $i \in \{1, 2, 3\}$. The elements $D_3^{(i)}$ are simple objects of $\mathcal{C}_{\mathrm{Spin}(4N)}$, $D_2^{(i)}$ are simple 1-endomorphisms of $D_3^{(i)}$, and $D_1^{(i)}$ are simple 2-endomorphisms of $D_2^{(i)}$. These fusion $\otimes$ on these elements follows the $\mathbb{Z}_2 \times \mathbb{Z}_2$ group law as in previous sections. The non-trivial part of the action of $\mathbb{Z}_2$ is the exchange of $D_i^{(S)}$ and $D_i^{(C)}$.

$\mathcal{C}_{\mathrm{Spin}(4N)}$ descends to a 3-category $\mathcal{C}_{\mathrm{Pin}^+(4N)}$ describing non-invertible symmetries in the $\mathrm{Pin}^+(4N)$ super Yang-Mills theory. We can easily determine key data of $\mathcal{C}_{\mathrm{Pin}^+(4N)}$ to be as follows. The simple objects modulo condensations of $\mathcal{C}_{\mathrm{Pin}^+(4N)}$ are

$$\mathcal{C}_{\mathrm{Pin}^+(4N)}^{\mathrm{ob}} = \left\{ D_3^{(\mathrm{id})}, D_3^{(SC)}, D_3^{(V)} \right\} , \tag{248}$$

where

$$D_3^{(SC)} = \left( D_3^{(S)} \oplus D_3^{(C)} \right)_{\mathcal{C}_{\mathrm{Spin}(4N)}} , \tag{249}$$

as an object of the 3-category $\mathcal{C}_{\mathrm{Spin}(4N)}$.

The fusion rules of these objects can be deduced to be

$$\begin{aligned}
D_3^{(\mathrm{id})} \otimes D_3^{(V)} &= D_3^{(V)} , \\
D_3^{(V)} \otimes D_3^{(V)} &= D_3^{(\mathrm{id})} , \\
D_3^{(\mathrm{id})} \otimes D_3^{(SC)} &= D_3^{(SC)} , \\
D_3^{(V)} \otimes D_3^{(SC)} &= D_3^{(SC)} , \\
D_3^{(SC)} \otimes D_3^{(SC)} &= \frac{D_3^{(\mathrm{id})}}{A^{(\mathrm{id})}} \oplus \frac{D_3^{(V)}}{A^{(V)}} ,
\end{aligned} \tag{250}$$

for some yet to be determined 2-algebras $A^{(\mathrm{id})}$ and $A^{(V)}$.

The simple 1-endomorphisms of simple objects in $\mathcal{C}_{\mathrm{Pin}^+(4N)}^{\mathrm{ob}}$ are

$$\mathcal{C}_{\mathrm{Pin}^+(4N)}^{\text{1-endo}} = \left\{ D_2^{(\mathrm{id})}, D_2^{(SC)}, D_2^{(V)} \right\} , \tag{251}$$

where each $D_2^{(i)}$ is the identity 1-endomorphism of $D_3^{(i)}$.

The simple 2-endomorphisms of simple 1-endomorphisms $D_2^{(i)}$ of $\mathcal{C}_{\mathrm{Pin}^+(4N)}$ are

$$\mathcal{C}_{\mathrm{Pin}^+(4N)}^{\text{2-endo}} = \left\{ D_1^{(\mathrm{id})}, D_1^{(-)}, D_1^{(SC)}, D_1^{(V)}, D_1^{(V_-)} \right\}, \tag{252}$$

where $D_2^{(i)}$ has identity 2-endomorphism $D_1^{(i)}$, and $D_1^{(-)}$, $D_1^{(V_-)}$ are non-identity 2-endomorphisms of $D_2^{(\mathrm{id})}$, $D_2^{(V)}$ respectively, arising due to dressings by $\mathbb{Z}_2$ Wilson lines.

The lines $D_1^{(-)}$ and $D_1^{(V_-)}$ can end on the 3-surface $D_3^{(SC)}$ leading to the conclusion that its fusion rule with itself is

$$D_3^{(SC)} \otimes D_3^{(SC)} = \frac{D_3^{(\mathrm{id})}}{\mathbb{Z}_2^{(1)}} \oplus \frac{D_3^{(V)}}{\mathbb{Z}_2^{(1)}}, \tag{253}$$

where

$$\frac{D_3^{(i)}}{\mathbb{Z}_2^{(1)}}, \tag{254}$$

for $i \in \{\mathrm{id}, V\}$ is a 3-dimensional topological defect obtained by gauging the $\mathbb{Z}_2$ 1-form symmetry of the 3-dimensional defect $D_3^{(i)}$. For $D_3^{(\mathrm{id})}$ the $\mathbb{Z}_2$ 1-form symmetry is generated by $D_1^{(\mathrm{id})}, D_1^{(-)}$, and for $D_3^{(V)}$ the $\mathbb{Z}_2$ 1-form symmetry is generated by $D_1^{(V)}, D_1^{(V_-)}$.

The other fusion rules are straightforward to determine.

## 7.2 Absolute 6d $\mathcal{N} = (2,0)$ SCFT of Type $\left[ SO(2n) \times SO(2n) \right] \rtimes \mathbb{Z}_2$

Relative 6d $\mathcal{N} = (2,0)$ SCFTs are known to be classified by simple A,D,E Lie algebras. Here relative means that the theory contains mutually non-local defects if one tries to define them as purely 6d theories. The locality is restored if one realizes the 6d theory as the boundary condition of a non-invertible 7d TQFT. On the other hand, an absolute 6d theory is one that can be defined as a purely 6d theory without encountering mutually non-local defects.

Consider 6d $\mathcal{N} = (2,0)$ SCFT based on Lie algebra $D_n$, which we refer to as 6d $\mathcal{N} = (2,0)$ SCFT of type $\mathrm{Spin}(2n)$, as it contains topological dimension-3 defects whose fusion is described by the group law of the center of $\mathrm{Spin}(2n)$. This is a relative theory, but can be made absolute by choosing a topological boundary condition for the attached 7d TQFT. We refer to the resulting absolute theory as 6d $\mathcal{N} = (2,0)$ SCFT of type $SO(2n)$, as it contains topological dimension-3 defects whose fusion is described by the group law of the center of $SO(2n)$, which is $\mathbb{Z}_2$. In other words, the 6d $\mathcal{N} = (2,0)$ SCFT of type $SO(2n)$ contains a

$$\Gamma^{(2)} = \mathbb{Z}_2 \tag{255}$$

2-form symmetry.

Now stack together two 6d $\mathcal{N} = (2,0)$ SCFTs of $SO(2n)$ type to obtain a 6d $\mathcal{N} = (2,0)$ SCFT of type $SO(2n) \times SO(2n)$ which has a

$$\Gamma^{(2)} = \mathbb{Z}_2 \times \mathbb{Z}_2 \tag{256}$$

2-form symmetry. This theory also has a

$$\Gamma^{(0)} = \mathbb{Z}_2 \tag{257}$$

0-form symmetry, which acts by exchanging the two $SO(2n)$ theories. Thus, it acts on $\Gamma^{(2)}$ by exchanging the two $\mathbb{Z}_2$ 2-form symmetries. The category describing $\Gamma_2$ is a 3-category which is isomorphic to the 3-category $\mathcal{C}_{\mathrm{Spin}(4n)}$ we discussed in the previous subsection.

Gauging the $\mathbb{Z}_2$ 0-form symmetry, we are lead to a 6d $\mathcal{N} = (2,0)$ SCFT of type

$$\left[SO(2n) \times SO(2n)\right] \rtimes \mathbb{Z}_2 \,, \tag{258}$$

which has a non-invertible symmetry described by a 3-category descending from $\mathcal{C}_{\mathrm{Spin}(4n)}$. This is precisely the $\mathcal{C}_{\mathrm{Pin}^+(4n)}$ category we discussed in the previous subsection, but now $\mathcal{C}_{\mathrm{Pin}^+(4n)}$ describes topological defects of a 6d theory.

# 8 Non-Invertibles and Fusion from Higher-Groups/Anomalies

## 8.1 Non-Invertibles from Higher-Groups via Gauging

In [17], a construction was presented that takes as an input a $d$-dimensional quantum field theory $\mathfrak{T}$ with a certain type of mixed anomaly, i.e. one that is linear in the background field $A_{p+1}$ corresponding to a $p$-form global symmetry $\Gamma^{(p)}$, and produces as an output a new quantum field theory $\mathfrak{T}'$ which contains $(d - p - 1)$-dimensional non-invertible defects. The descendent theory $\mathfrak{T}'$ is obtained by gauging some part of the symmetry structure of $\mathfrak{T}$ that is contained in the complement of $\Gamma^{(p)}$ and also appears manifestly in the anomaly action. Crucially, the anomaly, by definition, poses an obstruction to gauging that is alleviated by locally modifying the $\Gamma^{(p)}$ defect. In fact, the local modification is what causes the $\Gamma^{(p)}$ defect to become non-invertible in $\mathfrak{T}'$.

Concretely, let the symmetry structure of $\mathfrak{T}$ be a product of higher-form groups $\mathbb{G}_{\mathfrak{T}} = \prod_{a=0}^{d-2} \Gamma^{(a)}$. Some of the factors $\Gamma^{(a)}$ could be trivial.

It is more convenient to formulate everything in terms of background fields $A_{a+1}$ in terms of which the anomaly action is given by

$$\mathcal{A} = \int_{N_{d+1}} A_{p+1} \cup \xi(A_{p+1}^{\mathsf{c}}) \,, \quad A_{p+1}^{\mathsf{c}} = \{A_{a+1}\}_{a \neq p} \,, \tag{259}$$

where

$$\xi \in H^{d-p}\left(\mathbb{G}_{\mathsf{T}}/\Gamma^{(p)}, \Gamma_{\mathrm{dual}}^{(p)}\right) \,, \quad \xi(A_{p+1}^{\mathsf{c}}) \in H^{d-p}\left(N_{d+1}, \Gamma_{\mathrm{dual}}^{(p)}\right) \,, \tag{260}$$

where $\Gamma_{\mathrm{dual}}^{(p)} := \mathrm{hom}(\Gamma^{(p)}, \mathbb{R}/2\pi\mathbb{Z})$. $N_{d+1}$ is an auxiliary $d+1$-manifold, used to define the anomaly in (259), whose boundary is the $d$-manifold where $\mathfrak{T}$ lives. Since $\mathbb{G}_{\mathfrak{T}}$ is a product of higher groups, the quotient by $\Gamma^{(p)}$ should be understood more technically as taking the quotient on the classifying space $B\mathbb{G}$ which is a Cartesian product of $B^{a+1}\Gamma^{(a)}$. We suppress such technicalities since they make the presentation heavy without adding much content. Let the defect corresponding to $\mathrm{g} \in \Gamma^{(p)}$ be denoted as $D_{\mathrm{g}}$. If $D_{\mathrm{g}}$ is wrapped along a $(d - p - 1)$-dimensional sub-manifold $\Sigma_{d-p-1}$ of $M_d$, then we denote it as $D_{\mathrm{g}}(\Sigma_{d-p-1})$. Due to the anomaly, such a defect carries a non-trivial dependence on the background $A_{p+1}^{\mathsf{c}}$ which cannot be localized on $\Sigma_{d-p-1}$. Now consider gauging some subgroup of $\mathbb{G}_{\mathfrak{T}}/\Gamma^{(p)}$ on which $\xi$ depends. Doing so, we obtain a gauged theory $\mathfrak{T}'$ in which the defect $D_{\mathrm{g}}$ becomes ill-defined due to an anomaly. More precisely, it has a dependence on dynamical fields that cannot be localized on $\Sigma_{d-p-1}$. This situation can be remedied by a local modification to $D_{\mathrm{g}}$, which involves adding a topological field theory $\mathcal{X}_{\mathrm{g}}$ with a $\mathbb{G}_{\mathfrak{T}}/\Gamma^{(p)}$ 't-Hooft anomaly $\xi$. The defects in the gauged theory correspondingly are modified as

$$D_{\mathrm{g}} \longmapsto \mathcal{N}_{\mathrm{g}} = D_{\mathrm{g}} \mathcal{X}_{\mathrm{g}} \,. \tag{261}$$

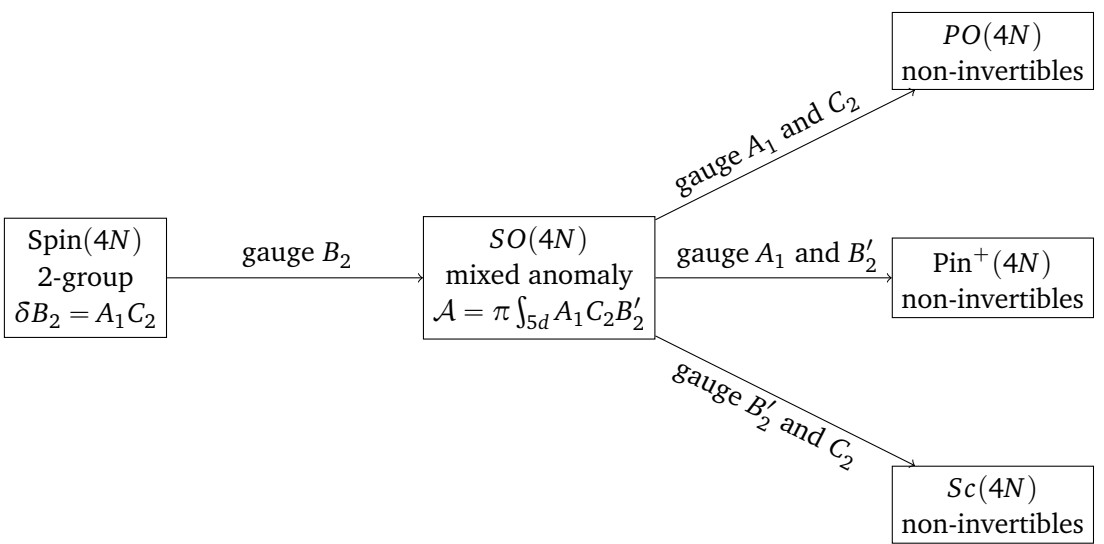

Figure 27: Overview of the theories with non-invertible symmetries that we can construct from gauging the 2-group in the $\mathrm{Spin}(4N)$ theory in 4d.

Notably any such theory $\mathfrak{T}$ with anomaly (259) can, in turn, be obtained from a theory $\mathfrak{T}_0$ with a non-anomalous higher-group symmetry $\mathbb{G}$, which sits in the short exact sequence

$$1 \longrightarrow B^{d-p-1}\Gamma^{(p)}_{\mathrm{dual}} \longrightarrow B\mathbb{G} \longrightarrow B\mathbb{G}_{\mathfrak{T}}/B\Gamma^{(p)} \longrightarrow 1\,, \tag{262}$$

with an extension class $\xi$. The symmetry structure of $\mathfrak{T}$ is obtained from the symmetry structure of $\mathfrak{T}_0$ by gauging $\Gamma^{(p)}_{\mathrm{dual}}$ in $\mathfrak{T}_0$. In summary, the non-invertibles discussed in [17] can be obtained by starting from a higher group and gauging in two steps.

## 8.2 Non-Invertibles from 2-Groups in Pure 4d $\mathfrak{so}(4N)$ Yang-Mills

Let us consider pure $\mathrm{Spin}(4N)$ gauge theory and for concreteness let us work in 4d. The theory has a $\Gamma^{(1)} = \mathbb{Z}_2^{(1),B} \times \mathbb{Z}_2^{(1),C}$ 1-form symmetry and a $\Gamma^{(0)} = \mathbb{Z}_2^{(0)}$ outer-automorphism 0-form symmetry. The two symmetries combine into a 2-group, which in terms of the background gauge fields reads

$$\delta B_2 = A_1 C_2\,, \tag{263}$$

where $B_2, C_2$ are the backgrounds for the two 1-form symmetry factors and $A_1$ is the background for the 0-form symmetry.

This 2-group is equivalent to the $\mathbb{Z}_2^{(0)}$ outer-automorphism action exchanging two $\mathbb{Z}_2^{(1)}$ subgroups of the 1-form symmetry, as can be understood pictorially in the following way (see figure 28). We denote the subgroups of $\Gamma^{(1)}$ which are exchanged by the $\mathbb{Z}_2^{(0)}$ action by $\mathbb{Z}_2^{(1),S}$ and $\mathbb{Z}_2^{(1),C}$, while $\mathbb{Z}_2^{(1),B}$ denotes the diagonal subgroup.[6] In terms of symmetry defects, the action is the following: if a topological surface defect $D_2^{(C)}$ associated to $\mathbb{Z}_2^{(1),C}$ crosses the codimension-1 defect $D_3^{(-)}$ for $\mathbb{Z}_2^{(0)}$, it emerges as the defect $D_2^{(S)}$ associated to $\mathbb{Z}_2^{(1),S}$. Since, following the $\Gamma^{(1)}$ group law, we have that $D_2^{(S)} = D_2^{(B)} \otimes D_2^{(C)}$, we can re-interpret the above action in this way: upon passing $D_2^{(C)}$ through a codimension-1 defect for $\mathbb{Z}_2^{(0)}$, we create a codimension-3 junction from which the defect $D_2^{(B)}$ associated to the $\mathbb{Z}_2^{(1),B}$ subgroup is emitted. The 2-group (263) states precisely this: at the junction of $D_3^{(-)}$ (on a 3-cycle Poincaré dual to

---

[6]Notice that this was denoted in the previous sections as $\mathbb{Z}_2^{(1),V}$.

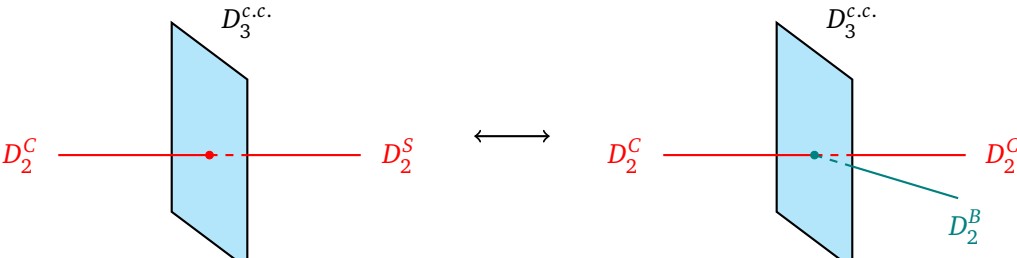

Figure 28: Left: exchange action of $\mathbb{Z}_2^{(0)}$ on the 1-form symmetry defects $D_2^{(S)}$ and $D_2^{(C)}$. Right: equivalently, the 1-form symmetry defect $D_2^{(B)}$ associated to the diagonal subgroup is emitted at the junction of $D_3^{(-)}$ and $D_2^{(C)}$. This represents pictorially the 2-group $\delta B_2 = A_1 C_2$.

$A_1$) and $D_2^{(C)}$ (on a 2-cycle Poincaré dual to $C_2$), there is a flux for $\mathbb{Z}_2^{(1),B}$ meaning that a non-trivial background $B_2$ is sourced.

We will show that by gauging various combination of the symmetries appearing in the 2-group (263) we can go to different theories that have non-invertible symmetries:

1. $PO(4N)$ theory:
   gauge $B_2, C_2, A_1$: we obtain a codimension-2 non-invertible defect;

2. $\text{Pin}^+(4N)$ theory:
   gauge $A_1$: we obtain a codimension-2 non-invertible defect;

3. $Sc(4N)$ theory:
   gauge $C_2$: we obtain a codimension-1 non-invertible defect.

A way of deriving this result is to first gauge the $\mathbb{Z}_2^{(1),B}$ subgroup of the 1-form symmetry to go to $SO(4N)$ gauge theory by promoting $B_2$ to a dynamical field $b_2$ (see figure 27). The $SO(4N)$ theory has an emergent dual 1-form symmetry $\mathbb{Z}_2^{(1),B'}$ (in 4d), whose background we denote by $B_2'$ and which couples as $\pi \int_{M_4} b_2 B_2'$. Due to the relation (263), this coupling is ill-defined, as it has a bulk dependency

$$\mathcal{A} = \pi \int_{M_5} \delta b_2 B_2' = \pi \int_{M_5} A_1 C_2 B_2'. \tag{264}$$

This results in a mixed 't Hooft anomaly for the $SO(4N)$ theory. Using this map from 2-groups to mixed 't Hooft anomalies, the fusion rules can then by derived by following the approach of [17], as we review in appendix B.

Before writing explicitly the fusion algebra in the theories mentioned above, we summarize the non-invertible defects that we obtain

- $\mathcal{N}(M_2; B_2')$: non-invertible defect in $PO(4N)$, corresponding to the codimension-2 defect generating $\mathbb{Z}_2^{(1),B'}$;

- $\mathcal{N}(M_2; C_2)$: non-invertible defect in $\text{Pin}^+(4N)$, corresponding to the codimension-2 defect generating $\mathbb{Z}_2^{(1),C}$;

- $\mathcal{N}(M_3; A_1)$: non-invertible defect in $Sc(4N)$, corresponding to the codimension-1 defect generating $\mathbb{Z}_2^{(0)}$.

### 8.2.1 Fusion Rules: $\text{Pin}^+(4N)$

This can be obtained by gauging $B_2'$ and $A_1$ in the $SO(4N)$ theory. Gauging $\mathbb{Z}_2^{(1),B'}$ we recover $\text{Spin}(4N)$, and gauging $A_1$ we obtain $\text{Pin}^+(4N)$. Therefore the overall effect of these gaugings is to gauge charge conjugation in $\text{Spin}(4N)$ theory.

The fusion algebra that we find is

$$
\begin{aligned}
\mathcal{N}(M_2; C_2) \times \mathcal{N}(M_2; C_2) &= \frac{1 + T(M_2)}{|H^0(M_2, \mathbb{Z}_2)|} \sum_{M_1 \in H_1(M_2, \mathbb{Z}_2)} L(M_1), \\
\mathcal{N}(M_2; C_2) \times T(M_2) &= \mathcal{N}(M_2; C_2), \\
\mathcal{N}(M_2; C_2) \times L(M_1) &= \mathcal{N}(M_2; C_2).
\end{aligned}
\tag{265}
$$

Here $T(M_2) = e^{i\pi \oint_{M_2} b_2'}$ is the defect generating the $\mathbb{Z}_2^{(1),B}$ 1-form symmetry and $L(M_1) = e^{i\pi \oint_{M_1} a_1}$ is the defect generating the $\mathbb{Z}_2^{(2)}$ 2-form symmetry dual to $\mathbb{Z}_2^{(0)}$.

This is precisely the theory which we studied in section 6.1 using the higher-categorical approach. In particular

$$
\begin{aligned}
D_2^{(SC)} &\longleftrightarrow \mathcal{N}(M_2; C_2), \\
D_2^{(V)} &\longleftrightarrow T(M_2), \\
D_1^{(-)} &\longleftrightarrow L(M_1),
\end{aligned}
\tag{266}
$$

and the identification of the identity surface and lines with $D_2^{(\text{id})}$ and $D_1^{(\text{id})}$, respectively. Note that in (265) we use $\times$ and not $\otimes$ as in section 6.1 to distinguish between the somewhat "mixed" fusion algebra, between objects of various dimensions and the 'proper' fusion algebra, in the higher category, that involves only objects and morphisms of the same dimension.

Notice also that the right hand side of the fusion $\mathcal{N}(M_2; C_2) \times \mathcal{N}(M_2; C_2)$ is precisely

$$
\frac{D_2^{(\text{id})}}{\mathbb{Z}_2}(M_2) \oplus \frac{D_2^{(V)}}{\mathbb{Z}_2}(M_2),
\tag{267}
$$

as we found using our approach in section 6.1.

For the sake of clarity we provide the details for this theory now.

**Gauging of $B_2'$ and $A_1$.** We gauge $B_2'$ and $A_1$ and expect the codimension-two defect implementing the $\mathbb{Z}_2^{(1),C}$ symmetry, which we denote as $D(M_2)$, to become non-invertible. Indeed, in the presence of the anomaly (264), only the following combination is invariant under background gauge transformations of $B_2'$ and $A_1$

$$
D(M_2) e^{i\pi \int_{M_3} A_1 B_2'}.
\tag{268}
$$

This implies that when we gauge $\mathbb{Z}_2^{(1),B'}$ and $Z_2^{(0)}$ and promote $B_2'$ and $A_1$ to dynamical fields $b_2'$ and $a_1$, we must couple $D(M_2)$ to an appropriate TQFT which absorbs the bulk dependency. We conjecture that in this case the TQFT we need is simply a 2d BF coupling, and we define the 2d defect (we will in following not include the background field in the labeling of $\mathcal{N}$ for simplicity)

$$
\mathcal{N}(M_2) \propto \int \mathcal{D}\phi_0 \mathcal{D}\gamma_1 \, D(M_2, b_2', a_1) \, e^{i\pi \int_{M_2} -\phi_0 \delta\gamma_1 + \phi_0 b_2' + \gamma_1 a_1},
\tag{269}
$$

where $\phi_0 \in C^0(M_2, \mathbb{Z}_2)$ is a 0-form field and $\gamma_1 \in C^1(M_2, \mathbb{Z}_2)$ is a 1-form field. The first term in the exponential is the BF coupling, while the other two are couplings between the TQFT fields and the bulk dynamical fields $b_2'$ and $a_1$. Using the $\phi_0$ equation of motions $\delta\gamma_1 = b_2'$, the variation of the exponential precisely gives $b_2' a_1$.

**Fusion Algebra.** We will now show that the defects $\mathcal{N}(M_2)$ satisfy a fusion algebra, and are non-invertible. First note that

$$\mathcal{N}(M_2) \times \mathcal{N}(M_2) \propto \int \mathcal{D}\phi_0 \mathcal{D}\gamma_1 \mathcal{D}\tilde{\phi}_0 \mathcal{D}\tilde{\gamma}_1 e^{i\pi \int_{M_2} (\phi_0 - \tilde{\phi}_0) b_2' + (\gamma_1 - \tilde{\gamma}_1) a_1 - \phi_0 \delta\gamma_1 + \tilde{\phi}_0 \delta\tilde{\gamma}_1}, \qquad (270)$$

where we used $D^2(M_2) = 1$, since it satisfies the $\mathbb{Z}_2$ fusion rules. We can shift variables $\phi_0 - \tilde{\phi}_0 = \hat{\phi}_0$ and $\gamma_1 - \tilde{\gamma}_1 = \hat{\gamma}_1$ to obtain the expression

$$\mathcal{N}(M_2) \times \mathcal{N}(M_2) \propto \int \mathcal{D}\phi_0 \mathcal{D}\gamma_1 \mathcal{D}\hat{\phi}_0 \mathcal{D}\hat{\gamma}_1 e^{i\pi \int_{M_2} \hat{\phi}_0 b_2' + \hat{\gamma}_1 a_1 - \phi_0 \delta\hat{\gamma}_1 - \hat{\phi}_0 \delta\gamma_1 + \hat{\phi}_0 \delta\hat{\gamma}_1}. \qquad (271)$$

Integrating out $\phi_0$ and $\gamma_1$ in the above expression sets $\delta\hat{\gamma}_1 = \delta\hat{\phi}_0 = 0$, so the term $\int_{M_2} \hat{\phi}_0 \delta\hat{\gamma}_1$ is actually trivial. Then we are left with

$$\mathcal{N}(M_2) \times \mathcal{N}(M_2) \propto \int \mathcal{D}\hat{\phi}_0 \mathcal{D}\hat{\gamma}_1 e^{i\pi \int_{M_2} \hat{\phi}_0 b_2' + \hat{\gamma}_1 a_1}. \qquad (272)$$

We can rewrite the above equation in discrete notation as

$$\mathcal{N}(M_2) \times \mathcal{N}(M_2) \propto \sum_{\hat{\phi}_0 \in H^0(M_2, \mathbb{Z}_2)} e^{i\pi \int_{M_2} \hat{\phi}_0 b_2'} \sum_{\hat{\gamma}_1 \in H^1(M_2, \mathbb{Z}_2)} e^{i\pi \int_{M_2} \hat{\gamma}_1 a_1}, \qquad (273)$$

which using $\int_{M_2} \hat{\gamma}_1 a_1 = \oint_{M_1} a_1$, where $M_1 \in H_1(M_2, \mathbb{Z}_2)$ is the Poincaré dual of $\hat{\gamma}_1$, reads

$$\mathcal{N}(M_2) \times \mathcal{N}(M_2) \propto (1 + e^{i\pi \oint_{M_2} b_2'}) \sum_{M_1 \in H_1(M_2, \mathbb{Z}_2)} e^{i\pi \oint_{M_1} a_1}. \qquad (274)$$

Here $e^{i\pi \oint_{M_2} b_2'} = T(M_2)$ is the codimension-2 defect generating the 1-form symmetry dual to $\mathbb{Z}_2^{(1),B'}$ and $e^{i\pi \oint_{M_1} a_1} = L(M_1)$ is the codimension-3 defect generating the 2-form symmetry dual to $\mathbb{Z}_2^{(0)}$. Hence we see that

$$\mathcal{N}(M_2) \times \mathcal{N}(M_2) = \frac{1 + T(M_2)}{|H^0(M_2, \mathbb{Z}_2)|} \sum_{M_1 \in H_1(M_2, \mathbb{Z}_2)} L(M_1), \qquad (275)$$

and so $\mathcal{N}(M_2)$ is a non-invertible defect. The normalization can be fixed with considerations similars to those in [17].

We can compute also the fusion rules of $\mathcal{N}(M_2)$ with the other operators in the theory. First we compute the fusion rule between $\mathcal{N}(M_2)$ and the surface operator $W(M_2)$. This is given by

$$\mathcal{N}(M_2) \times T(M_2) = \int \mathcal{D}\phi_0 \mathcal{D}\gamma_1 D(M_2, b_2', a_1) e^{i\pi \int_{M_2} -\phi_0 \delta\gamma_1 + \phi_0 b_2' + \gamma_1 a_1 + i\pi \oint_{M_2} b_2'} = \mathcal{N}(M_2). \qquad (276)$$

The fusion rule between $\mathcal{N}(M_2)$ and $L(M_1)$ is instead given by

$$\mathcal{N}(M_2) \times L(M_1) = \int \mathcal{D}\phi_0 \mathcal{D}\gamma_1 e^{i\pi \int_{M_2} -\phi_0 \delta\gamma_1 + \phi_0 b_2' + \gamma_1 a_1 + \oint_{M_1} a_1}$$

$$= \int \mathcal{D}\phi_0 \mathcal{D}\gamma_1 e^{i\pi \int_{M_2} -\phi_0 \delta\gamma_1 + \phi_0 b_2' + \gamma_1 a_1 + \int_{M_2} l_1 a_1}$$

$$= \int \mathcal{D}\phi_0 \mathcal{D}\gamma_1 e^{i\pi \int_{M_2} -\phi_0 \delta\gamma_1 + \phi_0 b_2' + (\gamma_1 + l_1) a_1} = \mathcal{N}(M_2), \qquad (277)$$

where $l_1 \in H^1(M_2, \mathbb{Z}_2)$ is the Poincaré dual of $M_1 \in H_1(M_2, \mathbb{Z}_2)$. In the last line we are free to shift $\gamma_1 \to \gamma_1 + l_1$. The first term in the exponential does not give an additional contribution since $\delta l_1 = 0$. Hence we recover exactly $\mathcal{N}(M_2)$.

### 8.2.2 Fusion Rules: $PO(4N)$

The fusion algebra is computed similarly, see also appendix B, and is given by

$$
\begin{aligned}
\mathcal{N}(M_2; B_2') \times \mathcal{N}(M_2; B_2') &= \frac{1 + W(M_2)}{|H^0(M_2, \mathbb{Z}_2)|} \sum_{M_1 \in H_1(M_2, \mathbb{Z}_2)} L(M_1)\,, \\
\mathcal{N}(M_2; B_2') \times W(M_2) &= \mathcal{N}(M_2; B_2')\,, \\
\mathcal{N}(M_2; B_2') \times L(M_1) &= \mathcal{N}(M_2; B_2')\,.
\end{aligned}
\tag{278}
$$

Here $L(M_1) = e^{i\pi \oint_{M_1} a_1}$ is the defect for the $\mathbb{Z}_2^{(2)}$ 2-form symmetry dual to $\mathbb{Z}_2^{(0)}$ and $W(M_2) = e^{i\pi \oint_{M_2} c_2}$ is the defect for the 1-form symmetry $\mathbb{Z}_2^{(1)}$ dual to $\mathbb{Z}_2^{(1),C}$.

### 8.2.3 Fusion Rules: $Sc(4N)$

The fusion algebra is determined in appendix B and is

$$
\begin{aligned}
\mathcal{N}(M_3; A_1) \times \mathcal{N}(M_3; A_1) &= \frac{1}{|H^0(M_3, \mathbb{Z}_2)|^2} \sum_{M_2, M_2' \in H_2(M_3, \mathbb{Z}_2)} W(M_2) V(M_2')\,, \\
\mathcal{N}(M_3; A_1) \times V(M_2') &= \mathcal{N}(M_3; A_1)\,, \\
\mathcal{N}(M_3; A_1) \times W(M_2) &= \mathcal{N}(M_3; A_1)\,,
\end{aligned}
\tag{279}
$$

where $W(M_2)$ and $V(M_2')$ are the codimension-two operators generating the one-form symmetries dual to $\mathbb{Z}_2^{(1),C} \times \mathbb{Z}_2^{(1),B'}$.

## 8.3 Non-Invertibles from 2-Groups in Pure 4d $\mathfrak{so}(4N+2)$ Yang-Mills

The 4d Spin$(4N+2)$ pure gauge theory has a $\mathbb{Z}_2^{(0)}$ charge conjugation 0-form symmetry, while the 1-form symmetry is $\mathbb{Z}_4^{(1)}$. They form a 2-group [60]

$$
\delta B_2 = \text{Bock}(C_2) + A_1 C_2\,,
\tag{280}
$$

where $A_1$ is the background for the 0-form symmetry, and $B_2, C_2$ are backgrounds for the two $\mathbb{Z}_2$ factors in the 1-form symmetry, which form an extension to $\mathbb{Z}_4^{(1)}$

$$
1 \to \mathbb{Z}_2 \to \mathbb{Z}_4 \to \mathbb{Z}_2 \to 1\,.
\tag{281}
$$

The Bock is the Bockstein homomorphism for this extension sequence.

By gauging various combination of the symmetries of Spin$(4N+2)$ we can go to different theories with non-invertibles

1. $PO(4N+2)$:
   gauge $B_2, A_1, C_2$: we obtain codimension-2 non-invertible defect;

2. Pin$^+(4N+2)$:
   gauge $A_1$: we obtain a codimension-2 non-invertible defect.

The non-invertible defects that we obtain are

- $\mathcal{N}(M_2; B_2')$: non-invertible defect in the $PO(4N+2)$ theory, corresponding to the codimension-2 defect for the $\mathbb{Z}_2^{(1)}$ symmetry dual to $\mathbb{Z}_2^{(1),B}$;

- $\mathcal{N}(M_2; C_2)$: non-invertible defect in the Pin$^+(4N+2)$ theory, corresponding to the codimension-2 defect for the $\mathbb{Z}_2^{(1),C}$ symmetry.

### 8.3.1 Fusion Rules for Pin$^+(4N + 2)$

The fusion algebra that we find in appendix is B.2

$$
\begin{aligned}
\mathcal{N}(M_2; C_2) \times \mathcal{N}(M_2; C_2) &= \frac{1 + T(M_2)}{|H^0(M_2, \mathbb{Z}_2)|} \sum_{M_1 \in H_1(M_2, \mathbb{Z}_2)} e^{i\pi Q(M_1)} L(M_1), \\
\mathcal{N}(M_2; C_2) \times T(M_2) &= \mathcal{N}(M_2; C_2), \\
\mathcal{N}(M_2; C_2) \times L(M_1) &= e^{i\pi Q(M_1)} \mathcal{N}(M_2; C_2).
\end{aligned}
\tag{282}
$$

Here we defined $Q(M_1) = \oint_{M_2} \mathrm{Bock}(\epsilon_1) = \oint_{M_2} \epsilon_1 \cup \epsilon_1$, where $\epsilon_1$ is the Poincaré dual of $M_1$. This additional phase is non-trivial only on non-orientable manifolds. $L(M_1) = e^{i\pi \oint_{M_1} a_1}$ is the defect for the $\mathbb{Z}_2^{(2)}$ 2-form symmetry dual to $\mathbb{Z}_2^{(0)}$ and $T(M_2) = e^{i\pi \oint b_2'}$ is the defect for the 1-form symmetry $\mathbb{Z}_2^{(1),B}$.

### 8.3.2 Fusion Rules for $PO(4N + 2)$

The fusion algebra is given by

$$
\begin{aligned}
\mathcal{N}(M_2; B_2') \times \mathcal{N}(M_2; B_2') &= \frac{1 + W(M_2)}{|H^0(M_2, \mathbb{Z}_2)|} \sum_{M_1 \in H_1(M_2, \mathbb{Z}_2)} e^{i\pi Q(M_1)} L(M_1), \\
\mathcal{N}(M_2; B_2') \times W(M_2) &= \mathcal{N}(M_2; B_2'), \\
\mathcal{N}(M_2; B_2') \times L(M_1) &= e^{i\pi Q(M_1)} \mathcal{N}(M_2; B_2').
\end{aligned}
\tag{283}
$$

Here we defined $Q(M_1)$ and $L(M_1)$ as above. $W(M_2) = e^{i\pi \oint_{M_2} c_2}$ is the defect implementing the 1-form symmetry dual to $\mathbb{Z}_2^{(1),C}$.

## 8.4 Extension to $\mathfrak{so}(4N)$ Yang-Mills Theories in any Dimension

The construction of non-invertible symmetries in the Spin pure gauge theories can be straightforwardly extended to generic dimension dimension $d$. Let us consider the Spin$(4N)$ case for concreteness. The theory has $\mathbb{Z}_2^{(1)} \times \mathbb{Z}_2^{(1)}$ 1-form symmetry, $\mathbb{Z}_2^{(0)}$ outer automorphism, and 2-group $\delta B_2 = A_1 C_2$. We now gauge the $\mathbb{Z}_2^{(1)}$ 1-form symmetry with background $B_2$ as we did above. The new coupling $\pi \int_{M_d} B_2 B_{d-2}$ has a bulk dependency

$$
\mathcal{A} = \pi \int_{M_{d+1}} A_1 C_2 B_{d-2}.
\tag{284}
$$

Here $B_{d-2}$ is the background for the $\mathbb{Z}_2^{(d-3)}$ $(d-3)$-form symmetry dual to $\mathbb{Z}_2^{(1)}$. Now the discussion is completely analogous to the one in 4d. We list the possibilities for non-invertible symmetries.

$Pin^+(4N)$. Gauge $A_1$ and $B_{d-2}$ first. In this case the codimension-2 defect generating $\mathbb{Z}_2^{(1)}$ becomes non-invertible. These satisfy the fusion algebra

$$
\begin{aligned}
&\mathcal{N}(M_{d-2}) \times \mathcal{N}(M_{d-2}) \\
&= \frac{|H^{d-5}(M_{d-2}, \mathbb{Z}_2)| \cdots}{|H^{d-4}(M_{d-2}, \mathbb{Z}_2)| \cdots} (1 + e^{i\pi \oint_{M_{d-2}} b_{d-2}}) \sum_{M_1 \in H_1(M_{d-2}, \mathbb{Z}_2)} e^{i\pi \oint_{M_1} a_1}.
\end{aligned}
\tag{285}
$$

$PO(4N)$. Consider the gauging of $A_1$ and $C_2$. In this case the defect generating $\mathbb{Z}_2^{(d-3)}$ becomes non-invertible (notice it has dimension 2). We have the fusion

$$
\begin{aligned}
&\mathcal{N}(M_2) \times \mathcal{N}(M_2) \\
&= \frac{1 + e^{i\pi \oint_{M_2} c_2}}{|H^0(M_2, \mathbb{Z}_2)|} \sum_{M_1 \in H_1(M_2, \mathbb{Z}_2)} e^{i\pi \oint_{M_1} a_1} \,.
\end{aligned}
\tag{286}
$$

$Sc(4N)$. Finally, consider gauging $C_2$ and $B_{d-2}$. In this case the codimension-1 defect implementing $\mathbb{Z}_2^{(0)}$ becomes non-invertible. We find the fusion rules

$$
\begin{aligned}
&\mathcal{N}(M_{d-1}) \times \mathcal{N}(M_{d-1}) \\
&= \frac{1}{|H^0(M_{d-1}, \mathbb{Z}_2)|} \frac{|H^{d-5}(M_{d-1}, \mathbb{Z}_2)| \dots}{|H^{d-4}(M_{d-1}, \mathbb{Z}_2)| \dots} \sum_{\substack{M_2 \in H_2(M_{d-1}, \mathbb{Z}_2) \\ M_{d-2} \in H_{d-2}(M_{d-1}, \mathbb{Z}_2)}} e^{i\pi \oint_{M_2} c_2} e^{i\pi \oint_{M_{d-2}} b_{d-2}} \,.
\end{aligned}
\tag{287}
$$

### 8.5 Non-Invertibles from 2-Groups in Pure 4d $O(2)$ Theory

Complementing the higher-category gauging analysis in section 6.3 we provide a derivation of the fusion of topological defects in the 4d $O(2)$ gauge theory using the mixed anomaly approach.

Recall that the $O(2)$ gauge theory can be obtained by $U(1)$ gauge theory by gauging $\mathbb{Z}_2^{(0)}$ charge conjugation. $U(1)$ gauge theory also has a $U(1)$ electric 1-form symmetry generated by Gukov-Witten operators. To our knowledge, the approach of [17] allows us only to consider a $\mathbb{Z}_4 \subset U(1)$ of the 1-form symmetry, so it is more limited than the higher-categorical approach we used in section 6.3 to study non-invertible symmetries in the same theory.

In particular, we start by considering a discrete $\mathbb{Z}_4^{(1)}$ subgroup of the 1-form symmetry. The analysis is very similar to the one for the Spin$(4N + 2)$ theory. Namely, we have a 2-group

$$
\delta B_2 = \text{Bock}(C_2) + C_2 A_1 \,,
\tag{288}
$$

where $B_2$ and $C_2$ are backgrounds for two $\mathbb{Z}_2^{(1)}$ symmetries inside $\mathbb{Z}_4^{(1)}$, while $A_1$ is the background for charge-conjugation. If we now gauge $B_2$, we obtain theory with a mixed anomaly

$$
\mathcal{A} = \pi \int_{M_5} B_2' \text{Bock}(C_2) + B_2' C_2 A_1 \,,
\tag{289}
$$

where $B_2'$ is the background for the dual $\mathbb{Z}_2^{(1)}$ 1-form symmetry. At this point we want to gauge $B_2'$ and $A_1$ to make the codimension-2 defect implementing the $\mathbb{Z}_2^{(1)}$ 1-form symmetry associated to $C_2$ non-invertible. Notice that the net effect of this series of gaugings is to gauge $A_1$, *i.e.* charge conjugation, in the $U(1)$ gauge theory, hence we expect to recover a subset of the non-invertible defects of the $O(2)$ gauge theory.

Using the result for the Spin$(4N + 2)$ theory, we obtain a non-invertible defect $\mathcal{N}(M_2)$ which has fusion

$$
\mathcal{N}(M_2) \times \mathcal{N}(M_2) = \frac{1 + T(M_2)}{|H^0(M_2, \mathbb{Z}_2)|} \sum_{M_1 \in H_1(M_2, \mathbb{Z}_2)} e^{i\pi Q(M_1)} L(M_1) \,.
\tag{290}
$$

Here $L(M_1)$ is the topological invertible line operator implementing the dual symmetry $\mathbb{Z}_2^{(2)}$, while $T(M_2)$ is the codimension-2 topological invertible operator generating the $\mathbb{Z}_2^{(1),B}$ 1-form

symmetry. We also defined $Q(M_1) = \oint_{M_2} \text{Bock}(\epsilon_1) = \oint_{M_2} \epsilon_1 \cup \epsilon_1$, where $\epsilon_1$ is the Poincaré dual of $M_1$. This additional phase is non-trivial only on non-orientable manifolds.

This is a special case of the fusion algebra of section 6.3 for the case of $\theta = \frac{1}{4}$ with the identifications

$$
\begin{aligned}
S_2^{(1/4)} &\longleftrightarrow \mathcal{N}(M_2), \\
D_2^{(1/2)} &\longleftrightarrow T(M_2), \\
D_1^{(-)} &\longleftrightarrow , L(M_1).
\end{aligned}
\tag{291}
$$

## 8.6 Non-Invertibles from Higher-Groups: 6d and 5d Theories

In this section we explore non-invertible symmetries appearing in 6d and 5d theories from the higher-group approach. This complements the analysis in section 7, where we used the higher-category gauging.

To apply the higher-group approach, we need to however restrict here to an absolute theory (e.g. pick a polarization on the defect group of 1-/2-form symmetries in 5d). In particular, we will find examples of non-invertibles in 6d $(2,0)$ *absolute* SCFTs and in 5d KK theories obtained as circle compactifications of 6d $(1,0)$ SCFTs. 6d $(2,0)$ SCFTs are an example of *relative* theories [61,62], in the sense that they have defects – in this case 2d surfaces – which are mutually non-local, meaning that there is phase ambiguity in defining the correlation function of two such defects. This implies that the theory is not well defined on its own, but rather must be thought as living at the boundary of a 7d TQFT. For 6d $(2,0)$ SCFTs specified by an ADE algebra $\mathfrak{g}$, the defect group $D$ [63] is simply given by the center of the simply connected group $\widetilde{G}$ with Lie algebra $\mathfrak{g}$

$$
D = Z_{\widetilde{G}}.
\tag{292}
$$

Given a relative theory, one can obtain an *absolute* theory by choosing subgroup a $L \subseteq D$ of the defect group corresponding to picking a subset of mutually local surface operators. This is often referred to as a choice of *polarization*. 6d $(2,0)$ absolute theories were classified (upto two simple factors) in [64]. The data entering this classification are a Lie algebra $\mathfrak{g} = \mathfrak{g}_1 \oplus \mathfrak{g}_2 \oplus ... \oplus \mathfrak{g}_r$, where each summand is of ADE type, and the aforementioned choice of polarization.

In particular, we will consider the theories

$$
(A_{15}, \mathbb{Z}_4), \; (D_n \oplus D_n, \mathbb{Z}_2 \times \mathbb{Z}_2),
\tag{293}
$$

where the first entry denotes the choice of algebra and the second one the choice of $L$. $L$ gives the 2-form symmetry group of the theory.

We will then consider example of non-invertibles in 5d theories obtained by circle compactification from 6d SCFTs. We pick 5d absolute KK theories with a 1-form symmetry group inherited from 6d, both from the defect group and the 1-form symmetry itself of the 6d SCFT [65]. In particular, we will look at non-Higgsable clusters (NHCs), which are building blocks for 6d $\mathcal{N} = (1,0)$ SCFTs with non-trivial defect group. The single node NHC consists of a single curve with negative self-intersection number $-n$ with non-Higgsable gauge algebra $\mathfrak{g}$. We will look in particular at two examples, namely $\mathfrak{su}(3)$ on a $-3$ curve, which has defect group $\mathbb{Z}_3$, and $\mathfrak{so}(8)$ on a $-4$ curve, which has defect group $\mathbb{Z}_4$. Upon compactification to 5d, we can pick an absolute 5d KK theory whose 1-form symmetry is given by [65]

$$
\Gamma_{5d}^{(1)} = D \oplus Z_{\widetilde{G}},
\tag{294}
$$

where $D$ is the 6d defect group and $Z_{\widetilde{G}}$ is the center of the simply connected group with algebra $\mathfrak{g}$.

### 8.6.1   6d $(2,0)$ **Theories of Type** $(D_n \oplus D_n)$

We first consider the 6d $(2,0)$ absolute theory of type $(D_n \oplus D_n)$. This case was also studied from the higher gauging in section 7. This theory has a 2-form symmetry $\mathbb{Z}_2^{(2),C} \times \mathbb{Z}_2^{(2),B}$ and a $\mathbb{Z}_2^{(0)}$ outer-automorphism 0-form symmetry symmetry which exchanges the two $D_n$ copies, and hence the two 2-form symmetries. Let us denote the background gauge fields for the 2-form symmetry by $B_3, C_3$ and by $A_1$ the background gauge field for the 0-form symmetry. The symmetries form a non-trivial 3-group which reads

$$\delta B_3 = A_1 C_3. \tag{295}$$

Now we gauge the $\mathbb{Z}_2^{(2),B}$ subgroup of the 2-form symmetry. We gain a dual 2-form symmetry $\mathbb{Z}_2^{(2),B'}$ whose background we denote by $B_3'$. Due to the 3-group structure (295), we obtain a mixed anomaly

$$\mathcal{A} = \pi \int_{M_7} A_1 C_3 B_3'. \tag{296}$$

Very similarly to the 4d $\mathrm{Spin}(4N)$ case, we can now gauge various combinations of symmetries appearing in (296) and obtain theories with non-invertible symmetries.

**Gauge $C_3$ and $B_3'$.**   After gauging the $\mathbb{Z}_2^{(2),C}$ the codimension-1 defect generating the 0-form symmetry becomes non-invertible. We denote the non-invertible defect by $\mathcal{N}(M_5; A_1)$. The fusion of two defects can be computed as in appendix B and gives

$$\begin{aligned}
\mathcal{N}(M_5; A_1) \times \mathcal{N}(M_5; A_1) &= \frac{|H^0(M_5, \mathbb{Z}_2)|^2}{|H^1(M_5, \mathbb{Z}_2)|^2} \sum_{M_3, N_3 \in H_3(M_5, \mathbb{Z}_2)} T(M_3) W(N_3), \\
\mathcal{N}(M_5; A_1) \times T(M_3) &= \mathcal{N}(M_5; A_1), \\
\mathcal{N}(M_5; A_1) \times W(M_3) &= \mathcal{N}(M_5; A_1).
\end{aligned} \tag{297}$$

Here $T(M_3) = e^{i\pi \oint_{M_3} b_3'}$ is the defect for $\mathbb{Z}_2$ and $W(N_3) = e^{i\pi \oint_{N_3} c_3}$ is the defect generating the 2-form symmetry dual to $\mathbb{Z}_2^{(2),C}$.

**Gauge $A_1$ and $C_3$.**   We can also gauge $A_1$ and $C_3$, effetively gauging the full 3-group. In this case what becomes non-invertible is the defect for the symmetry $\mathbb{Z}_2^{(2),B'}$. We denote the non-invertible defect by $\mathcal{N}(M_3; B_3')$.
    The fusion rules are given by

$$\begin{aligned}
\mathcal{N}(M_3; B_3') \times \mathcal{N}(M_3; B_3') &= \frac{|H^0(M_3, \mathbb{Z}_2)|}{|H^1(M_3, \mathbb{Z}_2)|} (1 + W(M_3)) \sum_{M_1 \in H_1(M_3, \mathbb{Z}_2)} L(M_1), \\
\mathcal{N}(M_3; B_3') \times W(M_3) &= \mathcal{N}(M_3; B_3'), \\
\mathcal{N}(M_3; B_3') \times L(M_1) &= \mathcal{N}(M_3; B_3'),
\end{aligned} \tag{298}$$

where $L(M_1) = e^{i\pi \oint_{M_1} a_1}$ is the defect for the 4-form symmetry dual to $\mathbb{Z}_2^{(0)}$, while $W(M_3)$ is defined as above.

**Gauge $A_1$ and $B_3'$.**   We can also gauge $A_1$ and $B_3'$, effetively gauging only charge conjugation in the original theory. In this case what becomes non-invertible is the defect for the symmetry

$\mathbb{Z}_2^{(2),C}$. We denote the non-invertible defect by $\mathcal{N}(M_3; C_3)$. The fusion rules are given by

$$
\begin{aligned}
\mathcal{N}(M_3; C_3) \times \mathcal{N}(M_3; C_3) &= \frac{|H^0(M_3, \mathbb{Z}_2)|}{|H^1(M_3, \mathbb{Z}_2)|} (1 + T(M_3)) \sum_{M_1 \in H_1(M_3, \mathbb{Z}_2)} L(M_1), \\
\mathcal{N}(M_3; C_3) \times T(M_3) &= \mathcal{N}(M_3; C_3), \\
\mathcal{N}(M_3; C_3) \times L(M_1) &= \mathcal{N}(M_3; C_3),
\end{aligned}
\tag{299}
$$

where $L(M_1)$ and $T(M_3)$ are defined as above.

### 8.6.2  6d $(2,0)$ of Type $A_{n^2-1}$ For $n = 4$

Consider the $A_{n^2-1}$ theory for $n = 4$. Here we have a $\mathbb{Z}_4^{(2)}$ 2-form symmetry and the $\mathbb{Z}_2^{(0)}$ outer-automorphism acts on it by sending a generator of $\mathbb{Z}_4^{(2)}$ to its inverse. In terms of background fields, we have a 3-group of the form

$$
\delta B_3 = \text{Bock}(C_3) + A_1 \cup C_3,
\tag{300}
$$

The discussion here is very similar to the one for $\text{Spin}(4N+2)$ YM. We write down explicitly only the case where we gauge the full 3-group. The defect that becomes non-invertible is the one implementing the 2-form symmetry $\mathbb{Z}_2^{(2),B'}$ dual to $\mathbb{Z}_2^{(2),B}$. Denote this defect by $\mathcal{N}(M_3; B_3')$. The fusion rules are derived in a similar fashion to the 4d examples, and we find

$$
\begin{aligned}
\mathcal{N}(M_3; B_3') \times \mathcal{N}(M_3; B_3') &= \frac{|H^0(M_3, \mathbb{Z}_2)|}{|H^1(M_3, \mathbb{Z}_2)|} (1 + W(M_3)) \sum_{M_1 \in H_1(M_3, \mathbb{Z}_2)} e^{i\pi Q(M_1)} L(M_1), \\
\mathcal{N}(M_3; B_3') \times W(M_3) &= \mathcal{N}(M_3; B_3'), \\
\mathcal{N}(M_3; B_3') \times L(M_1) &= e^{i\pi Q(M_1)} \mathcal{N}(M_3; B_3'),
\end{aligned}
\tag{301}
$$

where $L(M_1) = e^{i\pi \oint_{M_1} a_1}$ is the defect for the 4-form symmetry dual to $\mathbb{Z}_2^{(0)}$, while $W(M_3) = e^{i\pi \oint_{M_3} c_3}$ is the defect for the 2-form symmetry dual to $\mathbb{Z}_2^{(2),C}$. We also defined $Q(M_1) = \int_{M_3} \text{Bock}(\gamma_2)$, where $\gamma_2 \in H^2(M_3, \mathbb{Z}_2)$ is the Poincaré dual of $M_1$ in $M_3$.

## 8.7  5d Theories

There are several theories in 5d that have anomalies that are amenable to being gauged and result in non-invertible symmetries. We focus on 5d KK-theories, which are obtained from 6d SCFTs, by compactifications on $S^1$. The theories in 6d have 2-form symmetries and 1-form symmetries [65, 66], which descend in 5d to 1-form symmetries (for a particular choice of polarization) by (294).[7] We will focus on two examples: the non-Higgsable clusters, which correspond to an $\mathfrak{su}(3)$ ($\mathfrak{so}(8)$) singularity tuned over a $-3$ ($-4$) self-intersection curve in F-theory.

### 8.7.1  $\mathfrak{su}(3)$ on a $-3$ curve

This theory has a $\mathbb{Z}_3^{(1)} \times \mathbb{Z}_3^{(1)}$ 1-form symmetry. We denote the backgrounds for the two factors by $B_2$ and $C_2$ respectively. The two symmetries have a mixed anomaly [67, 69, 70]

$$
\mathcal{A} = \frac{2\pi}{3} \int_{M_6} C_2 \cup B_2 \cup B_2.
\tag{302}
$$

---

[7]As shown in [67], the 1-form symmetries that we will gauge, do not have a $B^3$ type anomaly [64, 68], which would obstruct gauging these symmetries.

Since we have a theory with a mixed anomaly, we can gauge part of the symmetries involved to obtain non-invertible defects.

Note however that gauging the $\mathbb{Z}_3^{(1)}$ associated to $C_2$ does not give a non-invertible defect, but rather results in a higher-group symmetry. Indeed, consider the partition function of the theory with $C_2$ gauged:

$$
\begin{aligned}
\widehat{Z}[M_5; C_3, B_2] &= \int \mathcal{D}c_2 \, Z[M_5; c_2, B_2] \, e^{\frac{2\pi i}{3} \int_{M_6} c_2 \cup B_2 \cup B_2} e^{\frac{2\pi i}{3} \int_{M_5} c_2 \cup C_3} \\
&= \int \mathcal{D}c_2 \, Z[M_5; c_2, B_2] \, e^{\frac{2\pi i}{3} \int_{M_6} c_2 \cup B_2 \cup B_2} e^{\frac{2\pi i}{3} \int_{M_6} c_2 \cup \delta C_3},
\end{aligned}
\tag{303}
$$

where $C_3$ is the background for the dual 2-form symmetry $\mathbb{Z}_3^{(2)}$. The bulk dependency is re-absorbed simply by setting $\delta C_3 = B_2 \cup B_2$, which results in a higher-group structure between the dual 2-form symmetry and the residual 1-form symmetry in the gauged theory.

Then to construct non-invertibles we consider the case in which we gauge $B_2$ and make the defect associated to $C_2$ non-invertible. We denote it by $D_{(n)}(M_3)$, with $n = 0, 1, 2$. Due to the mixed anomaly, we must consider the dressed defect

$$
D_{(n)}(M_3) \, e^{\frac{2\pi i n}{3} \int_{M_4} B_2 \cup B_2}.
\tag{304}
$$

Now consider gauging $B_2$ by making it a dynamical field by $b_2$.

To make $D_{(n)}(M_3)$ well defined we must couple it with a TQFT which cancels the anomaly

$$
\frac{2\pi i n}{3} \int_{M_4} b_2 \cup b_2 = \frac{4\pi i n}{3} \int_{M_4} \frac{\mathcal{P}(b_2)}{2},
\tag{305}
$$

where $\mathcal{P}(b_2)$ is the Pontryagin square of $b_2$. It was shown in [71] that there is a notion of minimal TQFT with such an anomaly. In particular, we have

$$
\mathcal{A}^{N,p} \quad \longleftrightarrow \quad \text{minimal TQFT living at the boundary of } 2\pi \frac{p}{N} \int_{M_4} \frac{\mathcal{P}(B_2)}{2}.
\tag{306}
$$

Thus we define

$$
\begin{aligned}
\mathcal{N}_{(1)}(M_3) &= D_{(1)}(M_3) \, \mathcal{A}^{3,-2}(M_3, b_2) = D_{(1)}(M_3) \, \mathcal{A}^{3,1}(M_3, b_2), \\
\mathcal{N}_{(2)}(M_3) &= D_{(2)}(M_3) \, \mathcal{A}^{3,-4}(M_3, b_2) = D_{(1)}(M_3) \, \mathcal{A}^{3,2}(M_3, b_2),
\end{aligned}
\tag{307}
$$

where we used $\mathcal{A}^{3,-2} \cong \mathcal{A}^{3,1}$ since $p$ is defined mod $N$ for $\mathcal{A}^{N,p}$ a spin TQFT.

To compute fusions between $\mathcal{N}_{(n)}(M_3)$ and its orientation reversal $\overline{\mathcal{N}}_{(n)}(M_3)$, we can use the following duality derived in [71]

$$
\mathcal{A}^{N,p} \otimes \mathcal{A}^{N,-p} \longleftrightarrow (\mathcal{Z}_N)_{-pN} \quad \text{when } \gcd(N, p) = 1.
\tag{308}
$$

Here $(\mathcal{Z}_N)_{-pN}$ is the $\mathbb{Z}_N$ DW discrete gauge theory, which can be described by the continuum action

$$
\int -\frac{pN}{4\pi} x dx + \frac{N}{2\pi} x dy,
\tag{309}
$$

with $x$ and $y$ $U(1)$ gauge fields.

For example, we can compute the fusion between $\mathcal{N}_{(1)}$ and its orientation reversal $\overline{\mathcal{N}}_{(1)}$. This

is given by

$$
\begin{aligned}
\overline{\mathcal{N}}_{(1)}(M_3) \times \mathcal{N}_{(1)}(M_3) &= \mathcal{A}^{3,-1}(M_3, b_2) \times \mathcal{A}^{3,1}(M_3, b_2) = (\mathcal{Z}_3)_{-3}(M_3, b_2) \\
&= \int \mathcal{D}x_1 \mathcal{D}y_1 \, e^{i\int_{M_3} -\frac{3}{4\pi}x_1 dx_1 + \frac{3}{2\pi}x_1 dy_1 + i\int_{M_3} x_1 b_2} \\
&= \sum_{\tilde{x}_1 \in H^1(M_3, \mathbb{Z}_3)} e^{-i\pi\int_{M_3} \tilde{x}_1 \mathrm{Bock}(\tilde{x}_1)} e^{\frac{2\pi i}{3}\int_{M_3} \tilde{x}_1 b_2} \\
&= \sum_{M_2 \in H_2(M_3, \mathbb{Z}_3)} e^{i\pi Q(M_2)} e^{\frac{2\pi i}{3}\oint_{M_2} b_2} ,
\end{aligned}
\tag{310}
$$

where $x_1 = \frac{2\pi}{3}\tilde{x}_1$ with $\tilde{x}_1 \in H^1(M_3, \mathbb{Z}_3)$ a discrete gauge field, $M_2$ is the Poincaré dual of $\tilde{x}_1$ in $M_3$ and $Q(M_2) = \int_{M_3} \tilde{x}_1 \mathrm{Bock}(\tilde{x}_1)$. In summary we obtain

$$
\overline{\mathcal{N}}_{(1)}(M_3) \times \mathcal{N}_{(1)}(M_3) = \frac{1}{|H^0(M_3, \mathbb{Z}_3)|} \sum_{M_2 \in H_2(M_3, \mathbb{Z}_3)} e^{i\pi Q(M_2)} W_{(1)}(M_2) ,
\tag{311}
$$

where $W_{(1)}(M_2) = e^{\frac{2\pi i}{3}\oint_{M_2} b_2}$ generates the dual $\mathbb{Z}_3^{(2)}$ 2-form symmetry.

### 8.7.2 $\mathfrak{so}(8)$ on a $-4$ curve

This theory has a $\mathbb{Z}_4^{(1)}$ 1-form symmetry, whose background we denote by $C_2$, and a $\mathbb{Z}_2^{(1)} \times \mathbb{Z}_2^{(1)}$ 1-form symmetry, whose backgrounds we denote by $B_2^{(1)}$ and $B_2^{(2)}$ respectively for the two factors. The theory has a mixed anomaly given by [67, 70]

$$
\mathcal{A} = \pi \int_{M_6} C_2 \cup B_2^{(1)} \cup B_2^{(2)} .
\tag{312}
$$

We denote by $D_{(n)}(M_3)$ the defect implementing the $\mathbb{Z}_4^{(1)}$ symmetry, where $n = 0, 1, 2, 3$. Due to the mixed anomaly, we can make it gauge invariant under background gauge transformations of $B_2^{(1)}$ and $B_2^{(2)}$ by considering the combination

$$
D_{(n)}(M_3) e^{\pi i n \int_{M_4} B_2^{(1)} \cup B_2^{(2)}} .
\tag{313}
$$

Note in particular that the defect $D_{(2)}(M_3)$ does not have an anomaly, so that $D_{(0)}(M_3)$ and $D_{(2)}(M_3)$ generate an anomaly free $\mathbb{Z}_2^{(1)}$ subgroup. Upon gauging $\mathbb{Z}_2^{(1)} \times \mathbb{Z}_2^{(1)}$, the defects $D_{(1)}$ and $D_{(3)}$ become non-invertible. We denote by $\mathcal{N}_{(1),(3)}$ the respective non-invertible defects. The fusions are given by

$$
\begin{aligned}
\mathcal{N}_{(1)}(M_3) \times \mathcal{N}_{(3)}(M_3) &= \frac{1}{|H^0(M_3, \mathbb{Z}_3)|^2} \sum_{M_2^{(1)}, M_2^{(2)} \in H_2(M_3, \mathbb{Z}_2)} V^{(1)}(M_2^{(1)}) V^{(2)}(M_2^{(2)}) , \\
\mathcal{N}_{(1)}(M_3) \times D_{(2)}(M_3) &= \mathcal{N}_{(3)}(M_3) , \\
\mathcal{N}_{(3)}(M_3) \times D_{(2)}(M_3) &= \mathcal{N}_{(1)}(M_3) .
\end{aligned}
\tag{314}
$$

Here $V^{(1),(2)}(M_2^{(1),(2)})$ are the defects for each $\mathbb{Z}_2$ subgroup of the 2-form symmetry dual to $\mathbb{Z}_2^{(1)} \times \mathbb{Z}_2^{(1)}$.

# 9 Outlook

We have provided an operational definition of higher-categorical symmetries in higher dimensions. It would be important to put this proposal on firm mathematical foundations, developing the theory of higher-categories, connecting it to the mathematical literature, and determining similarly stringent constraints as they are known in three and lower dimensions. We pass the sniff-test in that the proposal agrees in 3d with the known theory in [30,31]. A crucial consistency requirement, which needs to be fully integrated into this formalism are constraints such as hexagon identities.

There are numerous ways to extend the work in this paper. The most obvious extension is to gauging higher-form symmetries in higher-categories. Deriving the fusion after higher-form symmetry gauging can furthermore in some instances be compared with the approach using mixed anomalies or higher groups discussed in section 8.

Most of our examples have been non-supersymmetric gauge theories in various dimensions. Clearly there are numerous supersymmetric ones – we have given examples of 5d and 6d theories, but of course likewise 4d SCFTs will be equally amenable to our approach. Exploring higher-categorical symmetries in geometric engineering will be another important milestone, as it will open up studies both of strongly-coupled supersymmetric QFTs, but also will play a role in the context of the swampland program (concretely, the no global symmetry conjecture). Understanding the action of twist operators on the string theoretic topological operators that generate the higher-form symmetries (see [65, 72–74]) will be crucial in implementing this construction in string theory.

The approach in section 8 as well as the closely-related [17], on the other hand starts with a higher-group symmetry or mixed anomaly for discrete symmetries. We have given examples of 5d and 6d theories with such structures, but multitude of examples can be constructed using the recent advances in geometric engineering of such discrete higher-group symmetries.

Finally it would be interesting to make contact with the mathematics literature on higher-category theory, such as the works [27,75].

## Acknowledgements

We thank Mathew Bullimore, Federico Bonetti, Dewi Gould, David Reutter, Ingo Runkel and Jingxiang Wu for discussions. This work is supported by the European Union's Horizon 2020 Framework through the ERC grants 682608 (LB and SSN) and 787185 (LB). SSN is supported in part by the "Simons Collaboration on Special Holonomy in Geometry, Analysis and Physics". AT is supported by the Swedish Research Council (VR) through grants number 2019-04736 and 2020-00214.

# A Further Examples

In this appendix we provide further examples of symmetry categories for 3d and 4d QFTs.

## A.1 Pure Pin$^+(4N + 2)$ Gauge Theory in 3d

Let us now consider analogous construction of pure Pin$^+(4N+2)$ Yang-Mills theory by gauging outer-automorphism 0-form symmetry of pure Spin$(4N+2)$ Yang-Mills theory. Before gauging, we have a 1-form symmetry group coming from the center of Spin$(4N+2)$

$$\Gamma^{(1)} = \mathbb{Z}_4\,, \tag{315}$$

and the outer-automorphism of $\mathfrak{so}(4N + 2)$ gives rise to a 0-form symmetry group

$$\Gamma^{(0)} = \mathbb{Z}_2\,, \tag{316}$$

which acts on $\Gamma^{(1)} = \mathbb{Z}_4$ by interchanging the generator of $\mathbb{Z}_4$ with the inverse of the generator, and leaving invariant the $\mathbb{Z}_2$ subgroup of $\mathbb{Z}_4$.

The category $\mathcal{C}_{\mathrm{Spin}(4N+2)}$ for $\mathrm{Spin}(4N + 2)$ theory has objects

$$\mathcal{C}^{\mathrm{ob}}_{\mathrm{Spin}(4N+2)} = \left\{ D_1^{(\mathrm{id})}, D_1^{(S)}, D_1^{(V)}, D_1^{(C)} \right\}\,, \tag{317}$$

with $D_1^{(S)}$ corresponding to the generator of $\mathbb{Z}_4$, $D_1^{(C)}$ corresponding to the inverse of the generator, $D_1^{(\mathrm{id})}$ corresponding to the identity element of $\mathbb{Z}_4$, and $D_1^{(V)}$ corresponding to the generator of the $\mathbb{Z}_2$ subgroup of $\mathbb{Z}_4$. The fusion of objects follows the group law of $\mathbb{Z}_4$.

Now we gauge $\mathbb{Z}_2$ to obtain a category $\mathcal{C}_{\mathrm{Pin}^+(4N+2)}$ describing topological line defects and local operators of the $\mathrm{Pin}^+(4N + 2)$ theory. A subset of simple objects of $\mathcal{C}_{\mathrm{Pin}^+(4N+2)}$ arise as objects of $\mathcal{C}_{\mathrm{Spin}(4N+2)}$ left invariant by the $\mathbb{Z}_2$ outer automorphism action. These are $D_1^{(\mathrm{id})}$, $D_1^{(V)}$ and

$$D_1^{(SC)} := \left( D_1^{(S)} \oplus D_1^{(C)} \right)_{\mathcal{C}_{\mathrm{Spin}}}\,, \tag{318}$$

where the subscript $\mathcal{C}_{\mathrm{Spin}(4N+2)}$ on the RHS reflects that the object $D_1^{(SC)}$ is decomposed as this direct sum only in the category $\mathcal{C}_{\mathrm{Spin}(4N+2)}$, but it is a simple object in the category $\mathcal{C}_{\mathrm{Pin}^+(4N+2)}$.

Other simple objects of $\mathcal{C}_{\mathrm{Pin}^+(4N+2)}$ are obtained by dressing with Wilson line defects. Note that the stabilizer for $D_1^{(\mathrm{id})}$, $D_1^{(V)}$ is the whole 0-form symmetry group $\mathbb{Z}_2$, while the stabilizer for $D_1^{(SC)}$ is trivial. Thus, we obtain new simple objects of $\mathcal{C}_{\mathrm{Pin}^+(4N+2)}$ by dressing $D_1^{(\mathrm{id})}$, $D_1^{(V)}$ with the non-trivial irrep of $\mathbb{Z}_2$. We call the resulting simple objects as $D_1^{(-)}$, $D_1^{(V_-)}$ respectively. Thus, the full set of simple objects of $\mathcal{C}_{\mathrm{Pin}^+(4N+2)}$ is

$$\mathcal{C}^{\mathrm{ob}}_{\mathrm{Pin}^+(4N)} = \left\{ D_1^{(\mathrm{id})}, D_1^{(-)}, D_1^{(SC)}, D_1^{(V)}, D_1^{(V_-)} \right\}\,. \tag{319}$$

Note that, at the level of objects, the category for $\mathrm{Pin}^+(4N+2)$ is the same as that for $\mathrm{Pin}^+(4N)$ discussed in the previous subsection. In fact, the reader can imitate the arguments of previous subsection and find that fusion rules of the objects are also exactly the same for $\mathrm{Pin}^+(4N+2)$ and $\mathrm{Pin}^+(4N)$.

Thus the category for the $\mathrm{Pin}^+(4N+2)$ theory is also a Tambara-Yamagami category based on $\mathbb{Z}_2 \times \mathbb{Z}_2$. Is it the same as the category for the $\mathrm{Pin}^+(4N)$ theory? It turns out, by computing the associators, that the answer is yes. That is, the category for the $\mathrm{Pin}^+(4N + 2)$ theory is also

$$\mathcal{C}_{\mathrm{Pin}^+(4N+2)} = \mathrm{Rep}(D_8)\,. \tag{320}$$

This can again be understood by constructing $\mathrm{Pin}^+(4N + 2)$ theory as a gauging of the pure $PSO(4N + 2)$ theory in 3d. The latter theory is obtained by gauging the $\mathbb{Z}_4$ 1-form symmetry of the $\mathrm{Spin}(4N + 2)$ theory, leading to a dual $\mathbb{Z}_4$ 0-form symmetry in the $PSO(4N + 2)$ theory. The outer-automorphism acts by interchanging generators of $\mathbb{Z}_4$, so the full 0-form symmetry of $PSO(4N + 2)$ theory is

$$\Gamma^{(0)} = \mathbb{Z}_4 \rtimes \mathbb{Z}_2\,. \tag{321}$$

It can be easily seen that this group is isomorphic to $D_8$, that is we have

$$\Gamma^{(0)} = D_8\,. \tag{322}$$

On the other hand, the $PSO(4N + 2)$ theory has trivial 1-form symmetry. The $\text{Pin}^+(4N + 2)$ theory is obtained by gauging the $D_8$ symmetry of the $PSO(4N + 2)$ theory, which creates Wilson line defects for $D_8$, thus implying that

$$\mathcal{C}_{\text{Pin}^+(4N+2)} = \text{Rep}(D_8) \tag{323}$$

is the category of symmetries for the $\text{Pin}^+(4N + 2)$ theory.

## A.2  Pure $\widetilde{SU}(N)$ Gauge Theory in 4d

Another example of disconnected gauge theory with a non-invertible symmetry is the $\widetilde{SU}(N)$ gauge theory. Here the group $\widetilde{SU}(N)$ is the so-called *principal extension* of $SU(N)$ (see [76] for some field theoretic discussions). This is obtained by taking the semi-direct product of $SU(N)$ with the $\mathbb{Z}_2$ outer-automorphism of its Dynkin diagram. Recall that the outer-automorphism of the $\mathfrak{su}(N)$ Dynkin diagram acts by flipping the order of its nodes, which corresponds to exchanging the fundamental and anti-fundamental representation of $SU(N)$. In this sense, we can also identify this $\mathbb{Z}_2$ with charge conjugation, and interpret $\widetilde{SU}(N)$ as $SU(N)$ with charge conjugation gauged.

The construction of non-invertible defects is very closely related to the construction for $O(2)$ in section 6.3. $SU(N)$ Yang-Mills theory has the following symmetries

$$\Gamma^{(1)} = \mathbb{Z}_N \quad , \quad \Gamma^{(0)} = \mathbb{Z}_2 \,. \tag{324}$$

The 1-form symmetry is described by a 2-category $\mathcal{C}_{SU(N)}$ which can be recognized as a subcategory of the the 2-category $\mathcal{C}_{U(1)}$ with $\theta \in \mathbb{Z}_N \subseteq \mathbb{R}/\mathbb{Z}$.

Consequently, the descending 2-category $\mathcal{C}_{\widetilde{SU}(N)}$ describing non-invertible symmetries in the $\widetilde{SU}(N)$ theory is described as a subcategory of $\mathcal{C}_{O(2)}$ whose simple objects modulo condensations

$$\mathcal{C}^{\text{ob}}_{\widetilde{SU}(N)} = \left\{ D_2^{(0)}, S_2^{(\theta)} \right\} \,, \tag{325}$$

for $N$ odd and with $\theta$ constrained to lie in the set

$$\{\theta \in \mathbb{Z}_N \subseteq \mathbb{R}/\mathbb{Z}\} \cap \{0 < \theta < 1/2\} \,. \tag{326}$$

Similarly the simple objects modulo condensations for $N$ even are

$$\mathcal{C}^{\text{ob}}_{\widetilde{SU}(N)} = \left\{ D_2^{(0)}, D_2^{(1/2)}, S_2^{(\theta)} \right\} \,, \tag{327}$$

with $\theta$ constrained to lie in the set (326). The fusion rules follow from the fusion rules for $O(2)$.

The simple 1-endomorphisms of simple objects modulo condensations are

$$\mathcal{C}^{\text{1-endo}}_{\widetilde{SU}(N)} = \left\{ D_1^{(0)}, D_1^{(-)}, L_1^{(\theta)} \right\} \,, \tag{328}$$

for $N$ odd and with $\theta$ constrained to lie in the set (326), and

$$\mathcal{C}^{\text{1-endo}}_{\widetilde{SU}(N)} = \left\{ D_1^{(0)}, D_1^{(-)}, D_1^{(1/2)}, D_1^{(1/2,-)}, L_1^{(\theta)} \right\} \,, \tag{329}$$

for $N$ even and with $\theta$ constrained to lie in the set (326). The fusion rules follow from the fusion rules for $O(2)$.

### A.3 A $\left[\mathbf{Spin}(4N) \times \mathbf{Spin}(4N)\right] \rtimes D_8$ **Gauge Theory with Matter in 4d**

Consider a 4d gauge theory with $\text{Spin}(4N) \times \text{Spin}(4N)$ gauge group with a scalar field in the bi-vector representation $(\mathbf{4N}, \mathbf{4N})$. The center symmetries $\mathbb{Z}_2^4$ are broken to

$$\Gamma^{(1)} = \mathbb{Z}_2^3 \,, \tag{330}$$

by the matter field. If no masses and potentials are introduced, then there is furthermore a 0-form symmetry group

$$\Gamma^{(0)} = D_8 = \mathbb{Z}_4 \rtimes \mathbb{Z}_2 = (\mathbb{Z}_2 \times \mathbb{Z}_2) \rtimes \mathbb{Z}_2 \,, \tag{331}$$

given by the dihedral group $D_8$ of order 8, which is the group of outer automorphisms of the $\mathfrak{so}(4N) \oplus \mathfrak{so}(4N)$ gauge algebra. The group of outer automorphisms is computed as the group of symmetries of the Dynkin diagram of $\mathfrak{so}(4N) \oplus \mathfrak{so}(4N)$. Exchanging the spinor and cospinor nodes for each $\mathfrak{so}(4N)$ subfactor gives rise to the $\mathbb{Z}_2 \times \mathbb{Z}_2$ subgroup of $D_8$ in its presentation as

$$D_8 = (\mathbb{Z}_2 \times \mathbb{Z}_2) \rtimes \mathbb{Z}_2 \,. \tag{332}$$

The other $\mathbb{Z}_2$ comes from the exchange of the two $\mathfrak{so}(4N)$ Dynkin diagrams, which acts on $\mathbb{Z}_2 \times \mathbb{Z}_2$ non-trivially. Thus the full group structure is non-abelian and given by the $D_8$ group. As the bivector representation is left invariant by each $\mathbb{Z}_2$, the $D_8$ outer automorphism descends to a 0-form symmetry of the $\text{Spin}(4N) \times \text{Spin}(4N)$ gauge theory under consideration.

The elements of $\Gamma^{(1)}$ are

$$\Gamma^{(1)} = \{(\text{idid}), (\text{id}V), (V\text{id}), (VV), (SS), (SC), (CS), (CC)\} \,, \tag{333}$$

with group structure

$$(ij) \times (kl) = (mn) \,, \tag{334}$$

where $m$ and $n$ are obtained as follows

$$\begin{aligned} m &= ik \,, \\ n &= jl \,, \end{aligned} \tag{335}$$

using the group structure of

$$\mathbb{Z}_2 \times \mathbb{Z}_2 = \{\text{id}, S, C, V\} \tag{336}$$

discussed earlier.

The $\mathbb{Z}_4$ subgroup of $D_8$ in its presentation

$$D_8 = \mathbb{Z}_4 \rtimes \mathbb{Z}_2 \tag{337}$$

acts on $\Gamma^{(1)}$ as

$$\begin{aligned} \mathbb{Z}_4^{(0)}: \quad &(SS) \to (CS) \to (CC) \to (SC) \to (SS) \,, \\ &(\text{id}V) \leftrightarrow (V\text{id}) \,, \end{aligned} \tag{338}$$

while $(\text{idid})$ and $(VV)$ are left invariant. The other $\mathbb{Z}_2$ in $D_8$ does not commute with this and acts as follows:

$$\mathbb{Z}_2^{(0)}: \quad (SS) \leftrightarrow (SC) \,, \quad (CC) \leftrightarrow (CS) \,, \tag{339}$$

and leaves all other elements invariant.

### A.3.1 $\mathbb{Z}_4^{(0)}$ Gauging

The 2-category $\mathcal{C}_{\text{Spin}\times\text{Spin}}$ before gauging has simple objects $D_2^{(i)}$ with $(i) \in \Gamma^{(1)}$ whose fusion follows group law, and simple 1-endomorphisms $D_1^{(i)}$ of $D_2^{(i)}$ whose fusion $\otimes$ also follows group law. Let us begin by gauging $\mathbb{Z}_4$ subgroup of $D_8$. The 2-category $\mathcal{C}_{\text{Spin}\times\text{Spin}}$ descends to a 2-category $\mathcal{C}_{(\text{Spin}\times\text{Spin})\rtimes\mathbb{Z}_4}$ of the resulting $\left[\text{Spin}(4N) \times \text{Spin}(4N)\right] \rtimes \mathbb{Z}_4$ gauge theory.

The simple objects modulo condensations of $\mathcal{C}_{(\text{Spin}\times\text{Spin})\rtimes\mathbb{Z}_4}$ are

$$\mathcal{C}_{(\text{Spin}\times\text{Spin})\rtimes\mathbb{Z}_4}^{\text{ob}} = \left\{ D_2^{(\text{idid})}, D_2^{(VV)}, S_2^{(V\text{id})}, S_2^{(SC)} \right\}, \tag{340}$$

where

$$\begin{aligned}
S_2^{(V\text{id})} &= \left( D_2^{(V\text{id})} \oplus D_2^{(\text{id}V)} \right)_{\mathcal{C}_{\text{Spin}\times\text{Spin}}}, \\
S_2^{(SC)} &= \left( D_2^{(SS)} \oplus D_2^{(CS)} \oplus D_2^{(CC)} \oplus D_2^{(SC)} \right)_{\mathcal{C}_{\text{Spin}\times\text{Spin}}},
\end{aligned} \tag{341}$$

as objects of the 2-category $\mathcal{C}_{\text{Spin}\times\text{Spin}}$.

The simple 1-endomorphisms of simple objects in $\mathcal{C}_{(\text{Spin}\times\text{Spin})\rtimes\mathbb{Z}_4}^{\text{ob}}$ are

$$\begin{aligned}
&\mathcal{C}_{(\text{Spin}\times\text{Spin})\rtimes\mathbb{Z}_4}^{\text{1-endo}} \\
&= \left\{ D_1^{(\text{idid})}, D_1^{(\omega)}, D_1^{(\omega^2)}, D_1^{(\omega^3)}, D_1^{(VV)}, D_1^{(VV,\omega)}, D_1^{(VV,\omega^2)}, D_1^{(VV,\omega^3)}, L_1^{(V\text{id})}, L_1^{(V\text{id},-)}, L_1^{(SC)} \right\},
\end{aligned} \tag{342}$$

where $D_1^{(\text{idid})}$, $D_1^{(VV)}$, $L_1^{(V\text{id})}$ and $L_1^{(SC)}$ are identity 1-endomorphisms of the simple objects $D_2^{(\text{idid})}$, $D_2^{(VV)}$, $S_2^{(V\text{id})}$ and $S_2^{(SC)}$ respectively. $D_1^{\omega^i}$ and $D_1^{VV\omega^i}$ for $i \in \{1,2,3\}$ are non-identity 1-endomorphisms of $D_2^{(\text{id})}$ and $D_2^{(VV)}$ respectively. $L_1^{(V\text{id},-)}$ is a non-identity 1-endomorphism of $S_2^{(V\text{id})}$ obtained by dressing $L_1^{(V\text{id})}$ with a non-trivial Wilson line for $\mathbb{Z}_2$, because $\mathbb{Z}_2 \subseteq \mathbb{Z}_4$ is the stabilizer group for the orbit of $(V\text{id})$. The simple 1-endomorphisms of $D_2^{(\text{id})}, D_2^{(VV)}, S_2^{(V\text{id})}, S_2^{(SC)}$ follow $\mathbb{Z}_4, \mathbb{Z}_4, \mathbb{Z}_2, \mathbb{Z}_1$ group laws respectively under fusions inside the surfaces (i.e. fusions parametrized by objects).

The fusion of the objects in $\mathcal{C}_{(\text{Spin}\times\text{Spin})\rtimes\mathbb{Z}_4}^{\text{ob}}$ is as follows. $D_2^{(\text{idid})}$ and $D_2^{(VV)}$ have fusion rules with each other such that they form a $\mathbb{Z}_2$ 1-form symmetry of the $\left[\text{Spin}(4N) \times \text{Spin}(4N)\right] \rtimes \mathbb{Z}_4$ theory. In particular, $D_2^{(\text{idid})}$ is the identity surface defect. The fusion of $D_2^{(VV)}$ with other objects is

$$\begin{aligned}
D_2^{(VV)} \otimes S_2^{(V\text{id})} &= S_2^{(V\text{id})}, \\
D_2^{(VV)} \otimes S_2^{(SC)} &= S_2^{(SC)}.
\end{aligned} \tag{343}$$

The remaining fusions of $S_2^{(V\text{id})}$ are

$$\begin{aligned}
S_2^{(V\text{id})} \otimes S_2^{(V\text{id})} &= \frac{D_2^{(\text{idid})}}{\mathbb{Z}_2} \oplus \frac{D_2^{(VV)}}{\mathbb{Z}_2}, \\
S_2^{(V\text{id})} \otimes S_2^{(SC)} &= 2S_2^{(SC)}.
\end{aligned} \tag{344}$$

Finally, we have

$$S_2^{(SC)} \otimes S_2^{(SC)} = \frac{D_2^{(\text{idid})}}{\mathbb{Z}_4} \oplus \frac{D_2^{(VV)}}{\mathbb{Z}_4} \oplus 2\frac{S_2^{(V\text{id})}}{\mathbb{Z}_2}. \tag{345}$$

The fusion rules under the bulk fusion structure $\otimes$ of the 1-endomorphisms are

$$
\begin{aligned}
D_1^{(\omega^p)} \otimes D_1^{(\omega^q)} &= D_1^{(\omega^{p+q})}, \\
D_1^{(VV,\omega^p)} \otimes D_1^{(\omega^q)} &= D_1^{(VV,\omega^{p+q})}, \\
D_1^{(VV,\omega^p)} \otimes D_1^{(VV,\omega^q)} &= D_1^{(\omega^{p+q})}, \\
D_1^{(\omega)} \otimes L_1^{(V\mathrm{id})} = D_1^{(\omega^3)} \otimes L_1^{(V\mathrm{id})} &= L_1^{(V\mathrm{id},-)}, \\
D_1^{(\omega)} \otimes L_1^{(V\mathrm{id},-)} = D_1^{(\omega^3)} \otimes L_1^{(V\mathrm{id},-)} &= L_1^{(V\mathrm{id})}, \\
D_1^{(\omega^2)} \otimes L_1^{(V\mathrm{id})} &= L_1^{(V\mathrm{id})}, \\
D_1^{(\omega^2)} \otimes L_1^{(V\mathrm{id},-)} &= L_1^{(V\mathrm{id},-)}, \\
D_1^{(\omega^p)} \otimes L_1^{(SC)} &= L_1^{(SC)}, \\
D_1^{(VV)} \otimes L_1^{(V\mathrm{id})} = D_1^{(VV,\omega^2)} \otimes L_1^{(V\mathrm{id})} &= L_1^{(V\mathrm{id})}, \\
D_1^{(VV)} \otimes L_1^{(V\mathrm{id},-)} = D_1^{(VV,\omega^2)} \otimes L_1^{(V\mathrm{id},-)} &= L_1^{(V\mathrm{id},-)}, \\
D_1^{(VV,\omega)} \otimes L_1^{(V\mathrm{id})} = D_1^{(VV,\omega^3)} \otimes L_1^{(V\mathrm{id})} &= L_1^{(V\mathrm{id},-)}, \\
D_1^{(VV,\omega)} \otimes L_1^{(V\mathrm{id},-)} = D_1^{(VV,\omega^3)} \otimes L_1^{(V\mathrm{id},-)} &= L_1^{(V\mathrm{id})}, \\
D_1^{(VV,\omega^p)} \otimes L_1^{(SC)} &= L_1^{(SC)}, \\
L_1^{(V\mathrm{id})} \otimes L_1^{(V\mathrm{id},-)} &= D_1^{(\omega)} \oplus D_1^{(\omega^3)} \oplus D_1^{(VV,\omega)} \oplus D_1^{(VV,\omega^3)}, \\
L_1^{(V\mathrm{id},-)} \otimes L_1^{(SC)} &= L_1^{(SC)(a)} \oplus L_1^{(SC)(b)}, \\
L_1^{(V\mathrm{id},-)} \otimes L_1^{(V\mathrm{id},-)} &= L_1^{\left(D_2^{(\mathrm{idid})}/\mathbb{Z}_2\right)} \oplus L_1^{\left(D_2^{(VV)}/\mathbb{Z}_2\right)}, \\
L_1^{(V\mathrm{id})} \otimes L_1^{(V\mathrm{id},-)} &= L_1^{\left(D_2^{(\mathrm{idid})}/\mathbb{Z}_2\right);\mathbb{Z}_2} \oplus L_1^{\left(D_2^{(VV)}/\mathbb{Z}_2\right);\mathbb{Z}_2},
\end{aligned}
\tag{346}
$$

where $D_1^{(\omega^0)} := D_1^{(\mathrm{idid})}$ and $D_1^{(VV,\omega^0)} := D_1^{(VV)}$; $L_1^{(SC)(i)}$ for $i \in \{a,b\}$ are copies of $L_1^{(SC)}$ in the two copies of respective surfaces appearing in the fusion of surfaces; $L_1^{\left(D_2^{(\mathrm{idid})}/\mathbb{Z}_2\right)}$ and $L_1^{\left(D_2^{(VV)}/\mathbb{Z}_2\right)}$ are identity lines of the surfaces $D_2^{(\mathrm{idid})}/\mathbb{Z}_2$ and $D_2^{(VV)}/\mathbb{Z}_2$ respectively; and $L_1^{\left(D_2^{(\mathrm{idid})}/\mathbb{Z}_2\right);\mathbb{Z}_2}$ and $L_1^{\left(D_2^{(VV)}/\mathbb{Z}_2\right);\mathbb{Z}_2}$ are lines generating $\mathbb{Z}_2$ subgroups of $\mathbb{Z}_2 \times \mathbb{Z}_2$ 0-form symmetries localized on the surfaces $D_2^{(\mathrm{idid})}/\mathbb{Z}_2$ and $D_2^{(VV)}/\mathbb{Z}_2$ respectively.

### A.3.2 $D_8$-Gauging

We next gauge the additional $\mathbb{Z}_2^{(0)}$. In terms of the surface defects, the ones in $\mathcal{C}_{(\mathrm{Spin}\times\mathrm{Spin})\rtimes\mathbb{Z}_4}$ are $\mathbb{Z}_2$ invariant, and thus the simple objects modulo condensation are

$$
\mathcal{C}^{\mathrm{ob}}_{(\mathrm{Spin}\times\mathrm{Spin})\rtimes D_8} = \left\{ D_2^{(\mathrm{idid})}, D_2^{(VV)}, S_2^{(V\mathrm{id})}, S_2^{(SC)} \right\}.
\tag{347}
$$

Next consider the action on the 1-endomorphisms: the $\mathbb{Z}_2$ acts by exchanging

$$
D_1^{(\omega)} \leftrightarrow D_1^{(\omega^3)}, \quad D_1^{(VV,\omega)} \leftrightarrow D_1^{(VV,\omega^3)},
\tag{348}
$$

and leaving the other 1-endomorphisms invariant. The simple 1-endomorphisms are

$$
\begin{aligned}
\mathcal{C}_{(\text{Spin}\times\text{Spin})\rtimes D_8}^{\text{1-endo}} \\
= \Big\{ & D_1^{(\text{idid})}, D_1^{(\epsilon)}, D_1^{(\omega\omega^3)}, D_1^{(\omega^2)}, D_1^{(\omega^2,\epsilon)}, \\
& D_1^{(VV)}, D_1^{(VV,\epsilon)}, D_1^{(VV,\omega\omega^3)}, D_1^{(VV,\omega^2)}, D_1^{(VV,\omega^2,\epsilon)}, \\
& L_1^{(V\text{id})}, L_1^{(V\text{id},\epsilon)}, L_1^{(V\text{id},-)}, L_1^{(V\text{id},-,\epsilon)}, L_1^{(SC)}, L_1^{(SC,\epsilon)} \Big\},
\end{aligned}
\tag{349}
$$

where $D_1^{(\epsilon)}$ is the non-trivial line on the identity surface due to the gauged $\mathbb{Z}_2$ symmetry and

$$
\begin{aligned}
D_1^{(\omega\omega^3)} &= \left( D_1^{(\omega)} \oplus D_1^{(\omega^3)} \right)_{\mathcal{C}_{(\text{Spin}\times\text{Spin})\rtimes\mathbb{Z}_4}}, \\
D_1^{(VV,\omega\omega^3)} &= \left( D_1^{(VV,\omega)} \oplus D_1^{(VV,\omega^3)} \right)_{\mathcal{C}_{(\text{Spin}\times\text{Spin})\rtimes\mathbb{Z}_4}}.
\end{aligned}
\tag{350}
$$

Furthermore, the lines $D_1^{(i,\epsilon)}$ are the non-identity endomorphisms on the surfaces $i = \text{idid}, V\text{id}$, who have non-trivial stabilizer $\mathbb{Z}_2$.

We leave the determination of fusion rules to the interested reader.

# B Derivation of the Fusion Rules for Spin Yang-Mills in 4d

In this appendix we compute the fusion rules for the $\text{Spin}(4N)$ and $\text{Spin}(4N + 2)$ gauge theories, which we discussed in section 8.

## B.1 Non-Invertible Symmetries from Spin($4N$) Yang-Mills

We start with pure $\text{Spin}(4N)$ gauge theory and for concreteness let us work in 4d. The theory has a $\Gamma^{(1)} = \mathbb{Z}_2^{(1),B} \times \mathbb{Z}_2^{(1),C}$ 1-form symmetry and a $\Gamma^{(0)} = \mathbb{Z}_2^{(0)}$ outer-automorphism 0-form symmetry. The two symmetries combine into a 2-group (263). Gauging $\mathbb{Z}_2^{(1),B}$ yields the $SO(4N)$ gauge theory by promoting $B_2$ to a dynamical field $b_2$, and a dual 1-form symmetry $\mathbb{Z}_2^{(1),B'}$ (in 4d), with background field $B_2'$. This couples as $\int_{M_4} b_2 B_2'$. Due to the 2-group (263) this coupling is ill-defined, since it has a bulk dependency and yields an t' Hooft anomaly (264) for the $SO(4N)$ theory. As described in the main text, we now gauge various combinations of global symmetries, which result in the following distinct 4d gauge theories:

1. Pin$^+(4N)$ theory:
   gauge $B_2'$ and $A_1$: the codimension-2 defect implementing $\mathbb{Z}_2^{(1),C}$ becomes non-invertible

2. $PO(4N)$ theory:
   gauge $C_2, A_1$: the codimension-2 defect implementing $\mathbb{Z}_2^{(1),B'}$ becomes non-invertible

3. $Sc(4N)$ theory:
   gauge $C_2$ and $B_2'$ the codimension-1 defect implementing $\mathbb{Z}_2^{(0)}$ becomes non-invertible.

The first case is already presented in detail in the main text in section 8.

### B.1.1  Non-Invertible Symmetries of $PO(4N)$ YM

We can obtain a codimension-2 non-invertible defect by gauging $C_2$ and $A_1$. We denote the corresponding dynamical fields by $c_2$ and $a_1$. Here we expect the codimension-2 defect implementing the $\mathbb{Z}_2^{(1),B'}$ symmetry, which we denote as $D(M_2)$, to become non-invertible.

Let us call the non-invertible defect $\mathcal{N}'(M_2)$. Similarly to the previous case of Pin$^+$, we have the fusion

$$\mathcal{N}(M_2) \times \mathcal{N}(M_2) \propto (1 + W(M_2)) \sum_{M_1 \in H_1(M_2, \mathbb{Z}_2)} L(M_1), \tag{351}$$

where $W(M_2) = e^{i\pi \oint_{M_2} c_2}$ and $L(M_1) = e^{i\pi \oint_{M_1} a_1}$. The derivation is completely analogous to the one in the main text.

### B.1.2  Non-Invertible Symmetries in $Sc(4N)$ YM

We can also gauge both the two 1-form symmetries and promote $C_2$, $B_2'$ to dynamical fields $c_2$, $b_2'$. This gives a $Sc(4N)$ theory. Here we expect the codimension-1 defect which implements the outer-automorphism $\mathbb{Z}_2^{(0)}$ symmetry to become non-invertible. Let us denote by $D(M_3)$ the defect implementing the $\mathbb{Z}_2^{(0)}$. Due to the anomaly (264), when we gauge $\mathbb{Z}_2^{(1),C} \times \mathbb{Z}_2^{(1),B'}$ and promote $C_2$ and $B_2'$ to dynamical fields $c_2$ and $b_2'$, we must dress $D(M_3)$ with an appropriate TQFT to maintain gauge invariance. Our proposal is also in the case a BF coupling, and we define

$$\mathcal{N}(M_3) \propto \int \mathcal{D}\phi_1 \mathcal{D}\gamma_1 D(M_3, c_2, b_2') e^{i\pi \int_{M_3} \phi_1 c_2 + \gamma_1 b_2' - \gamma_1 \delta\phi_1}. \tag{352}$$

Imposing $\delta\phi_1 = b_2'$ the variation of the action precisely gives $b_2' c_2$.

Now let us compute the fusion rule between two such operators, which is given by

$$\mathcal{N}(M_3) \times \mathcal{N}(M_3) \propto \int \mathcal{D}\phi_1 \mathcal{D}\gamma_1 \mathcal{D}\tilde{\phi}_1 \mathcal{D}\tilde{\gamma}_1 e^{i\pi \int_{M_3} (\phi_1 - \tilde{\phi}_1)c_2 + (\gamma_1 - \tilde{\gamma}_1)b_2' - \gamma_1 \delta\phi_1 + \tilde{\gamma}_1 \delta\tilde{\phi}_1}$$

$$= \int \mathcal{D}\phi_1 \mathcal{D}\gamma_1 \mathcal{D}\hat{\phi}_1 \mathcal{D}\hat{\gamma}_1 e^{i\pi \int_{M_3} \hat{\phi}_1 c_2 + \hat{\gamma}_1 b_2' - \gamma_1 \delta\hat{\phi}_1 + \hat{\gamma}_1 \delta\phi_1 + \hat{\gamma}_1 \delta\hat{\phi}_1}. \tag{353}$$

Following the same discussion around eq. (273), we obtain

$$\mathcal{N}(M_3) \times \mathcal{N}(M_3) \propto \sum_{\Sigma, \Sigma' \in H_2(M_3, \mathbb{Z}_2)} e^{i\pi \oint_\Sigma c_2 + i\pi \oint_{\Sigma'} b_2'}. \tag{354}$$

We can rewrite the above expression as

$$\mathcal{N}(M_3) \times \mathcal{N}(M_3) \propto \sum_{\Sigma, \Sigma' \in H_2(M_3, \mathbb{Z}_2)} W(\Sigma) V(\Sigma'), \tag{355}$$

where $\Sigma \in H_2(M_3, \mathbb{Z}_2)$ is Poincaré dual to $\hat{\phi}_1 \in H^1(M_3, \mathbb{Z}_2)$, $\Sigma' \in H_2(M_3, \mathbb{Z}_2)$ is Poincaré dual to $\hat{\gamma}_1 \in H^1(M_3, \mathbb{Z}_2)$ and $W(\Sigma)$ and $V(\Sigma')$ are the codimension-two operators generating the 1-form symmetries dual to $\mathbb{Z}_2^{(1),C} \times \mathbb{Z}_2^{(1),B'}$.

## B.2  Non-Invertible Symmetries from Spin$(4N + 2)$ Yang-Mills

The 4d Spin$(4N + 2)$ pure gauge theory has a $\mathbb{Z}_2^{(0)}$ charge conjugation 0-form symmetry, while the 1-form symmetry is $\mathbb{Z}_4^{(1)}$. They form a 2-group [60]

$$\delta B_2 = \text{Bock}(C_2) + A_1 C_2, \tag{356}$$

where $A_1$ is the background for the 0-form symmetry, and $B_2, C_2$ are backgrounds for the two $\mathbb{Z}_2$ factors in the 1-form symmetry, which form an extension to $\mathbb{Z}_4^{(1)}$

$$1 \to \mathbb{Z}_2 \to \mathbb{Z}_4 \to \mathbb{Z}_2 \to 1. \tag{357}$$

The Bock is the Bockstein homomorphism for this extension sequence.

Now let us gauge $B_2$ to go to a $SO(4N+2)$ theory, and turn $B_2$ into a dynamical field $b_2$. Because of the 2-group, the coupling $\int_{M_4} b_2 B_2'$, where $B_2'$ is the background for the dual 1-form symmetry, has the bulk dependency

$$\mathcal{A} = \pi \int_{M_5} \delta b_2 B_2' = \pi \int_{M_5} A_1 C_2 B_2' + \text{Bock}(C_2) B_2'. \tag{358}$$

This is a mixed 't Hooft anomaly in the $SO(4N+2)$ theory. Notice that (358) has an additional piece compared to (264), due to the fact that the short exact sequence $1 \to \mathbb{Z}_2 \to \mathbb{Z}_4 \to \mathbb{Z}_2 \to 1$ does not split. Nevertheless, the discussion is quite similar to that of the Spin($4N$) case, so we work out explicitly only the case in which we gauge $C_2$ and $A_1$ and go to a $PO(4N+2)$ theory. In this case, the defect implementing the $\mathbb{Z}_2^{(1),B'}$ 1-form symmetry becomes non-invertible in $PO(4N+2)$. We denote such defect $D(M_2, c_2, a_1)$ in the presence of the background fields for the two symmetries we are gauging. The non-invertible defect is obtained by dressing $D(M_2, c_2, a_1)$ by an appropriate TQFT which cancels the anomaly (358). Then we define

$$\mathcal{N}(M_2) = \int \mathcal{D}\phi_0 \mathcal{D}\gamma_1 D(M_2, c_2, a_1) e^{i\pi \int_{M_2} \phi_0 c_2 + \gamma_1 a_1 - \phi_0 \delta\gamma_1 + \frac{\delta\tilde{\gamma}_1 - \tilde{c}_2}{2}}, \tag{359}$$

where $\phi_0 \in C^0(M_2, \mathbb{Z}_2)$, $\gamma_1 \in C^1(M_2, \mathbb{Z}_2)$ and $\tilde{\gamma}_1, \tilde{c}_2$ denote the lifts of $\gamma_1, c_2$ to $\mathbb{Z}_4$ cochains.[8]

Now let us compute the fusion rules between two $\mathcal{N}(M_2)$ defects.

$$\mathcal{N}(M_2) \times \mathcal{N}(M_2) = \int \mathcal{D}\phi_0 \mathcal{D}\gamma_1 \mathcal{D}\hat{\phi}_0 \mathcal{D}\hat{\gamma}_1 e^{i\pi \int_{M_2} (\phi_0 - \hat{\phi}_0)c_2 + (\gamma_1 - \hat{\gamma}_1)a_1 - \phi_0 \delta\gamma_1 + \hat{\phi}_0 \delta\hat{\gamma}_1 + \frac{\delta\tilde{\gamma}_1 - \tilde{c}_2}{2} - \frac{\delta\tilde{\hat{\gamma}}_1 - \tilde{c}_2}{2}}. \tag{362}$$

Here we used $D(M_2, a_1, c_2)^2 = 1$ since it obeys the $\mathbb{Z}_2$ fusion rules. Making the change of variables $\varphi_0 = \phi_0 - \hat{\phi}_0$ and $\epsilon_1 = \gamma_1 - \hat{\gamma}_1$, we obtain

$$\mathcal{N}(M_2) \times \mathcal{N}(M_2) = \int \mathcal{D}\phi_0 \mathcal{D}\gamma_1 \mathcal{D}\varphi_0 \mathcal{D}\epsilon_1 e^{i\pi \int_{M_2} \varphi_0 c_2 + \epsilon_1 a_1 - \phi_0 \delta\epsilon_1 - \varphi_0 \delta\gamma_1 + \varphi_0 \delta\epsilon_1 + \frac{\delta\tilde{\epsilon}_1}{2}}. \tag{363}$$

The equations of motion of $\phi_0$ and $\gamma_1$ impose $\delta\epsilon_1 = 0$ and $\delta\varphi_0 = 0$ (mod 2). Notice that then last term gives $\delta\tilde{\epsilon}_1/2 = \text{Bock}(\epsilon_1)$. Integrating $\phi_0$ and $\gamma_1$ out and collecting the non trivial terms, we are left with

$$\mathcal{N}(M_2) \times \mathcal{N}(M_2) \propto \sum_{\varphi_0 \in H^0(M_2, \mathbb{Z}_2), \epsilon_1 \in H^1(M_2, \mathbb{Z}_2)} e^{i\pi \int_{M_2} \varphi_0 c_2 + \epsilon_1 a_1 + \text{Bock}(\epsilon_1)}. \tag{364}$$

We can rewrite this as

$$\mathcal{N}(M_2) \times \mathcal{N}(M_2) \propto \left(1 + e^{i\pi \oint_{M_2} c_2}\right) \sum_{M_1 \in H_1(M_2, \mathbb{Z}_2)} e^{i\pi Q(M_1)} e^{i\pi \oint_{M_1} a_1}, \tag{365}$$

---

[8]Notice that there are other choices of TQFTs that cancel the anomaly, and in particular the Bock part. For example, we could use the TQFT

$$\mathcal{I}_2 = \phi_0 \cup c_2 + \gamma_1 \cup a_1 - \phi_0 \cup \delta\gamma_1 + \gamma_1 \cup \gamma_1 + \gamma_1 \cup_1 \delta\gamma_1, \tag{360}$$

since also in this case

$$\delta\mathcal{I}_2 = \delta\gamma_1 \cup a_1 - \delta\gamma_1 \cup_1 \delta\gamma_1 = c_2 \cup a_1 - Sq^1(c_2) = c_2 \cup a_1 - \text{Bock}(c_2). \tag{361}$$

One can check that using this TQFT we obtain the same fusion rules.

where $M_1$ is the Poincaré dual of $\epsilon_1 \in H^1(M_2, \mathbb{Z}_2)$ and we defined $Q(M_1) = \oint_{M_2} \text{Bock}(\epsilon_1) = \oint_{M_2} \epsilon_1 \cup \epsilon_1$. This additional phase is non-trivial only on non-orientable manifolds.

The case in which we gauge $A_1$ and $B_2'$, hence obtaining a $\text{Pin}^+(4N+2)$ theory, is very similar. Indeed, notice that we can rewrite the anomaly as

$$\pi \int_{M_5} A_1 C_2 B_2' + \text{Bock}(C_2)B_2' = \pi \int_{M_5} A_1 C_2 B_2' + C_2 \text{Bock}(B_2'). \tag{366}$$

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
