# Peer review of "Non-Invertible Higher-Categorical Symmetries"

_SciPost Physics, doi:SciPost Phys. 14, 007 (2023)_

## Round 2 · Referee Report · Anonymous (Referee 1) · 2022-8-28

Report

In this paper the authors construct QFTs with non-invertible symmetries in dimensions $\ge$ 3 in two ways. The main approach is to gauge a discrete 0-form symmetry which acts nontrivially on a higher-form symmetry, and the alternative approach is to start from a system with a specific nontrivial mixed anomaly and gauging an appropriat esubgroup. The applicability of the two approaches overlap, and several examples were studied from both points of view.

The discussions are clear and detailed, and the referee thinks that the paper can be published after the following points are taken care of:

  1. The paper is too long. Having many examples worked out is not in general a bad thing, but in the case of this paper (and other papers from the same collaboration) it is on the verge of becoming too many. The referee would like to remind the authors that there is no need to publish all examples the authors happen to have worked out! The main point of the paper can be illustrated by a smaller number of examples.

  2. A related issue is the placement of the Appendix A. In other parts of the paper, the detailed derivations as in the Appendix A were incorporated directly in the main text. Why did the authors choose to separate the content of the Appendix A and only the Appendix A? The authors should either incorporate Appendix A into the main text, or separate detailed derivations in other parts of the main text into individual Appendices.

  3. In the introduction, the authors describes the first approach fully, but the authors only refer to the second approach as the approach used in [17]. To aid the reader, the authors should briefly explain how non-invertible symmetries arise in this second approach, using maybe about half a page.

  4. In the general discussion of Sec.2, the referee thinks that the higher categories can be shifted by one level, by considering different d-dimensional theories separated by topological domain walls at the same time. Then, the objects are the different theories, the 1-morphisms are topological codimension-1 walls separateing them, the 2-morphisms are topological codimension-2 walls, etc.

For example, take a 2d theory T with a fusion category C as its symmetry. Then you can take various gauged versions as objects, topological domain walls separating them as 1-morphisms, etc. The authors already have this structure on their surface defects, and it is not clear to the referee why they did not also allow this extension on the "0-th level".

  1. In p.9, the authors to the number of vacua on a defect $D_{d-1}$. A definition (which should be valid for topological walls in a non-topological theory) should be provided.

  2. In p.31, there's a sentence saying "These line operators themselves organize themselves into..." which does not sound right. It is OK to write an extremely long paper, but it has to be properly proofread.

  3. In p.33, the authors use the notation $D/A$ to denote a defect $D$ whose algebra $A$ is gauged; this notation can already be introduced in Sec. 3 where the authors discuss condensations on defects.

  4. Also, at this point, the fusion of two surface defects is found to contain a surface defect of type $D/A$. Then, the authors should discuss how to determine the fusion $D/A \otimes D'/A'$; this will presumably be given by $(D\otimes D')/(A\boxtimes A')$ but it needs to be discussed nonetheless. Otherwise an untrained reader will be confused whether the fusion rules presented by the authors close among themselves or not.

  5. In Sec.5.6 and elsewhere, where the authors discuss continuous non-invertible symmetries, they should stress that they only consider flat backgrounds. Relatedly, they should explain physical interpretations of the simple objects (5.75) (as flux tubes carrying the holonomy $\theta$ around it, presumably). The same can be said to the 4d case, where the phrase "Gukov-Witten operators" appears in Sec.8.5 but not in Sec.6.4.

  6. In p.50, the referee thinks it is wrong to write the set of simple objects modulo condensations as $\mathcal{C}^\text{ob}$. Shouldn't $\mathcal{C}^\text{ob}$ stand the set of simple objects, not modulo condensations?

  7. Just before Sec.7.1, there is a stray sentence fragment "we do." Again, the authors should proofread their own paper.

  8. In p.80, the authors should quickly give a derivation of the equation (8.5), and explain why it is equivalent to a 2-group of the form $0\to \mathbb{Z}_2^{(1)}\times \mathbb{Z}_2^{(1)} \to \Gamma\to \mathbb{Z}_2^{(0)}\to 0$ where the extension class is zero but the 0-form part acts nontrivially on the 1-form part by switching the two $\mathbb{Z}_2^{(1)}$ factors.

  9. At the beginning of Sec.8.4, the comma in "theories,can" should be removed.

  10. In (8.42), $w_1$ was denoted as $A_1$ up to this point.

---

## Round 2 · Referee Report · Chi-Ming Chang (Referee 2) · 2022-9-3

Strengths

1- This paper formulates a set of rules for fusion and condensation of topological defects in general dimensions phased in the language of the higher category.

2- This paper discovers many non-invertible topological defects in many examples in 3, 4, 5, and 6 dimensions by applying the rules.

3- In section 8, the authors develop an alternative approach using background gauge fields and obtain the same results. This gives a strong cross-check on both methods.

Report

This paper contains many interesting new results. I recommend the publication of this paper on SciPost.

Requested changes

1- Above (2.9), it states that the fusion of 1-morphisms is only defined when $d\ge 3$. I do not understand why this condition is needed, as I do not see any problem with fusing two topological local operators on a topological defect line in 2d. The authors could add some explanation on this point.

2- The quantum dimension on both sides of the equation (4.18) does not match when the dimension of the representation R is greater than one. I think a factor of $\dim(R)$ should be included on the right-hand side. Similarly, I think the defects on the right-hand side of (4.15) could have nontrivial multiplicity in general.

3- Around (4.25), it is stated without explanation that the algebra object of the generalized gauging on top of $D_2^{(O'')}$ is given by (4.25). After some thought, I realized that (4.25) follows from (3.1) together with the fact that the fusion number of the fusion $D_1^{(O''),(OO')}\otimes D_1^{(OO'),(O'')}$ equals the dimension of the vector space associated with the trivalent junction in Figure 24. However, this point is not very obvious to me. I think the author could add some explanation to it.

---

## Round 3 · Referee Report · Chi-Ming Chang (Referee 2) · 2022-9-9

Report

The authors have addressed all the comments in my previous report. I recommend this paper for publication on SciPost.

---

## Round 3 · Referee Report · Anonymous (Referee 1) · 2022-9-9

Report

The improvements made by the authors were appropriate and satisfactory. (I apologize for the missing words and typos in the point 5. The authors correctly guessed what I meant and made the necessary modifications.) The paper can now be recommended to be published.

---

## Round 3 · Author Response

REFEREE 1:

We thank the referee for their very careful reading and insightful comments and suggestions.
We have implemented changes according to the referee's suggestion, as detailed below.

1.) We very much appreciate the comment by the referee that the paper is long. In the new version we have tried to streamline the presentation and moved some of the examples from section 5 and 6 to the appendices. The reason for providing such a large number of examples is, however, that each illustrates a somewhat different point, which -- in subsequent works following this paper -- were used by various other authors.
The outer automorphism gauging of Spin$(4N+2)$ and Spin$(4N)$ are indeed similar and we retained only one in the main text.
However, the non-abelian gauging, as well as $O(2)$ gauge theory examples reveal some distinct features, which we would like to highlight and thus retain in the main text.

2.) The decision to relegate most of the details of section 8 to an appendix is largely due to the fact that the conceptual point of this construction is due to the paper by Kaidi, Ohmori, Zheng (KOZ), and thus not original to this paper. We do extend some of their analysis to other theories, including the use of other minimal TQFTs that are used to stack the invertible defects with, but conceptually the main points were made in KOZ. However we do think that presenting the results is very insightful as they provide an independent cross-check to some of our results in section 6.

3.) The new version includes a half page summary of the method based on mixed anomalies.

4.) This is a very nice observation and we very much agree with this, however the motivation of the paper is to study the symmetry category of a single theory, rather than the space of all theories. We in fact hope to return to the latter question in a future paper.

5.) We are unsure what the referee was trying to say here, clearly there is some typo in the report... We are guessing that the question is to define what the number of vacua for a defect is. This is number of topological local operators.

6.) 11.) 13.)14.) These are all implemented in the new version.

7.) That is good point and we added the definition of D/A already in section 3.2.

8.) Regarding the fusion of D/A:
This in fact requires further technology, which was developed by a subset of us (LB, SSN) with Wu, and is not included in this paper. We have added a forward reference to three papers, which subsequently have elucidated this point, see right after (4.26).
The full symmetry category will of course close on itself, however since in this paper we do not discuss (or had the knowledge to compute) the fusion of condensation defects, the analysis is limited to fusion of simple objects modulo condensation.

9.) We added comments in sections 5 and 6 to clarify the interpretation of the operators. Also we stressed the flatness of the backgrounds.

10.) We clarified that the notation used here -- we define $\mathcal{C}^{\text{ob}}$ as simple objects modulo condensation. In the full symmetry category the simple objects would include the condensation defects as well, however as we explained under 8.) we discuss only a subset of fusions here, which excludes condensation defects.

We appreciate that points 8./10.) are confusing, and have thus added a -- hopefully -- clarifying remark in the introduction as well.

12.) We added some explanations for this.

Again, thank you very much for the detailed reading and very useful suggestions for improvement!

REFEREE 2:

We thank Prof. Chang for his report and questions regarding our paper. Let us respond to the three queries that he made:

1.) The key point to distinguish here is composition versus fusion. What the referee has in mind is composition, which is indeed defined also for other dimensions. Fusion on the other hand, as we define it, requires the existence of 2-morphisms that live at the junction of 1-morphisms. In the case of 3d, these would be local operators (2-morphisms).

2.) In the general section 4 we restrict our analysis to abelian groups, in which case this point does not arise. Note however that in section 5.3 we do provide a non-abelian example, and indeed there is a deviation from this formula in (4.19), which is applicable for abelian groups. For example the fusion in (5.51). We added another clarification about this in the paper, stating the restriction to abelian groups in section 4.

3.) We have added some comments to explain this.

---

## Editorial Decision

published